# Perturbative unitarity and the wavefunction of the Universe

Soner Albayrak[1,2]⋆, Paolo Benincasa[3]† and Carlos Duaso Pueyo[1,4]‡

**1** Institute of Physics, University of Amsterdam, 1098 XH, The Netherlands
**2** Center for Theoretical Physics, National Taiwan University, Taipei 10617, Taiwan
**3** Max-Planck-Institut für Physik, Werner-Heisenberg-Institut, D-80805 München, Germany
**4** Department of Applied Mathematics and Theoretical Physics, University of Cambridge, Wilberforce Road, Cambridge, CB3 0WA, UK

⋆ s.albayrak@uva.nl , † pablowellinhouse@anche.no , ‡ cd820@cam.ac.uk

## Abstract

Unitarity of time evolution is one of the basic principles constraining physical processes. Its consequences in the perturbative Bunch-Davies wavefunction in cosmology have been formulated in terms of the cosmological optical theorem. In this paper, we re-analyse perturbative unitarity for the Bunch-Davies wavefunction, focusing on: $i$) the role of the $i\epsilon$-prescription and its compatibility with the requirement of unitarity; $ii$) the origin of the different "cutting rules"; $iii$) the emergence of the flat-space optical theorem from the cosmological one. We take the combinatorial point of view of the cosmological polytopes, which provide a first-principle description for a large class of scalar graphs contributing to the wavefunctional. The requirement of the positivity of the geometry together with the preservation of its orientation determine the $i\epsilon$-prescription. In kinematic space it translates into giving a small negative imaginary part to all the energies, making the wavefunction coefficients well-defined for any value of their real part along the real axis. Unitarity is instead encoded into a non-convex part of the cosmological polytope, which we name *optical polytope*. The cosmological optical theorem emerges as the equivalence between a specific polytope subdivision of the optical polytope and its triangulations, each of which provides different cutting rules. The flat-space optical theorem instead emerges from the non-convexity of the optical polytope. On the more mathematical side, we provide two definitions of this non-convex geometry, none of them based on the idea of the non-convex geometry as a union of convex ones.

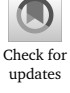

# 1   Introduction

Our understanding of physical phenomena relies on the fundamental principles of locality of the interactions as well as causality and unitarity of time evolution. In particle physics, they fix all the possible three-particle couplings [1,2], the consistent interacting theories with a finite number of particles [1–5], charge conservation and the equivalence principle [1,6] among other fundamental results. All these theorems manifest themselves as consequences of constraints on the structure of the S-matrix elements: locality fixes both the source of the singularities and their type [7]; unitarity is reflected into the factorisation properties as well as the positivity of their coefficients [8–10]; finally, causality is the principle whose imprint is probably the least understood, but yet it reflects itself in their analytic structure [11] and, in particular, in terms of the Steinmann relations which require that double discontinuities in partially overlapping channels vanish in the physical region [12–20].

That level of understanding is not yet available in expanding universes, and just recently we have begun to get insights on how these fundamental principles are encoded into cosmological processes. In this context, the relevant observable is the wavefunction of the universe,[1] whose squared modulus provides the probability distribution of field configurations at the late-time boundary that allows to compute any type of correlation of operators built out of such fields. Considering states with a flat-space counterpart, the perturbative wavefunction turns out to reduce to (the high energy limit of) the flat-space scattering amplitude as the sheet in kinematic space identified by the vanishing locus $E_{\text{tot}} := \sum_{j=1}^{n} |\vec{p}_j|$ is approached, $\vec{p}_j$ being the momentum of the external $j$-th state [21–23].[2] Also, there are other vanishing loci $E_{\mathfrak{g}} := \sum_{s \in \mathcal{V}_{\mathfrak{g}}} |\vec{p}_s| + \sum_{e \in \mathcal{E}_{\mathfrak{g}}^{\text{ext}}} |\vec{p}_e|$, involving the total energy of a subprocess, where the perturbative wavefunction factorises into a lower-point scattering amplitude and a linear combination of the same lower-point wavefunction computed for negative and positive energy of the internal state [24,25] – see Figure 1. Despite these properties have been used to reconstruct the wavefunction in a graph-by-graph fashion [25,26] and to recover some known theorems [25], their origins in terms of fundamental principles still remain obscure. This is also the case for the recently proven Steinmann-like relations [27], whose potential connection to causality has not been demonstrated yet. Only recently, inspired by [28,29], a non-relativistic notion of causality and its relation to the analytic structure of the flat-space wavefunction has been explored [30].

---

[1]The wavefunction of the universe can be considered as an *observable* in the same, loose, way as the S-matrix: none of them are quantities which can actually be detected in an observation or measured in an experiment; nevertheless, they share the very same *physical* properties, such as gauge invariance, as the actual observable; and, they turn out to be simpler, more primitive quantities that allow to extract a great deal of physical information as well as to compute the observables themselves.

[2]Importantly, notice that all the moduli $|\vec{p}_j|$, which with an abuse of language we will refer to as *energies*, are positive; hence, the locus $E_{\text{tot}} = 0$ can be reached only via analytic continuation outside of the physical region.

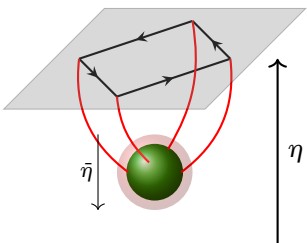 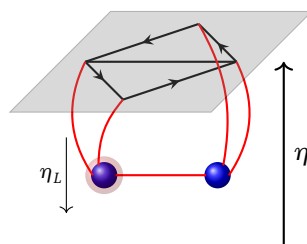

Figure 1: Singular loci in kinematic space for the Bunch-Davies wavefunction. As the total energy of the process or of any subprocess vanishes the Bunch-Davies wavefunction becomes singular. The singularities can be understood as the sheets in kinematic space where the wavefunction does not vanish as the center-of-mass time of the full process (left) and subprocess (right) are taken to early times. These loci live outside of the physical sheet and the coefficients of the singularities respectively correspond to the high-energy limit of the flat-space scattering amplitude for the full process and a factorisation into the scattering amplitude for the subprocess and a lower-point/lower-loop wavefunction associated to the complementary subprocess [31].

In recent years, the first steps have been taken towards the understanding of the imprint of unitarity into the wavefunction and correlators, both perturbatively [32–38] and non-perturbatively [39,40].[3] For the perturbative wavefunction, a *cosmological optical theorem* has been formulated [32], providing constraints on the wavefunction from the unitarity requirement for the evolution operator, as well as sets of cutting rules [35–38]. Such cutting rules can be schematically written as

$$\Delta\psi_n := \psi_n(|\vec{p}_j|, \hat{p}_j \cdot \hat{p}_k) + \psi_n^\star(-|\vec{p}_j|, \hat{p}_j \cdot \hat{p}_k) = -\sum_{\text{“cuts”}} \psi_n, \qquad (1)$$

where $\hat{p}_j := \vec{p}_j/|\vec{p}_j|$ is the unit momentum vector for the $j$-th state and $\hat{p}_j \cdot \hat{p}_k$ parametrise the angles between the momenta of the $j$-th and $k$-th states.

Despite the fact that the equation (1) is reminiscent of the cutting rules for flat-space scattering amplitudes, there is a very important difference: while in flat-space the cuts are directly related to the discontinuities along the scattering amplitudes' branch cuts, the left-hand-side of (1) *is not* a discontinuity, and the cosmological cutting rules (1) represent a functional identity for the wavefunction instead. Interestingly, this important difference can be explained with the fact that, contrarily to the flat-space scattering amplitude case, all the wavefunction branch cuts live outside of the physical region: because of the Bunch-Davies condition, the physical region is defined by the *energies* to be positive and the location of the wavefunction singularities is identified by the vanishing of certain sums of such positive energies, which can never occur in the physical region for non-trivial processes.

More than providing us information about the singularity structure of the wavefunction and its physical content, as it is the case for the optical theorem for flat-space scattering amplitudes, the relation (1) provides a representation for the wavefunction of the universe in terms of lower-point wavefunctions as well as an auxiliary quantity, $\psi_n^\dagger(-|\vec{p}_j|, \hat{p}_j \cdot \hat{p}_k)$, all of them containing *folded singularities*. Different representations make different features of the

---

[3]As far as locality is concerned, the so called *manifestly local test* has been formulated to provide constraints on the perturbative wavefunction involving massless scalars, spin-2 state, and *manifestly local* interactions, *i.e.* interactions that are either polynomial or have (positive) derivatives [34]. In flat-space, the notion of locality is intimately related to the cluster decomposition principle [7]; however, it is known that massless and light states do not satisfy such a principle in expanding universes: the wavefunction undergoes a branched diffusion process [41] with cluster decomposition just on the separate branches, and an avatar of such a structure is given by the property of *ultrametricity* [42–45].

wavefunction manifest [46]: as old-fashioned-perturbation theory exposes both its *physical* singularities and its underlying combinatorial structure [24] and the frequency representation shows a recursive structure at tree level [24], this new representation makes unitarity manifest at the expense of introducing spurious singularities. Because the r.h.s. of (1) encodes both $\psi_n(|\vec{p}_j|, \hat{p}_j \cdot \hat{p}_k)$ and $\psi_n^\dagger(-|\vec{p}_j|, \hat{p}_j \cdot \hat{p}_k)$, which contain different types of singularities, the cosmological cutting rules can allow to compute the perturbative wavefunction only if they are supplemented by the additional requirement of the absence of such folded singularities.[4] In addition, the r.h.s. of (1) is not unique, *i.e.* there exist several equivalent ways to perform the cuts [37]. Said differently, $\Delta\psi_n$ can be decomposed as different but equivalent sums of terms, all of which are interpretable as "cut" diagrams. It is therefore desirable to have an invariant way to describe and understand the imprint of unitarity in the wavefunction.

Finally, the cosmological optical theorem in (1) understood as functional identity does not reproduce the flat-space cutting rules on the total energy conservation sheet: the "cuts" do not show any total energy singularity and hence the r.h.s. of (1) vanishes as the total energy is going to zero. Another way to understand this fact is that the coefficient of the total energy singularity in $\psi_n(|\vec{p}_j|)$ and $\psi_n^\dagger(-|\vec{p}_j|)$ is the same up to a sign and, therefore, the flat-space limit of their sum looks trivial. As we will show, this issue is related to the subtleties in the $i\epsilon$-prescription or, in other words, with the distributional interpretation of (1).

A framework that allows to address both issues is provided by the so-called *cosmological polytopes* [24, 26]. They are a special class of positive geometries[5] defined in projective space which has its own intrinsic definition, with no reference to neither space-time nor Hilbert space, and provides a first-principle combinatorial definition for the wavefunction. They are characterised by a *canonical form* with logarithmic singularities on, and only on, the boundaries of the polytopes. Such singularities are in $1-1$ correspondence with the singularities of the wavefunction, with the canonical form – modulo the standard measure of the projective space where the cosmological polytope lives – providing a Feynman graph $\mathcal{G}$ contribution to the wavefunction. Such singularities are in correspondence to the vanishing loci $E_{\text{tot}} := \sum_j |\vec{p}_j|$ and $E_{\mathfrak{g}} := \sum_{s\in\mathcal{V}_{\mathfrak{g}}} |\vec{p}_s| + \sum_{e\in\mathcal{E}_{\mathfrak{g}}^{\text{ext}}} |\vec{p}_e|$, which identify the facets of the cosmological polytopes. The total energy singularity identifies the *scattering facet,* which is a polytope living on a codimension-one boundary of the cosmological polytope and whose canonical form returns the relevant scattering amplitude. Interestingly, its vertex structure makes the flat-space unitarity manifest, as the facets of such a polytope factorise into two lower dimensional scattering facets and a simplex encoding the Lorentz invariant phase-space measure: this is the combinatorial formulation of the cutting rules, which encode flat-space unitarity [48].

What is then the combinatorial statement for cosmological unitarity? Is it possible to understand it prescinding of the several ways in which the cuts can be performed? And how is it related to the flat-space one?

As observed earlier, the cosmological cutting rules (1) provide a class of novel representations for the wavefunction, all of them involving the auxiliary quantity $\psi_n^\dagger(-|\vec{p}_j|, \hat{p}_j \cdot \hat{p}_k)$. As the wavefunction contribution $\psi_{\mathcal{G}}$ of a graph $\mathcal{G}$ is related to the canonical form of the cosmological polytope $\mathcal{P}_{\mathcal{G}}$, as well as to the volume of the dual polytope $\widetilde{\mathcal{P}}_{\mathcal{G}}$, any representation for $\psi_{\mathcal{G}}$ can be obtained from the *signed triangulations* of $\mathcal{P}_{\mathcal{G}}$ and $\widetilde{\mathcal{P}}_{\mathcal{G}}$: the canonical form of $\omega(\mathcal{Y}, \mathcal{P}_{\mathcal{G}})$ is then given as the sum of the canonical forms of the collection of polytopes which signed-triangulate $\mathcal{P}_{\mathcal{G}}$, consequently providing a decomposition for the wavefunction.

The previous questions can then be reformulated by asking whether there exists the possibility of canonically identifying all those triangulations of $\mathcal{P}_{\mathcal{G}}$ such that the canonical form of one of the elements returns $\psi_{\mathcal{G}}^\dagger(-|\vec{p}_j|, \hat{p}_j \cdot \hat{p}_k)$, or equivalently, if there is an intrinsic definition for a polytope whose canonical form encodes $\Delta\psi_{\mathcal{G}}$ and whose triangulations return all the

---

[4]We thank Austin Joyce for discussions about this point.

[5]For a general discussion of positive geometries which prescinds of any physical interpretation, see [47].

possible ways to perform the cuts of the wavefunction.

In this paper, we provide a positive answer to such a question. Surprisingly enough, the polytopes encoding $\Delta\psi_{\mathcal{G}}$ are *non-convex*. Nevertheless, they also have an intrinsic definition and can be obtained from a related convex polytope by smoothly moving a set of its vertices inside $\mathcal{P}_{\mathcal{G}}$. Geometrically, the absence of a total energy singularity for the cosmological optical theorem is reflected by the fact that the scattering facet of $\mathcal{P}_{\mathcal{G}}$ is not a facet of the optical polytope $\mathcal{O}_{\mathcal{G}}$. However, despite not being a facet, the intersection between $\mathcal{O}_{\mathcal{G}}$ and the hyperplane $\mathcal{W}^{(\mathcal{G})}$ defined by $E_{\text{tot}} = 0$ turns out not to be empty in higher codimensions. This suggests that the flat space optical theorem is encoded in the cosmological one, and therefore the total energy limit presents some subtlety. As we will show, the flat-space cutting rules are beautifully encoded into the structure of $\mathcal{O}_{\mathcal{G}} \cap \mathcal{W}^{(\mathcal{G})}$.

The paper is organized as follows. In Section 2, we review the current understanding of perturbative unitarity for cosmological processes and the associated cutting rules, complementing it with an extensive discussion of the $i\epsilon$-prescription. We also provide novel, holomorphic, cutting rules. In Section 3, we switch gears by reviewing the combinatorial description of the Bunch-Davies wavefunction in terms of cosmological polytopes, and in Section 4 we discuss how the $i\epsilon$-prescriptions emerge in this context along with their relation to the positivity of the geometry. Section 5 discusses the combinatorial formulation of the cosmological optical theorem. It turns out to be encoded into non-convex polytopes, the *optical polytopes $\mathcal{O}_{\mathcal{G}}$*, contained into the cosmological polytope associated to the same graph. We provide two invariant definitions for it: one as a limit of a convex polytope, and the other as a result of compatibility conditions. The cosmological optical theorem then arises as the equivalence between a certain polytope subdivision of $\mathcal{O}_{\mathcal{G}}$ and the triangulations of $\mathcal{O}_{\mathcal{G}}$. We show how the flat-space cutting rules emerge from the cosmological ones, as a special codimension-2 boundary of $\mathcal{O}_{\mathcal{G}}$. Section 6 is devoted to the conclusion and outlook.

## Summary of results

As the paper presents some proofs which are somehow technical, we consider useful, for the sake of clarity, to highlight here the main results, organised conceptually.

**The $i\epsilon$-prescription.** The wavefunctional of the universe shows divergences in the infinite past, both perturbatively and non-perturbatively, and they need to be regularised. The usual regularisation prescription deforms the contour of the time integration around infinity by a small negative imaginary part: $-\infty \longrightarrow -\infty(1 - i\epsilon)$, $\epsilon > 0$. This makes the expression convergent for positive external energies but turns out to break unitarity. We describe a class of $i\epsilon$ prescriptions obtained by analytically continuing the energies to become complex with a negative imaginary part. This prescription does not break unitarity and the wavefunctional becomes convergent for the real part of the energies running along the full real axis, making the analytic continuation of the energies outside the physical region well defined. We also show that the energies associated to internal propagators have a natural $i\epsilon$-prescription. All of them come naturally in the combinatorial language of the cosmological polytopes from the requirement of positivity of the geometry and agreement with its orientation. See Sections 2 and 4.

**An invariant formulation of perturbative unitarity.** Perturbative unitarity manifests itself in the cosmological optical theorem, which relates the wavefunctional to its complex conjugate evaluated at negative external energies. In the combinatorial language of the cosmological polytope, this information is encoded in a non-convex part of the cosmological polytope:

the optical polytope. We provide a characterisation for it in terms of compatibility conditions, which translate into conditions on the multiple discontinuities. The left-hand-side of the cosmological optical theorem emerges as a specific polytope subdivision, via the hyperplane that contains the scattering facet of the cosmological polytope (which *is not* a facet of the optical polytope). The right-hand-side of the cosmological optical theorem, instead, is obtained as triangulations of the optical polytope. This provides a combinatorial-geometric origin for the plethora of "cutting rules" that can be written, going beyond the ones proposed in the original cosmological optical theorem [32], the ones obtained exploiting the properties of the bulk-to-bulk propagators [35,37], and the holomorphic ones formulated in Section 2 of this paper by writing the hermitian conjugate of the transfer operator as a perturbative series in the transfer operator itself – see Section 5.3. The optical polytope provides a description for a universal integrand which needs to be integrated over the external energies in order to inform about the actual wavefunction coefficients. We provide evidences of how the cutting rules obtained from all the triangulations map once the integration is performed: the physical singularities and zeroes appear to be mapped into physical singularities and zeroes of the integrated functions, while the spurious singularities get mapped into spurious singularities. The cutting rules assume the form (124) of the sum of products of functions of the physical singularities, with spurious singularities appearing to make the arguments of these functions dimensionless, and are associated to the specific triangulation. Triangulations with no spurious singularities translate into a dependence on a scale which signals the appearance of a spurious infra-red singularity – see Section 5.4.

**The emergence of the flat-space optical theorem.** We spell out how the flat-space optical theorem emerges from the cosmological one, *i.e.* how the latter encodes the imaginary part of the flat-space scattering amplitudes, and its equivalence to the flat-space cutting rules. After a little algebra, it can be obtained via the careful implementation of the $i\epsilon$-prescription that we describe in this paper. However, it is more transparently encoded into the non-convex structure of the optical polytope, in particular in its codimension-2 intersection with the hyperplane containing the scattering facet of the cosmological polytope. The flat-space cutting rules than can be written in terms of the canonical form of this intersection (132) – see Section 5.5.

**The mathematical side: non-convex polytopes.** Non-convex polytopes are usually defined as union of convex polytopes, and their triangulations require the use of either new vertices or overlapping simplices [49,50]. We provide two alternative, invariant, definitions. The first one is as a smooth limit of a convex polytope: it comes equipped with a canonical form which is the limit of the canonical form of the convex polytope. The second definition is in terms of compatibility conditions, which identify all the higher-codimension boundaries and hence also its adjoint surface, which now can intersect the non-convex polytope in its interior and inside its boundaries – see Section 5.2. To our knowledge, such definitions and characterisations are not known in the, however not extensive, literature on non-convex polytopes.

## 2 Unitarity and the $i\epsilon$-prescription

In this section, we re-examine unitarity in cosmology and its consequences on the structure of the Bunch-Davies wavefunction of the universe.

**The wavefunction of the universe and the evolution operator.** Let us begin with considering a system described by a certain time-dependent Hamiltonian $\hat{H}(\eta)$. Its evolution from early times to the space-like boundary at $\eta_\circ$ is then described by the operator $\hat{U}(\eta_\circ, -\infty)$

which satisfies the first order differential equation

$$i\partial_{\eta_\circ}\hat{U}(\eta_\circ,-\infty) = \hat{H}(\eta_\circ)\hat{U}(\eta_\circ,-\infty). \tag{2}$$

Formally, its solution can be written as

$$\hat{U}(\eta_\circ,-\infty) = \widehat{\mathcal{T}}\left\{\exp\left[-i\int_{-\infty}^{\eta_\circ}d\eta\,\hat{H}(\eta)\right]\right\}, \tag{3}$$

where $\widehat{\mathcal{T}}$ is the time-ordering operator. In cosmology, we are interested in the evolution of our universe from its early stages at $\eta \longrightarrow -\infty$ to the space-like boundary at the end of inflation at $\eta = \eta_\circ \longrightarrow 0^-$. The probability distribution $\mathfrak{P}[\Phi]$ for the state $\langle\Phi|$ at such space-like boundary is given by the squared modulus $|\Psi[\Phi]|^2$ of the wavefunctional of the universe $\Psi[\Phi]$, which is defined as the transition amplitude between the vacuum $|0\rangle$ at early-times and $|\Phi\rangle$ at $\eta = \eta_\circ$:

$$\Psi[\Phi] := \lim_{\eta_\circ\longrightarrow 0^-}\langle\Phi|\hat{U}(\eta_\circ,-\infty)|0\rangle = \lim_{\eta_\circ\longrightarrow 0^-}\mathcal{N}\int_{\phi(-\infty)=0}^{\phi(\eta_\circ)=\Phi}\mathcal{D}\phi\,e^{iS[\phi]}, \tag{4}$$

with the second equality representing the path-integral formulation of such a transition amplitude, where $\phi$ collectively indicates all the modes in the system described by the action $S[\phi]$, and $\mathcal{N}$ is a normalisation constant. As boundary condition at early times, we consider the Bunch-Davies vacuum, which selects the positive energy modes in that limit.

In perturbation theory, the Hamiltonian $\hat{H}(\eta)$ can be split as $\hat{H}(\eta) := \hat{H}_{\text{free}}(\eta) + \hat{H}_{\text{int}}(\eta)$ into its free and interaction parts. Then, (4) can be formally written as

$$\Psi[\Phi] = \Psi_{\text{free}}[\Phi] \times \left\{1 + \sum_{n\geq 2}\int\prod_{j=1}^{n}\left[\frac{d^dp_j}{(2\pi)^d}\Phi(\vec{p}_j)\right]\sum_{L\geq 0}\psi_n^{'(L)}(\vec{p}_1,\ldots,\vec{p}_n)\right\}, \tag{5}$$

where $\Psi_{\text{free}}[\Phi]$ is the wavefunction(al) for the free system, which is computed using the evolution operator $\hat{U}_{\text{free}}$ defined via the free Hamiltonian $\hat{H}_{\text{free}}(\eta)$ and is given by a Gaussian in the fields $\Phi(\vec{p})$ at the boundary

$$\Psi_{\text{free}}[\Phi] := \langle\Phi|\hat{U}_{\text{free}}|0\rangle = e^{iS_{\text{free}}^{(\text{cl})}[\Phi]} = \exp\left\{-\int\prod_{j=1}^{2}\left[\frac{d^dp_j}{(2\pi)^d}\Phi(\vec{p}_j)\right]\psi_2^{(0)}(\vec{p}_1,\vec{p}_2)\right\}, \tag{6}$$

while $\Phi(\vec{p}_j) := \langle\Phi|\vec{p}_j\rangle$, and $\psi_n^{'(L)}(\vec{p}_1,\ldots,\vec{p}_n)$ is given by the sum of all the $n$-point Feynman graphs in momentum space at $L$-loops and its connected part, which we will indicate as $\psi_n^{(L)}(\vec{p}_1,\ldots,\vec{p}_n)$, are the so-called wavefunction coefficients and have support on the total spatial momentum-conserving $\delta$-function which arises as a consequence of spatial translation invariance at the space-like boundary. The term of (5) in curly brackets is obtained as a series expansion of the interaction part of the evolution operator $\hat{U}_{\text{int}}$, which is defined via the interaction Hamiltonian $\hat{H}_{\text{int}}(\eta)$ and is required to be unitary:

$$\hat{U}_{\text{int}}\hat{U}_{\text{int}}^\dagger = \hat{\mathbb{1}} = \hat{U}_{\text{int}}^\dagger\hat{U}_{\text{int}}. \tag{7}$$

The choice of the Bunch-Davies vacuum implies that the modes behave as plane waves with positive frequencies only as the past infinity is approached

$$\phi(\vec{p},\eta) \xrightarrow{\eta\longrightarrow -\infty} f(\eta)e^{iE\eta}, \tag{8}$$

where $E := |\vec{p}| > 0$ indicates the modulus of the momentum, which, with an abuse of language, we will refer to as *energy*; the function $f(\eta)$ is determined by the cosmology. However, as early times are approached, $\eta \longrightarrow -\infty$, the infinite oscillations in (8) make the wavefunction ill-defined and an appropriate regularisation becomes necessary. The usual regularisation prescription is to deform the contour of the time integration at early times, $\eta \longrightarrow -\infty(1-i\epsilon)$ with $\epsilon > 0$. It indeed makes the infinitely oscillating phase (8) decay exponentially in the infinite past and regularises the wavefunction (4). However, it has an important drawback as the regulated evolution operator $\hat{U}_{\text{int}}(0, -\infty(1-i\epsilon))$ is no longer unitary.

**The $i\epsilon$-prescription.** Let us consider the operator $\hat{U}_{\text{int}}^{(\epsilon)}$ defined on a generic $i\epsilon$-deformed contour $\gamma_\epsilon$, as well as its Hermitian conjugate

$$\hat{U}_{\text{int}}^{(\epsilon)} := \widehat{\mathcal{T}}\left\{\exp\left[-i\int_{\gamma_\epsilon} d\eta\, \hat{H}_{\text{int}}(\eta)\right]\right\}, \qquad \hat{U}_{\text{int}}^{\dagger(\epsilon)} := \widehat{\overline{\mathcal{T}}}\left\{\exp\left[+i\int_{\gamma_\epsilon^\star} d\eta\, \hat{H}_{\text{int}}(\eta)\right]\right\}, \qquad (9)$$

$\widehat{\overline{\mathcal{T}}}$ being the anti time-ordering operator. Note that the Hermitian conjugate $\hat{U}_{\text{int}}^{\dagger(\epsilon)}$ is instead defined via the contour $\gamma_\epsilon^\star$. In order for $\hat{U}_{\text{int}}^{(\epsilon)}$ to be unitary, its Hermitian conjugate should coincide with its inverse. This holds if and only if $\gamma_\epsilon^\star = \gamma_\epsilon$. Taking $\gamma_\epsilon = ]-\infty(1-i\epsilon), 0]$, this condition in fact does not hold: such a choice for the deformed contour is not consistent with unitarity. It is interesting to observe that this is a direct consequence of the lack of time-reversal invariance of the physics in cosmology or, more generally, of any process in a space with a space-like boundary.[6] In particular, the inverse of the evolution operator integrated over the path $\gamma_\epsilon = ]-\infty(1-i\epsilon), 0]$ coincides with the Hermitian of the evolution operator integrated over the contour $[0, +\infty(1+i\epsilon)[$:

$$[\hat{U}_{\text{int}}^{(\epsilon)}(0, -\infty(1-i\epsilon))]^{-1} = [\hat{U}_{\text{int}}^{(\epsilon)}(+\infty(1+i\epsilon), 0)]^\dagger. \qquad (12)$$

In order to draw any conclusion about the imprint of unitarity in the wavefunction, it is necessary to have a regularisation of the evolution operator which preserves the unitarity condition $\hat{U}_{\text{int}}^{(\epsilon)}\hat{U}_{\text{int}}^{\dagger(\epsilon)} = \hat{\mathbb{1}} = \hat{U}_{\text{int}}^{\dagger(\epsilon)}\hat{U}_{\text{int}}^{(\epsilon)}$. Such a problem can be bypassed by regularising the Hamiltonian $\hat{H}_{\text{int}}(\eta)$ rather than the integration contour, in such a way that the Hermiticity of the $\hat{H}_{\text{int}}(\eta)$ is preserved [51]:

$$\hat{U}_{\text{int}}^{(\epsilon)} := \lim_{\eta_\circ \longrightarrow 0^-} \widehat{\mathcal{T}}\left\{\exp\left[-i\int_{-\infty}^{\eta_\circ} d\eta\, e^{\epsilon\eta}\hat{H}_{\text{int}}(\eta)\right]\right\}, \qquad (13)$$

which is manifestly unitary. However, an $i\epsilon$-prescription which both provides well-defined expressions and is compatible with unitarity is not unique. Requiring causality in the processes should restrict further the space of possible $i\epsilon$-prescriptions.[7] Despite a full-fledge discussion

---

[6]It is straightforward to understand this if we consider the operator $\hat{U}_{\text{int}}$ for flat-space without fixed-time boundaries and regularised by deforming the integration contour at both past and future infinity

$$\hat{U}_{\text{int}}^{(\epsilon)}(+\infty, -\infty) := \hat{\mathcal{T}}\left\{\exp\left[-i\int_{-\infty(1-i\epsilon)}^{+\infty(1+i\epsilon)} d\eta\, \hat{H}_{\text{int}}(\eta)\right]\right\}. \qquad (10)$$

Its Hermitian conjugate $\hat{U}_{\text{int}}^{\dagger(\epsilon)}$ is then given by

$$\hat{U}_{\text{int}}^{\dagger(\epsilon)}(+\infty, -\infty) := \hat{\mathcal{T}}\left\{\exp\left[+i\int_{-\infty(1+i\epsilon)}^{+\infty(1-i\epsilon)} d\eta\, \hat{H}_{\text{int}}(\eta)\right]\right\} = \hat{\overline{\mathcal{T}}}\left\{\exp\left[+i\int_{-\infty(1-i\epsilon)}^{+\infty(1+i\epsilon)} d\eta\, \hat{H}_{\text{int}}(-\eta)\right]\right\}. \qquad (11)$$

It coincides with the inverse $\hat{U}_{\text{int}}^{-1(\epsilon)}$ provided that the Hamiltonian is invariant under time-reversal: $\hat{H}_{\text{int}}(-\eta) = \hat{H}_{\text{int}}(\eta)$. In the cosmological case, the presence of a space-like boundary at a finite time $\eta = 0$ makes the contour deformation $]-\infty(1-i\epsilon), 0]$ inconsistent with unitarity.

[7]For a discussion of the $i\epsilon$-prescription for the flat-space S-matrix, see [52].

of causality is beyond the scope of the present work, it is important to bear in mind that, among all the possible $i\epsilon$-prescriptions, a subset of them is compatible with physical principles. Here we will be concerned with the ones which are compatible with unitarity. As a final remark, the basic requirement for the introduction of the $i\epsilon$-prescription at this stage is to have a well-defined convergent expression for the wavefunction of the universe. Meanwhile, the Feynman $i\epsilon$-prescription we are accustomed to for the S-matrix, despite not yet having a physical meaning and being also required for having well-defined expressions in the physical region, allows to choose the correct, causal, integration contour for the Feynman propagator. As we will see shortly, the contour deformations which allow to have a well-defined wavefunction of the universe can be thought of as kinematic $i\epsilon$-prescriptions, that are obtained by simply deforming the external kinematics. In the context of the S-matrix, there are not known arguments relating this class of prescriptions to causality nor to the Feynman-$i\epsilon$ [52]. However, a class of unitary cosmological $i\epsilon$-prescriptions turn out to reduce to the Feynman-$i\epsilon$ for the S-matrix on the total energy conservation sheet [48].

**Unitarity and the wavefunction of the universe.** Let $\hat{U}_{\text{int}}^{(\epsilon)}$ be a deformation of $\hat{U}_{\text{int}}$ which preserves unitarity. It defines a regularisation for the contribution of the interactions to the wavefunction of the universe:

$$\Psi_{\text{int}}^{(\epsilon)}[\Phi] = \lim_{\eta_\circ \longrightarrow 0^-} \langle\Phi|\hat{U}_{\text{int}}^{(\epsilon)}(\eta_\circ, -\infty)|0\rangle\,. \tag{14}$$

For a free theory $\hat{U}_{\text{int}}^{(\epsilon)}$ is the identity operator $\hat{\mathbb{1}}$. We can therefore split it into the identity and a transfer operator, as it is customary for the S-matrix

$$\hat{U}_{\text{int}}^{(\epsilon)}(0, -\infty) = \hat{\mathbb{1}} + \hat{V}^{(\epsilon)}(0, -\infty)\,, \tag{15}$$

where $\hat{V}^{(\epsilon)}(0, -\infty)$ is the (regularised) transfer operator encoding the non-trivial interactions. The unitarity of $\hat{U}_{\text{int}}^{(\epsilon)}(0, -\infty)$ then translates into the following condition for $\hat{V}^{(\epsilon)}(0, -\infty)$:

$$\hat{V}^{(\epsilon)}(0, -\infty) + \hat{V}^{\dagger(\epsilon)}(0, -\infty) = -\hat{V}^{\dagger(\epsilon)}(0, -\infty)\hat{V}^{(\epsilon)}(0, -\infty)\,. \tag{16}$$

In terms of the transition amplitudes from the vacuum $|0\rangle$ to some momentum state $\langle[\vec{p}]| := \langle\vec{p}_1 \ldots \vec{p}_n|$, the unitarity condition becomes

$$\langle[\vec{p}]|\hat{V}^{(\epsilon)}|0\rangle + \langle[\vec{p}]|\hat{V}^{\dagger(\epsilon)}|0\rangle = -\langle[\vec{p}]|\hat{V}^{\dagger(\epsilon)}\hat{V}^{(\epsilon)}|0\rangle$$
$$= -\int\left[\frac{d^d q}{(2\pi)^d}\frac{1}{2\text{Re}\{\psi_2^{(0)}(\vec{q})\}}\right]\langle[\vec{p}]|\hat{V}^{\dagger(\epsilon)}|[\vec{q}]\rangle\langle[\vec{q}]|\hat{V}^{(\epsilon)}|0\rangle\,, \tag{17}$$

and can be further written as

$$\langle[\vec{p}]|\hat{V}^{(\epsilon)}(0, -\infty)|0\rangle + \overline{\langle0|\hat{V}^{(\epsilon)}(+\infty, 0)|[\vec{p}]\rangle}$$
$$= -\int\left[\frac{d^d q}{(2\pi)^d}\frac{1}{2\text{Re}\{\psi_2^{(0)}(\vec{q})\}}\right]\overline{\langle[\vec{q}]|\hat{V}^{(\epsilon)}(+\infty, 0)|[\vec{p}]\rangle}\langle[\vec{q}]|\hat{V}^{(\epsilon)}(0, -\infty)|0\rangle\,. \tag{18}$$

The left-hand-side of (18) involves a transition amplitude $\langle[\vec{p}]|\hat{V}^{(\epsilon)}(0, -\infty)|0\rangle$ from the vacuum at past infinity $|0\rangle$ to the state $\langle[\vec{p}]|$ labelled by the set of momenta $[\vec{p}]$ at the boundary as well as the complex conjugate of the transition amplitude $\langle0|\hat{V}^{(\epsilon)}(+\infty, 0)|[\vec{p}]\rangle$ from the $|[\vec{p}]\rangle$ at the boundary to the vacuum $\langle0|$ at some future infinity. The former can be related to the wavefunction coefficient $\psi_n([\vec{p}], [E])$; the latter can be interpreted as the complex conjugate of the transition amplitude $\langle[-\vec{p}]|\hat{V}^{(\epsilon)}(0, -\infty)|0\rangle$ from the vacuum $|0\rangle$ to the state $\langle[-\vec{p}]|$ identified by the very same momenta but with opposite direction (implying also opposite signs

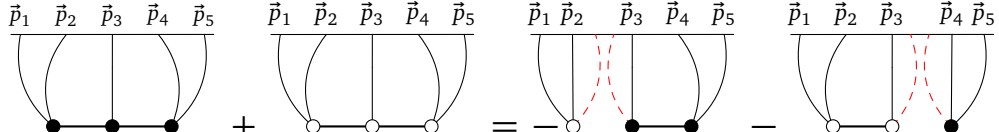

Figure 2: Cutting rules corresponding to the unitarity condition $\hat{V}^{(\epsilon)} + \hat{V}^{\dagger(\epsilon)} = -\hat{V}^{\dagger(\epsilon)}\hat{V}^{(\epsilon)}$. The quantities $\psi(E)$ and $\psi^\dagger(-E)$ are identified by the graphs with black and white sites respectively. The red dashed line indicates an external state with energy $\sigma y_\ell$ and each factor in each term in the rhs has to be summed over $\sigma = \pm$.

for the energies: $[E] \longrightarrow [-E]$) for real fields. Hence it can be related to the wavefunction coefficient $\psi_n^\dagger([-\vec{p}],[-E])$ and (18) acquires the form

$$\langle[\vec{p}]|\hat{V}^{(\epsilon)}(0,-\infty)|0\rangle + \overline{\langle[-\vec{p}]|\hat{V}^{(\epsilon)}(0,-\infty)|0\rangle}$$
$$= -\int \left[\frac{d^d q}{(2\pi)^d}\frac{1}{2\text{Re}\{\psi_s^{(0)}(\vec{q})\}}\right]\overline{\langle[-\vec{p}|\hat{V}^{(\epsilon)}(0,-\infty)|[-\vec{q}]]\rangle}\langle[\vec{q}]|\hat{V}^{(\epsilon)}(0,-\infty)|0\rangle. \quad (19)$$

Despite the unitarity condition (19) is formally generically valid, so far it has a clear and useful interpretation just in perturbation theory. Expanding $\hat{V}^{(\epsilon)}$

$$\hat{V}^{(\epsilon)}(0,-\infty) = -i\int_{-\infty}^0 d\eta\, \hat{H}_{\text{int}}^{(\epsilon)}(\eta) + \frac{(-i)^2}{2!}\mathcal{T}\left\{\int_{-\infty}^0 d\eta_1 \int_{-\infty}^0 d\eta_2\, \hat{H}_{\text{int}}^{(\epsilon)}(\eta_1)\hat{H}_{\text{int}}^{(\epsilon)}(\eta_2)\right\} + \dots,$$

and recollecting the terms of both sides with the same number of Hamiltonian insertions contributing to the same perturbative order, equation (19) can be written directly in terms of the wavefunction coefficients. Let $\mathcal{N} := \{1, \dots, n\}$ be the set of external states, with $\mathcal{L} \cup \mathcal{R} = \mathcal{N}$ be a partition of $\mathcal{N}$, with $n_\mathcal{L} = \dim\{\mathcal{L}\}$ and $n_\mathcal{R} = \dim\{\mathcal{R}\}$. Let also $\mathscr{E}_k$ be a set of $k$ "cut" edges, while $\mathscr{E}$ be the set of edges between the subprocesses containing $\mathcal{L}$ and $\mathcal{R}$. Then

$$\psi_n^{(L)}(E_j, y_e) + \psi_n^{\dagger(L)}(-E_j, y_e)$$
$$= -\sum_{k=1}^L \sum_{\{\mathscr{E}_k\}}\int\left[\prod_{\ell\in\mathscr{E}_k}\frac{d^d q_\ell}{(2\pi)^d}\frac{1}{2\text{Re}\{\psi_2(y_\ell)\}}\right]\left[\psi_{n+2k}^{(L-k)}(E_j, y_\ell; y_{e'}) + \psi_{n+2k}^{\dagger(L-k)}(-E_j, y_\ell; y_{e'})\right]$$
$$- \sum_{\{\mathcal{L}\}\{\mathcal{R}\}}\int\left[\prod_{\ell\in\mathscr{E}}\frac{d^d q_\ell}{(2\pi)^d}\frac{1}{2\text{Re}\{\psi_2(y_\ell)\}}\right]\sum_{\sigma=\pm}\left[\sigma\psi_{n_\mathcal{L}+n_\sharp}^{\dagger(L_\mathcal{L})}(-E_\mathcal{L}, -\sigma y_\ell; y_{e_\mathcal{L}})\right]$$
$$\times \sum_{\sigma=\pm}\left[\sigma\psi_{n_\mathcal{R}+n_\sharp}^{(L_\mathcal{R})}(E_\mathcal{R}, -\sigma y_\ell; y_{e_\mathcal{R}})\right], \quad (20)$$

where $y_\ell := |\vec{q}_\ell|$, $E_{\mathcal{L}/\mathcal{R}}$ are the energies of the external states in $\mathcal{L}/\mathcal{R}$, and $L_{\mathcal{L}/\mathcal{R}}$ is the loop order for the wavefunction at the left/right of the "cut" – see Figure 2 for an illustration. Notice that an equivalent rule can be obtained from (20) by having $\psi$ and $\psi^\dagger$ on the left/right side of the "cut", which is equivalent to the unitarity condition $\hat{U}_{\text{int}}\hat{U}_{\text{int}}^\dagger = \hat{\mathbb{1}}$.

**Holomorphic cutting rules.** Notice also that there is a different type of relation which can be extracted and, in the case of the S-matrix leads to the holomorphic cutting rules [52]. We can formally solve (16) in terms of $\hat{V}^{\dagger(\epsilon)}$, $\hat{V}^{\dagger(\epsilon)} = -\hat{V}^{(\epsilon)}(\hat{\mathbb{1}} + \hat{V}^{(\epsilon)})^{-1}$, insert it in the right-hand-side of (16) and expand it perturbatively

$$\hat{V}^{(\epsilon)} + \hat{V}^{\dagger(\epsilon)} = -\hat{V}^{(\epsilon)}(\hat{\mathbb{1}} + \hat{V}^{(\epsilon)})^{-1}\hat{V}^{(\epsilon)} = -\sum_{c\geq 1}(-1)^{c+1}(\hat{V}^{(\epsilon)})^{c+1}, \quad (21)$$



Figure 3: Holomorphic cutting rules. They are realised just in terms of $\psi(E)$, identified by graphs with black dots, when the "cuts" divide the graph into disconnected subgraphs.

which, in terms of the matrix elements, becomes

$$\langle[\vec{p}]|\hat{V}^{(\epsilon)}|0\rangle + \overline{\langle 0|\hat{V}^{(\epsilon)}|[\vec{p}]\rangle} = -\sum_{c\geq 1}(-1)^{c+1}\int\prod_{j=1}^{c}\left[\frac{d^d q_j}{(2\pi)^d}\frac{1}{2\mathrm{Re}\{\psi_2^{(0)}(\vec{q}_j)\}}\right]\langle[\vec{p}]|\hat{V}^{(\epsilon)}|[\vec{q}_c]\rangle$$
$$\times\prod_{j=1}^{c-1}\langle[\vec{q}_{j+1}]|\hat{V}^{(\epsilon)}|[\vec{q}_j]\rangle\langle[\vec{q}_1]|\hat{V}^{(\epsilon)}|0\rangle. \qquad (22)$$

Proceeding in a similar fashion as in the previous section, the relation (17) can be rewritten in terms of the wavefunction coefficients

$$\psi_n^{(L)}(E_j, y_e) + \psi_n^{\dagger(L)}(-E_j, y_e)$$
$$= -\sum_{k=1}^{L}\sum_{\{\mathscr{E}_k\}}\int\left[\prod_{\ell\in\mathscr{E}_k}\frac{d^d q_\ell}{(2\pi)^d}\frac{1}{2\mathrm{Re}\{\psi_2(y_\ell)\}}\right]\left[\psi_{n+2k}^{(L-k)}(E_j, y_\ell; y_{e'}) + \psi_{n+2k}^{\dagger(L-k)}(-E_j, y_\ell; y_{e'})\right]$$
$$-\sum_{c\geq 1}(-1)^{c+1}\int\left[\prod_{\ell\in\mathscr{E}}\frac{d^d q_\ell}{(2\pi)^d}\frac{1}{2\mathrm{Re}\{\psi_{q_\ell}\}}\right]\prod_{j=1}^{c+1}\sum_{\sigma=\pm}\sigma\psi_{n_{\mathcal{N}_j}+n_{\mathscr{E}_j}}^{(L_{\mathcal{N}_j})}\left(E_{\mathcal{N}_j}, -\sigma y_\ell; y_{e_{\mathcal{N}_j}}\right), \qquad (23)$$

where $\{\mathcal{N}_j\}_{j=1}^{c+1}$ is a partition of the set $\mathcal{N}$ of external states, while $\mathscr{E}_j \subseteq \mathscr{E}$ is the subset of cut edges incident on the subgraph $\mathcal{N}_j$ belongs to. One feature of the right-hand-side of (23) is that the terms that correspond to disconnected subgraphs represent the wavefunction coefficients $\psi(E)$ and do not depend on the hermitian conjugates.

## 3 Cosmological polytopes: A crash course

Let us consider a scalar with a time dependent mass and time-dependent polynomial interactions in a $(d+1)$-dimensional flat space-time:

$$S[\phi] = -\int d^d x \int_{-\infty}^{0} d\eta \left[\frac{1}{2}(\partial\phi)^2 - \frac{1}{2}m^2(\eta)\phi^2 - \sum_{k\geq 3}\frac{\lambda_k(\eta)}{k!}\phi^k\right]. \qquad (24)$$

This action describes a class of toy models which contain conformally-coupled and states with general masses in Friedmann-Robertson-Walker (FRW) cosmologies, upon the identification

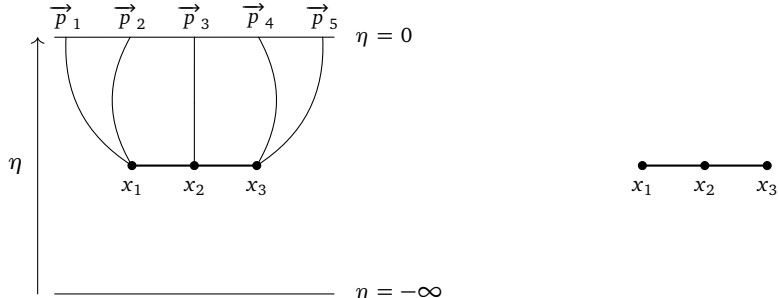

Figure 4: From Feynman graphs to reduced graphs. A Feynman graph contribution to the wavefunction (on the left) is characterised by the external lines reaching the late-time boundary, sites, and edges connecting the sites, respectively representing external states, interactions, and internal states. On the left, we show a Feynman graph that contributes to the wavefunction of the universe. On the right, we depict the associated reduced graph, which is obtained from the Feynman graph by suppressing the external lines [27].

of the functions $m^2(\eta)$ and $\lambda_k(\eta)$ with

$$
m^2(\eta) = m^2 a^2(\eta) + 2d\left(\xi - \frac{d-1}{4d}\right)\left[\partial_\eta\left(\frac{\dot{a}}{a}\right) + \frac{d-1}{2}\left(\frac{\dot{a}}{a}\right)^2\right],
$$
$$
\lambda_k(\eta) = \lambda_k\,\vartheta(-\eta)\,[a(\eta)]^{2+(2-k)(d-1)/2},
\tag{25}
$$

where $m^2$ and $\lambda_k$ appearing in the right-hand-sides are constants, $\xi$ is a parameter which can be either zero or $(d-1)/4d$ depending on whether the scalar is respectively minimally or conformally coupled, " $\cdot$ " is the derivative with respect to the conformal time $\eta$ and $a(\eta)$ is the time-dependent warp factor for the conformally-flat metric

$$
ds^2 = a^2(\eta)\left[-d\eta^2 + \delta_{ij}dx^i dx^j\right], \qquad i,j = 1,\ldots,d,
\tag{26}
$$

with $\eta \in ]-\infty, 0]$ – see [24, 31, 53]. The perturbative wavefunction coefficients $\psi_n^{(L)}(\vec{p}_1,\ldots,\vec{p}_n)$ introduced in (5) can be computed via Feynman graphs. Given a graph $\mathcal{G}$, defined by the sets $\mathcal{V}$ and $\mathcal{E}$ of sites[8] and edges, then the wavefunction coefficient $\widetilde{\psi}_\mathcal{G}$ associated to $\mathcal{G}$ is given by

$$
\widetilde{\psi}_\mathcal{G} = \int_{-\infty}^{0} \prod_{s\in\mathcal{V}}\left[d\eta_s\, i\lambda_k(\eta_s)\,\phi_\circ^{(s)}(\eta_s)\right]\prod_{e\in\mathcal{E}}\widetilde{G}(y_e,\eta_{s_e},\eta_{s'_e}),
\tag{27}
$$

where $\phi_\circ^{(s)}(\eta_s)$ is the product of the bulk-to-boundary propagators stretching from the site $s$ to the boundary, while $\widetilde{G}(y_e,\eta_{s_e},\eta_{s'_e})$ is the bulk-to-bulk propagator with internal energy $y_e$ connecting the sites $s_e$ and $s'_e$. We will be interested in FRW space-times with warp factors of the form $a(\eta) = (-\ell/\eta)^\gamma$, for which the bulk-to-boundary propagators are given in terms of Hankel functions of the second type if $\gamma = 1$, as well as for generic $\gamma$ and the bare parameter $m$ equal zero [31, 53].

For light-states, the mode functions can be obtained as differential operators in energy space acting on the flat-space massless mode functions, *i.e.* exponentials of the type $e^{i|\vec{p}|\eta}$. Using the integral representation for the time-dependent coupling $\lambda_k(\eta)$

$$
\lambda_k(\eta) = \int_{-\infty}^{+\infty} dz\, e^{iz\eta}\,\tilde{\lambda}_k(z),
\tag{28}
$$

---

[8]As the word *vertex* can refer both to an element of a graph and to the highest codimension boundary of a polytope, we reserve it for the latter while for the former we use *site* in order to avoid any language clash.

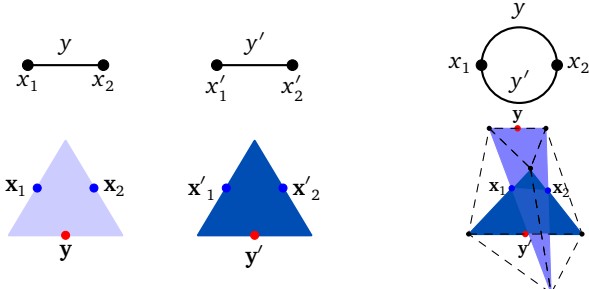

Figure 5: Graphs and cosmological polytopes. Each graph $\mathcal{G}$ can be seen as a collection of 2-site line graphs merged in a subset of their sites. As each two-site line graph is in one-to-one correspondence with a triangle, a general graph $\mathcal{G}$ is associated to a polytope defined as a suitable intersection of such triangles in the midpoints of a subset of their edges. Here we show a collection of two two-site line graphs and the associated triangles as well as the two-site one-loop graph obtained by merging their sites in pairs and the associated cosmological polytope. The dashed lines show the convex hull obtained by intersecting the two triangles, while the coloured areas represent the defining triangles themselves.

the wavefunction $\widetilde{\psi}_{\mathcal{G}}$ for light-states can be written in terms of integro-differential operators acting on a universal integrand $\psi_{\mathcal{G}}(x_s, y_e)$ [53][9]

$$\psi_{\mathcal{G}}(x_s, y_e) = \int_{-\infty}^{0} \prod_{s \in \mathcal{V}} \left[ d\eta_s\, i\, e^{ix_s\eta_s} \right] \prod_{e \in \mathcal{E}} G(y_e, \eta_{s_e}, \eta_{s'_e}), \tag{29}$$

where now $x_s$ is the sum of the energies of the external states at the site $s$, while $G(y_e, \eta_{s_e}, \eta_{s'_e})$ is given by

$$G(y_e, \eta_{s_e}, \eta_{s'_e}) = \frac{1}{2y_e} \left[ e^{-y_e(\eta_{s_e} - \eta_{s'_e})} \vartheta(\eta_{s_e} - \eta_{s'_e}) + e^{+y_e(\eta_{s_e} - \eta_{s'_e})} \vartheta(\eta_{s'_e} - \eta_{s_e}) - e^{+y_e(\eta_{s_e} + \eta_{s'_e})} \right], \tag{30}$$

where the first two terms represent the time-ordered part of the propagator (retarded and advanced, respectively), while the last term is fixed by the boundary condition that the fluctuations vanish at the boundary $\eta = 0$. The wavefunction integrand $\psi_{\mathcal{G}}(x_s, y_e)$ is universal as it is common to any conformally-flat cosmology, whose details are encoded into the functions $\tilde{\lambda}_k(z)$ which play the role of a measure of integration in the space of external energies. The differential operators mentioned earlier change the type of scalar states involved into the process – for further details see [24, 31, 53].

The universal integrand $\psi_{\mathcal{G}}(x_s, y_e)$ depends on the sum of the energies at each site $\{x_s, s \in \mathcal{V}\}$. Consequently, it can be represented as a reduced graph by suppressing the lines associated to bulk-to-boundary propagators. Hence a reduced graph is a collection of $n_s$ sites $\mathcal{V}$ and $n_e$ edges $\mathcal{E}$ connecting them and respectively weighed by the sets of labels $\{x_s, s \in \mathcal{V}\}$ and $\{y_e, e \in \mathcal{E}\}$ – see Figure 4.

A reduced graph $\mathcal{G}$ turns out to be in one-to-one correspondence with a *cosmological polytope*, whose associated canonical form provides the universal integrand $\psi_{\mathcal{G}}(x_s, y_e)$ [24]. Given a reduced graph $\mathcal{G}$ with $n_s$ sites and $n_e$ edges, it can be thought of as a collection of $n_e$ 2-site and 1-edge graphs in which some of the sites have been identified. Each of the 2-site 1-edge graphs can be associated to a triangle living in $\mathbb{P}^{3n_e-1}$ identified by its midpoints given by

---

[9]In a similar fashion, the universal integrand (29) serves as a seed for a differential representation for the wavefunction in de Sitter space involving massless scalars and gravitons [54].

the triple of vectors $\{\mathbf{x}_{s_e}, \mathbf{y}_e, \mathbf{x}_{s'_e}\}$. The collection $\{\mathbf{x}_{s_e}, \mathbf{y}_e, \mathbf{x}_{s'_e}\}_{e \in \mathcal{E}}$ of all such triples provides the canonical basis of $\mathbb{P}^{3n_e-1}$. Equivalently, each of these triangles is the convex hull of the vertices $\{\mathbf{x}_{s_e} - \mathbf{y}_e + \mathbf{x}_{s'_e}, \mathbf{x}_{s_e} + \mathbf{y}_e - \mathbf{x}_{s'_e}, -\mathbf{x}_{s_e} + \mathbf{y}_e + \mathbf{x}_{s'_e}\}$. A graph $\mathcal{G}$ with $n_s$ sites and $n_e$ edges is then obtained by identifying $r = 2n_e - n_s$ sites of the collection of 2-site 1-edge graphs. This corresponds to intersecting the associated triangles in the midpoints $\mathbf{x}_{s_e}, \mathbf{x}_{s_{e'}}, \ldots$ and projecting it down to $\mathbb{P}^{3n_e-r-1} := \mathbb{P}^{n_s+n_e-1}$. The cosmological polytope $\mathcal{P}_{\mathcal{G}}$ associated to the graph $\mathcal{G}$ is thus given by the convex hull of the vertices of the triangles $\{\mathbf{x}_{s_e} - \mathbf{y}_e + \mathbf{x}_{s'_e}, \mathbf{x}_{s_e} + \mathbf{y}_e - \mathbf{x}_{s'_e}, -\mathbf{x}_{s_e} + \mathbf{y}_e + \mathbf{x}_{s'_e}\}_{e \in \mathcal{E}}$ suitably intersected – see Figure 5.

Given a cosmological polytope $\mathcal{P}_{\mathcal{G}} \subset \mathbb{P}^{n_s+n_e-1}$ and given a point $\mathcal{Y} \in \mathbb{P}^{n_s+n_e-1}$, it is equipped with a unique[10] canonical form

$$\omega(\mathcal{Y}, \mathcal{P}_{\mathcal{G}}) = \Omega(\mathcal{Y}, \mathcal{P}_{\mathcal{G}}) \langle \mathcal{Y} d^{n_s+n_e-1} \mathcal{Y} \rangle, \tag{31}$$

defined such that it only has logarithmic singularities on the boundaries of $\mathcal{P}_{\mathcal{G}}$. The coefficient $\Omega(\mathcal{Y}, \mathcal{P}_{\mathcal{G}})$, obtained from (31) by stripping off the measure on $\mathbb{P}^{n_s+n_e-1}$, is named *canonical function* and is precisely the universal wavefunction integrand $\psi_{\mathcal{G}}(x_s, y_e)$ associated to the graph $\mathcal{G}$:

$$\Omega(\mathcal{Y}, \mathcal{P}_{\mathcal{G}}) = \psi_{\mathcal{G}}(x_s, y_e), \tag{32}$$

with the labels $\{x_s, s \in \mathcal{V}\}$ and $\{y_e, e \in \mathcal{E}\}$, respectively associated to the sites and edges of $\mathcal{G}$, *i.e.* to the energies, being a system of local coordinates in $\mathbb{P}^{n_s+n_e-1}$.

The canonical function $\Omega(\mathcal{Y}, \mathcal{P}_{\mathcal{G}})$ has just simple poles, all of them along the boundary of $\mathcal{P}_{\mathcal{G}}$. Because of the equivalence (32) between $\Omega(\mathcal{Y}, \mathcal{P}_{\mathcal{G}})$ and $\psi_{\mathcal{G}}(x_s, y_e)$, the boundaries of $\mathcal{P}_{\mathcal{G}}$ capture the residues of $\psi_{\mathcal{G}}(x_s, y_e)$. Furthermore, the codimension-1 boundaries, called *facets*, are in one-to-one correspondence with the connected subgraphs of $\mathcal{G}$ [24] and are identified by the intersection $\mathcal{P}_{\mathcal{G}} \cap \mathcal{W}^{(\mathfrak{g})}$ between the cosmological polytope and the hyperplane characterised by the dual vector[11]

$$\mathcal{W}^{(\mathfrak{g})} = \sum_{s \in \mathcal{V}_{\mathfrak{g}}} \tilde{\mathbf{x}}_s + \sum_{e \in \mathcal{E}_{\mathfrak{g}}^{\text{ext}}} \tilde{\mathbf{y}}_e, \tag{33}$$

with $\mathfrak{g} \subseteq \mathcal{G}$ being a connected subgraph and $\mathcal{E}_{\mathfrak{g}}^{\text{ext}}$ the set of edges departing from $\mathfrak{g}$, while $\tilde{\mathbf{x}}_s$ and $\tilde{\mathbf{y}}_e$ are dual vectors of $\mathbf{x}_s$ and $\mathbf{y}_e$, *i.e.* they satisfy $\tilde{\mathbf{x}}_s \cdot \mathbf{x}_{s'} = \delta_{ss'}$, $\tilde{\mathbf{y}}_e \cdot \mathbf{y}'_e = \delta_{ee'}$, and $\tilde{\mathbf{x}}_s \cdot \mathbf{y}_e = \tilde{\mathbf{y}}_e \cdot \mathbf{x}_s = 0$. A point $\mathcal{Y} \in \mathbb{P}^{n_s+n_e-1}$ belongs to $\mathcal{P}_{\mathcal{G}}$ if $\mathcal{Y} \cdot \mathcal{W}^{(\mathfrak{g})} \geq 0$ for all $\mathfrak{g} \subseteq \mathcal{G}$, with the equality satisfied when it is on the boundary identified by $\mathcal{W}^{(\mathfrak{g})}$. The quantity $\mathcal{Y} \cdot \mathcal{W}^{(\mathfrak{g})}$ is just the total energy associated to the subgraph $\mathfrak{g}$

$$E_{\mathfrak{g}} = \mathcal{Y} \cdot \mathcal{W}^{(\mathfrak{g})} = \sum_{s \in \mathcal{V}_{\mathfrak{g}}} x_s + \sum_{e \in \mathcal{E}_{\mathfrak{g}}^{\text{ext}}} y_e, \tag{34}$$

and hence the boundary $\mathcal{P}_{\mathcal{G}} \cap \mathcal{W}^{(\mathfrak{g})}$ is approached as $E_{\mathfrak{g}} \longrightarrow 0$.

In order to characterise a facet $\mathcal{P}_{\mathcal{G}} \cap \mathcal{W}^{(\mathfrak{g})}$, we need to determine which vertices $\mathcal{Z}$ of $\mathcal{P}_{\mathcal{G}}$ are on it, *i.e.* which vertices satisfy the condition $\mathcal{Z} \cdot \mathcal{W}^{(\mathfrak{g})} = 0$. The one-to-one correspondence between graphs and cosmological polytopes, and between subgraphs and facets, allows to provide such a characterisation in a very simple and graphical way via the introduction of a marking that identifies those vertices which *are not* on the facet:

$$\mathcal{W} \cdot (\mathbf{x}_i - \mathbf{y}_e + \mathbf{x}'_i) > 0, \qquad \mathcal{W} \cdot (\mathbf{x}_i + \mathbf{y}_e - \mathbf{x}'_i) > 0, \qquad \mathcal{W} \cdot (-\mathbf{x}_i + \mathbf{y}_e + \mathbf{x}'_i) > 0.$$

---

[10]The canonical form is unique up to an overall constant, which can be chosen to be 1.

[11]Vectors and dual vectors respectively carry an upper and a lower index $I = 1, \ldots, n_s + n_e$, which can be suppressed for notational convenience if such a suppression does not generate any confusion, as in (33).

Hence, given a subgraph $\mathfrak{g} \subseteq \mathcal{G}$, the vertex structure of the associated facet is identified by marking all the internal edges of $\mathfrak{g}$ in the middle, as well as the edges departing from $\mathfrak{g}$ close to the sites in $\mathfrak{g}$ [24]: this marking excludes all the vertices of $\mathcal{P}_{\mathcal{G}}$ which are not on $\mathcal{P}_{\mathcal{G}} \cap \mathcal{W}^{(\mathfrak{g})}$. This association fully characterises the facets of $\mathcal{P}_{\mathcal{G}}$.

The one-to-one correspondence between markings on $\mathcal{G}$ and the vertex structure of facets $\mathcal{P}_{\mathcal{G}} \cap \mathcal{W}^{(\mathfrak{g})}$ ($\mathfrak{g} \subseteq \mathcal{G}$) allows to fully characterise the faces of $\mathcal{P}_{\mathcal{G}}$ of arbitrary codimension [27, 31, 46] in terms of compatibility conditions among the facets, *i.e.* conditions which allow to identify when $\mathcal{P}_{\mathcal{G}} \cap \mathcal{W}^{(\mathfrak{g}_1)} \cap \cdots \cap \mathcal{W}^{(\mathfrak{g}_k)}$, with $\{\mathfrak{g}_j \subseteq \mathcal{G}, \, j = 1, \ldots, k\}$, is non-empty in codimension-$k$. They also allow to fully fix the canonical function, and consequently the wavefunction of the universe, in an invariant way: the canonical function is a rational function whose denominator is identified by the facets $\{\mathcal{Y} \cdot \mathcal{W}^{(\mathfrak{g})}, \, \mathfrak{g} \subseteq \mathcal{G}\}$, while the numerator, which provides the zeroes of the $\Omega(\mathcal{Y}, \mathcal{P}_{\mathcal{G}})$, is encoded in the locus of the points where $\{\mathcal{P}_{\mathcal{G}} \cap \mathcal{W}^{(\mathfrak{g}_1)} \cap \cdots \cap \mathcal{W}^{(\mathfrak{g}_k)} = \varnothing, \, \mathfrak{g}_j \subseteq \mathcal{G}, \, \forall \, j\}$ – *i.e.* the surface identified by the intersections of the facets outside of $\mathcal{P}_{\mathcal{G}}$ in codimension-$k$[12] [56]. Alternatively, in order to compute the canonical function, it is possible to resort to *canonical form triangulations*.[13] Given a cosmological polytope $\mathcal{P}_{\mathcal{G}}$ in $\mathbb{P}^{n_s + n_e - 1}$ and a collection $\{\mathcal{P}_{\mathcal{G}}^{(j)} \subset \mathbb{P}^{n_s + n_e - 1}\}_{j=1}^{n}$ of polytopes, then $\mathcal{P}_{\mathcal{G}}$ is canonical-form triangulated by the elements $\mathcal{P}_{\mathcal{G}}^{(j)}$ of our collection if

$$\omega(\mathcal{Y}, \mathcal{P}_{\mathcal{G}}) = \sum_{j=1}^{n} \omega\left(\mathcal{Y}, \mathcal{P}_{\mathcal{G}}^{(j)}\right). \tag{35}$$

Such a notion generalises the one of *regular triangulations*: despite (35) also holds for them, they involve only the vertices of the polytope which gets triangulated. Interestingly, all the regular triangulations of a cosmological polytope $\mathcal{P}_{\mathcal{G}}$ generate spurious poles in the decomposition (35) of the canonical form as a consequence of the introduction of spurious boundaries, while all the signed triangulations through the adjoint surface return a decomposition (35) with physical poles only[14] [46].

# 4 Cosmological polytopes and the $i\epsilon$-prescription

The combinatorial-geometrical picture of the cosmological polytope provides a natural and explicit way to introduce a full class of $i\epsilon$-prescriptions. As for any projective polytope, the canonical function $\Omega(\mathcal{Y}, \mathcal{P}_{\mathcal{G}})$ has the following contour integral representation [31, 47]

$$\Omega(\mathcal{Y}, \mathcal{P}_{\mathcal{G}}) = \frac{1}{(n_e + n_s - 1)!(2\pi i)^{2n_e - n_s}} \int_{\mathbb{R}^{3n_e}} \prod_{k=1}^{n_e} \prod_{j=1}^{3} \frac{dc_k^{(j)}}{c_k^{(j)} - i\epsilon_k^{(j)}} \, \delta^{(n_e + n_s)}\left(\mathcal{Y} - \sum_{k=1}^{n_e} \sum_{j=1}^{3} c_k^{(j)} \mathcal{Z}_k^{(j)}\right), \tag{36}$$

where $\{\mathcal{Z}_k^{(j)}, \, k = 1, \ldots, n_e, \, j = 1, 2, 3\}$ is the set of vertices of $\mathcal{P}_{\mathcal{G}}$, with the index $k$ running over the edges of the associated graph $\mathcal{G}$ while $j$ labels the three vertices associated to each edge $k$. The $\delta$-function fully localises the integrand if and only if the number of vertices of the polytope is equal to the number of dimensions of the affine space in which the polytope lives, *i.e.* if the polytope is a simplex – this occurs just when $\mathcal{P}_{\mathcal{G}}$ is the triangle in $\mathbb{P}^2$ associated to the two-site line graph and for all the scattering facets $\mathcal{S}_{\mathcal{G}} := \mathcal{P}_{\mathcal{G}} \cap \mathcal{W}^{(\mathcal{G})}$ associated to any tree-level graph. In all the other cases, just a subset of the integration variables are localised:

---

[12]In the mathematics literature, the locus of the intersections of the facets outside of a polytope is called *adjoint surface* [55].

[13]This is a specific case of a more general notion of *signed triangulations*, none of which is specific to the cosmological polytopes. Rather, they are defined for any positive geometry [47].

[14]The triangulations through the adjoint surface correspond to the regular triangulations of the dual polytope $\tilde{\mathcal{P}}_{\mathcal{G}}$ obtained by mapping codimension-$k$ faces into codimension-$(n_s + n_e - k)$ ones. As the canonical function $\Omega(\mathcal{Y}, \mathcal{P}_{\mathcal{G}})$ coincides with the volume of $\tilde{\mathcal{P}}_{\mathcal{G}}$, the singularities are now encoded into the vertices of $\tilde{\mathcal{P}}_{\mathcal{G}}$.

for all the rest one must perform contour integrals, and the inequivalent contours that can be chosen correspond to all possible regular triangulations of $\mathcal{P}_{\mathcal{G}}$ through its vertices. Finally, the overall constant ensures the correct normalisation.

Note that the $i\epsilon$-prescription in (36) is needed to make the integral well-defined as, otherwise, the integrand would have poles at $\{c_k^{(j)} = 0, j = 1, 2, 3, k = 1, \ldots, n_e\}$ lying on the integration path. Importantly, the precise prescription in (36) ensures that $\mathcal{Y}$ is an (arbitrary) internal point of $\mathcal{P}_{\mathcal{G}}$ while preserving the orientation of $\mathcal{P}_{\mathcal{G}}$. As already mentioned, the $\delta$-function localises $n_e + n_s$ of the integration variables. The choice of the variables to be localised is not unique: let $\mathfrak{C} := \{c_k^{(j)}, j = 1, 2, 3, k = 1, \ldots, n_e\}$ be the set of all the integration variables, then any subset $\mathfrak{c} \subset \mathfrak{C}$ can be localised by the $(n_e + n_s)$-dimensional delta-function if and only if all the elements of $\mathfrak{c}$ are associated to vertices of $\mathcal{P}_{\mathcal{G}}$ that are linearly independent among each other and, thus, form a basis of the affine space $\mathbb{R}^{n_s+n_e}$. Let $\hat{\mathfrak{c}}$ be the chosen subset, then the integral (36) acquires the form

$$\Omega(\mathcal{Y}, \mathcal{P}_{\mathcal{G}}) = \frac{1}{(n_e + n_s - 1)!(2\pi i)^{2n_e - n_s}} \int_{\mathbb{R}^{2n_e - n_s}} \prod_{c_b \in \mathfrak{C} \backslash \hat{\mathfrak{c}}} \frac{dc_b}{c_b - i\epsilon_b} \prod_{\hat{c} \in \hat{\mathfrak{c}}} \frac{1}{\hat{c}(c_b) - i\hat{\epsilon}} |\mathcal{J}|^{-1}, \qquad (37)$$

with $\mathcal{J}$ the Jacobian obtained from the $\delta$-function, which is given by the contraction of the vertices of $\mathcal{P}_{\mathcal{G}}$ associated to the elements of $\hat{\mathfrak{c}}$ via the Levi-Civita tensor in $\mathbb{R}^{n_s+n_e}$

$$\mathcal{J} = \langle a_1 \ldots a_{n_s+n_e} \rangle := \epsilon_{I_1 \ldots I_{n_s+n_e}} \mathcal{Z}_{(a_1)}^{I_1} \cdots \mathcal{Z}_{(a_{n_s+n_e})}^{I_{n_s+n_e}}, \qquad (38)$$

where the $I$'s are $\mathbb{R}^{n_s+n_e}$ indices, while each label $(a)$ is a short-hand notation for the pair of indices $k, j$ associated to the elements of $\hat{\mathfrak{c}}$. Each $\hat{c}(c_b) \in \hat{\mathfrak{c}}$ is now a linear inhomogeneous polynomial in the unfixed variables $c_b \in \mathfrak{C} \backslash \hat{\mathfrak{c}}$. The remaining integrations can be viewed as contour integrations and can be performed one at the time. Let $c_{\hat{b}}$ be the variable that we chose to integrate out first. As the coefficients of the polynomials $\hat{c}(c_{\hat{b}})$ can be either positive or negative, when we look at the location of the poles in the complex plane of $c_{\hat{b}}$, some of the poles lie in the upper-half plane (UHP) and others in the lower-half one (LHP). Such integration can therefore be performed by equivalently closing the integration contour in the UHP or in the LHP, each choice providing a different representation for the integrated function

$$\begin{aligned}
\Omega(\mathcal{Y}, \mathcal{P}_{\mathcal{G}}) &\sim \int_{\mathbb{R}^{2n_e - n_e - 1}} \prod_{c_b \in \mathfrak{C} \backslash (\hat{\mathfrak{c}} \cup \{c_{\hat{b}}\}} \frac{dc_b}{c_b - i\epsilon_b} |\mathcal{J}|^{-1} \sum_{\hat{c}_{\hat{b}} \in \mathrm{UHP}} \mathrm{Res}_{c_{\hat{b}} = \hat{c}_{\hat{b}}} \left\{ \prod_{\hat{c} \in \hat{\mathfrak{c}}} \frac{1}{\hat{c}(c_b) - i\hat{\epsilon}_b} \right\} \\
&= -\int_{\mathbb{R}^{2n_e - n_e - 1}} \prod_{c_b \in \mathfrak{C} \backslash (\hat{\mathfrak{c}} \cup \{c_{\hat{b}}\})} \frac{dc_b}{c_b - i\epsilon_b} |\mathcal{J}|^{-1} \sum_{\hat{c}_{\hat{b}} \in \mathrm{LHP}} \mathrm{Res}_{c_{\hat{b}} = \hat{c}_{\hat{b}}} \left\{ \prod_{\hat{c} \in \hat{\mathfrak{c}}} \frac{1}{\hat{c}(c_b) - i\hat{\epsilon}_b} \right\},
\end{aligned} \qquad (39)$$

where $\sim$ indicates the omission of an overall constant factor which is irrelevant to our discussion. This integration provides two inequivalent polytope subdivisions of $\mathcal{P}_{\mathcal{G}}$, depending on whether the integration is performed in the UHP or LHP,

$$\mathcal{P}_{\mathcal{G}} = \bigcup_{a \in \mathfrak{u}} \mathcal{P}^{(a)} = \bigcup_{a \in \mathfrak{l}} \mathcal{P}^{(a)}, \qquad (40)$$

where $\mathfrak{u}$ and $\mathfrak{l}$ are the sets of polytopes forming the subdivision associated to the poles in the UHP and LHP, respectively. If we perform all the integrations, all the inequivalent integration paths will return the regular triangulations of $\mathcal{P}_{\mathcal{G}}$. However, in order to understand the role of the $i\epsilon$-prescription in (36), we can focus our discussion on the single integration we already performed and, consequently, just on the integrand of (39),

$$\sum_{\hat{c}_{\hat{b}} \in \mathrm{UHP}} \mathrm{Res}_{c_{\hat{b}} = \hat{c}_{\hat{b}}} \left\{ \prod_{\hat{c} \in \hat{\mathfrak{c}}} \frac{1}{\hat{c}(c_{\hat{b}}) - i\hat{\epsilon}_{\hat{b}}} \right\} = -\sum_{\hat{c}_{\hat{b}} \in \mathrm{LHP}} \mathrm{Res}_{c_{\hat{b}} = \hat{c}_{\hat{b}}} \left\{ \prod_{\hat{c} \in \hat{\mathfrak{c}}} \frac{1}{\hat{c}(c_{\hat{b}}) - i\hat{\epsilon}_{\hat{b}}} \right\}. \qquad (41)$$

First, all $\hat{c}_{\hat{b}}$, representing the location of the poles in the $c_{\hat{b}}$-plane, have a term $\pm i\hat{\epsilon}_{\hat{b}}$, whose sign depends on whether the contour of integration is closed in the upper half plane. Secondly, the denominators in (41) now show a linear combination of different $\epsilon$'s which can either have the same sign or different sign. In the former case, the location of the pole in the complex plane of any of the other integration variables is fixed, while in the latter in principle a hierarchy among the $\epsilon$'s should be arbitrarily chosen in order to determine their precise location. It turns out that such poles correspond to the facets of the elements of the subdivision which are shared among them and are not facets of $\mathcal{P}_\mathcal{G}$, *i.e.* they are spurious poles.

Let us now change the $i\epsilon$-prescription in (36) such that we have poles of both types: $c_k^{(j)} - i\epsilon_k^{(j)}$ and $c_k^{(j)} + i\epsilon_k^{(j)}$. When we analyse the location of the poles in the $\tilde{c}_k^{(j)}$-plane, some of them will be shifted from the UHP to the LHP and vice-versa, returning a different result than (39) and hence no longer computing the canonical function of $\mathcal{P}_\mathcal{G}$. The change of the location of some of the poles is equivalent to changing the orientation of the corresponding elements of the subdivision (40), returning a polytope $\mathcal{P}' \subset \mathcal{P}_\mathcal{G}$. Said differently, changing the location of some of the poles is equivalent to moving some of the vertices inside the convex hull of the others. Finally, choosing the $i\epsilon$-prescription such that $c_k^{(j)} + i\epsilon_k^{(j)}$ is equivalent to swapping *all* the poles from one half-plane to the other changing the orientation of the full polytope at each integration.

It is useful to illustrate this phenomenon with a toy, but nevertheless explicit, example.

**An illustrative example.** Let us consider a square $\mathcal{P} \subset \mathbb{P}^2$ given by the convex hull of the vertices $\{\mathcal{Z}_{(j)} \in \mathbb{P}^2, j = 1, 2, 3, 4\}$ or equivalently by the inequalities

$$\langle \mathcal{Y}12 \rangle > 0, \quad \langle \mathcal{Y}23 \rangle > 0, \quad \langle \mathcal{Y}34 \rangle > 0, \quad \langle \mathcal{Y}41 \rangle > 0,$$
$$\langle 123 \rangle > 0, \quad \langle 234 \rangle > 0, \quad \langle 341 \rangle > 0, \quad \langle 412 \rangle > 0. \tag{42}$$

Each inequality $\langle ijk \rangle > 0$ indicates that the point $\mathcal{Z}_{(i)}$ lies in the positive half-plane identified by the line $\mathcal{W}_I^{(jk)} := \epsilon_{IJK} \mathcal{Z}_{(j)}^J \mathcal{Z}_{(k)}^K$. Consequently, the four conditions in the first line ensure that an arbitrary point $\mathcal{Y}$ of the square $\mathcal{P}$ lies in the region of $\mathbb{P}^2$ defined by the four positive half-spaces determined by the lines $\mathcal{W}^{(i,i+1)}$ ($i = 1, \ldots, 4$), while the four conditions in the second line guarantee that no vertex $\{\mathcal{Z}_{(i)}, i = 1, \ldots, 4\}$ is inside the triangle identified by the other three.

The contour integral representation (36) for this square is then given by

$$\Omega(\mathcal{Y}, \mathcal{P}) \sim \frac{1}{2\pi i} \int_{\mathbb{R}^4} \prod_{j=1}^{4} \frac{dc_j}{c_j - i\epsilon_j} \, \delta^{(3)}\left(\mathcal{Y} - \sum_{j=1}^{4} c_j \mathcal{Z}_{(j)}\right). \tag{43}$$

We can choose the vertices $\{\mathcal{Z}^{(j)}, j = 1, 2, 3\}$ as the basis for the affine space $\mathbb{R}^3$ and fix the coefficients $\{c_j, j = 1, 2, 3\}$ as functions of $c_4$ via the $\delta$-function in (43):

$$\mathcal{Y}^I = \sum_{j=1}^{4} c_j \mathcal{Z}_{(j)}^I \quad \Longrightarrow \quad \begin{cases} \langle \mathcal{Y}23 \rangle = \langle 123 \rangle c_1 + \langle 234 \rangle c_4, \\ \langle \mathcal{Y}31 \rangle = \langle 123 \rangle c_2 - \langle 341 \rangle c_4, \\ \langle \mathcal{Y}12 \rangle = \langle 123 \rangle c_3 + \langle 412 \rangle c_4, \end{cases} \tag{44}$$

where the coefficients $\langle \cdots \rangle$ have been arranged to be all positive. Hence

$$\Omega(\mathcal{Y}, \mathcal{P}_\mathcal{G}) \sim \frac{1}{2\pi i} \int_{\mathbb{R}} \frac{dc_4}{c_4 - i\epsilon_4} \frac{1/\langle 123 \rangle}{\left[\frac{\langle \mathcal{Y}23 \rangle}{\langle 123 \rangle} - \frac{\langle 234 \rangle}{\langle 123 \rangle} c_4 - i\epsilon_1\right]\left[\frac{\langle \mathcal{Y}31 \rangle}{\langle 123 \rangle} + \frac{\langle 341 \rangle}{\langle 123 \rangle} c_4 - i\epsilon_2\right]\left[\frac{\langle \mathcal{Y}12 \rangle}{\langle 123 \rangle} - \frac{\langle 412 \rangle}{\langle 123 \rangle} c_4 - i\epsilon_3\right]}. \tag{45}$$

In the $c_4$-plane, two poles turn out to be in the UHP, $c_4 = i\epsilon_4$ and $c_2(c_4) = i\epsilon_2$, while the other two lie in the LHP – see Figure 6. Let $\mathcal{I}$ be the integrand, then the contour integration yields

$$
\begin{aligned}
\Omega(\mathcal{Y}, \mathcal{P}) &\sim \operatorname{Res}_{c_4=i\epsilon_4}\mathcal{I} + \operatorname{Res}_{c_2(c_4)=i\epsilon_2}\mathcal{I} = -\operatorname{Res}_{c_1(c_4)=i\epsilon_1}\mathcal{I} - \operatorname{Res}_{c_3(c_4)=i\epsilon_3}\mathcal{I} \\
&= \Omega(\mathcal{Y}, \mathcal{P}^{(123)}) + \Omega(\mathcal{Y}, \mathcal{P}^{(341)}) = \Omega(\mathcal{Y}, \mathcal{P}^{(234)}) + \Omega(\mathcal{Y}, \mathcal{P}^{(412)}),
\end{aligned}
\tag{46}
$$

where the two sides of the equality in the first line represent the result of the UHP- and LHP-integration respectively, which correspond to the two possible regular triangulations of the square $\mathcal{P}$, $\mathcal{P}^{(123)} \cup \mathcal{P}^{(341)}$ and $\mathcal{P}^{(234)} \cup \mathcal{P}^{(412)}$, as emphasised in the second line and pictorially showed here:

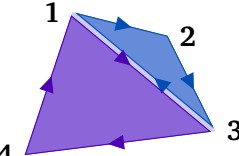
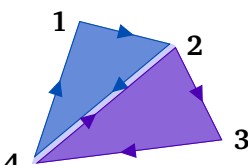

The arrows in the picture above represent the orientation of $\mathcal{P}$ and of the triangulating simplices, which ought to agree with each other. As mentioned above, the boundaries of the simplices which are not boundaries of $\mathcal{P}$, *i.e.* the segments (31) and (24), have different orientations and they turn out to receive an ambiguous $i\epsilon$ deformation. Explicitly:

$$
\begin{aligned}
\Omega(\mathcal{Y}, \mathcal{P}_{\mathcal{G}}) &\sim \frac{1}{\langle\mathcal{Y}31\rangle - i(\langle123\rangle\epsilon_2 - \langle341\rangle\epsilon_4)} \left[ \frac{\langle123\rangle^2}{[\langle\mathcal{Y}12\rangle - i(\langle412\rangle\epsilon_4 + \langle123\rangle\epsilon_3)][\langle\mathcal{Y}23\rangle - i(\langle234\rangle\epsilon_4 + \langle123\rangle\epsilon_1)]} \right.\\
&\quad \left. - \frac{\langle341\rangle^2}{[\langle\mathcal{Y}34\rangle - i(\langle234\rangle\epsilon_2 + \langle341\rangle\epsilon_1)][\langle\mathcal{Y}41\rangle - i(\langle412\rangle\epsilon_2 + \langle341\rangle\epsilon_3)]} \right] \\
&= \frac{1}{\langle\mathcal{Y}24\rangle - i(\langle234\rangle\epsilon_3 - \langle412\rangle\epsilon_1)} \left[ \frac{\langle412\rangle^2}{[\langle\mathcal{Y}41\rangle - i(\langle341\rangle\epsilon_3 + \langle412\rangle\epsilon_2)][\langle\mathcal{Y}12\rangle - i(\langle123\rangle\epsilon_3 + \langle412\rangle\epsilon_4)]} \right.\\
&\quad \left. - \frac{\langle234\rangle^2}{[\langle\mathcal{Y}23\rangle - i(\langle123\rangle\epsilon_1 + \langle234\rangle\epsilon_4)][\langle\mathcal{Y}34\rangle - i(\langle341\rangle\epsilon_1 + \langle234\rangle\epsilon_2)]} \right],
\end{aligned}
\tag{47}
$$

where the first two lines in the r.h.s. represent the integration in the UHP and hence the triangulation $\mathcal{P} = \mathcal{P}^{(123)} \cup \mathcal{P}^{(341)}$, while the last two are the result of the integration in the LHP and provide the triangulation $\mathcal{P} = \mathcal{P}^{(412)} \cup \mathcal{P}^{(234)}$. However, the ambiguity in the sign of $i\epsilon$ in the overall term in both triangulations is the manifestation of the fact that this common boundary has opposite orientation in the two simplices and it is resolved by the fact that its residue is zero: considering the relation $(x \mp i\epsilon)^{-1} = \text{P.V.}\{x^{-1}\} \pm i\pi\delta(x)$, the contribution from the $\delta$-function is identically zero.

Let us now go back to the contour integral (43) and change the $i\epsilon$-prescription for one of the $c_j$'s, namely $c_2$, which now appears as $c_2 + i\epsilon_2$. The poles are now splitted in such a way that one, corresponding to the solution $c_4 = i\epsilon_4$ is on the UHP, while the other three lie in the LHP. Integrating over the real axis and closing the contour in the UHP and in the LHP we obtain

$$
\operatorname{Res}_{c_4=i\epsilon_4}\mathcal{I} = -\operatorname{Res}_{c_2(c_4)=i\epsilon_2}\mathcal{I} - \operatorname{Res}_{c_1(c_4)=i\epsilon_1}\mathcal{I} - \operatorname{Res}_{c_3(c_4)=i\epsilon_3}\mathcal{I}.
\tag{48}
$$

With this second prescription, the integral in the UHP returns simply the canonical function for the triangle $\mathcal{P}^{(123)}$, while the contour in the LHP returns a triangulation of $\mathcal{P}^{(123)}$ through the point $\mathcal{Z}_{(4)}$:

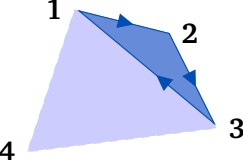
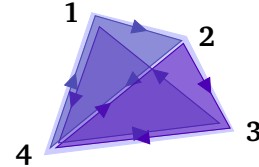

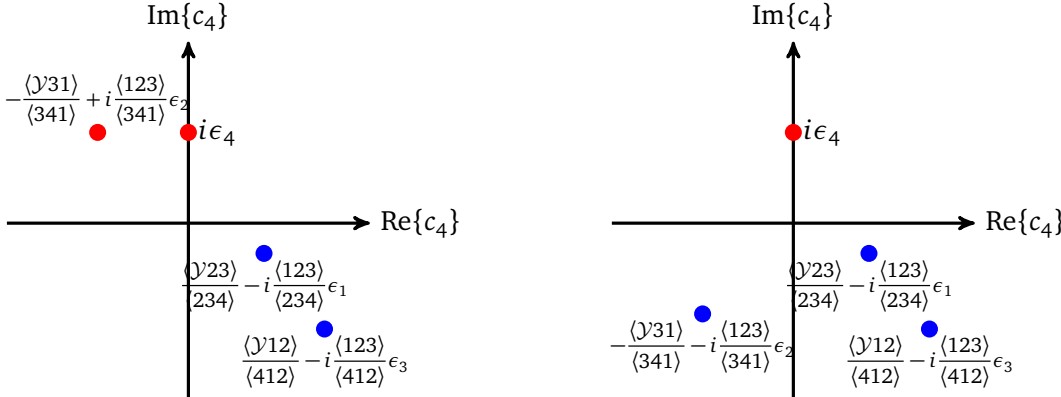

Figure 6: Poles location in the $c_4$-plane for the contour integral representation of the canonical function of a square in $\mathbb{P}^2$. *On the left*. The prescription $\{c_j \longrightarrow c_j - i\epsilon_j, j = 1,\ldots,4\}$ moves a subset of the poles to the UHP and the complementary subset into the LHP. The integrations in the UHP and LHP provide different representation of the canonical function corresponding to the two different regular triangulations of the square. *On the right*. The prescription $\{c_j \longrightarrow c_j - i\epsilon_j, j = 3, 4, 1\}$ and $c_2 \longrightarrow c_2 + i\epsilon_2$ moves 1 pole to the UHP and three into the LHP. Closing the contour of the integration in the UHP or LHP provide two different representation for the canonical form of the triangle $\mathcal{P}^{(123)}$ by changing the orientation of some boundaries. This is equivalent to considering the vertex $\mathcal{Z}_{(4)}$ as inside the triangle $\mathcal{P}^{(123)}$.

This is equivalent to keeping the prescription $c_2 - i\epsilon_2$ in the original integral and taking $\langle 431 \rangle > 0$, *i.e.* the vertex $\mathcal{Z}_{(4)}$ lies now in the positive half-plane identified by the line $\mathcal{W}_I^{(31)} = \epsilon_{IJK} \mathcal{Z}_{(3)}^J \mathcal{Z}_{(1)}^K$. As $\langle 412 \rangle$ and $\langle 423 \rangle$ are kept positive, the vertex $\mathcal{Z}_{(4)}$ is also in the half-planes identified by the lines $\mathcal{W}_I^{(12)}$ and $\mathcal{W}_I^{(23)}$, so $\mathcal{Z}_{(4)}$ lies *inside* the triangle $\mathcal{P}^{(123)}$:

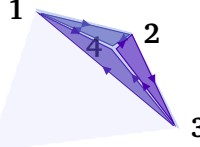

## 4.1 From the geometry to kinematic space

The $i\epsilon$-prescription in the context of cosmological polytopes arises from the requirement of having a well-defined integral representation, as otherwise the latter would have poles lying along the integration contour. This is similar to what happens in the standard definition of the wavefunction of the universe in terms of a time integration, where all the singularities arise from the infinitely oscillating behaviour of the integrand as the early time region is approached.

In the time integral picture, the regularisation associated to a deformation of the Hamiltonian operator occurs by taking the external energies to be complex and requiring that their imaginary part is negative. This can be implemented, compatibly with unitarity, via $x_j \longrightarrow x_j - i\epsilon_j$ ($\epsilon_j > 0$). With such a deformation, the wavefunction shows singularities of the form $\sum_{s \in \mathcal{V}_\mathfrak{g}} (x_s - i\epsilon_s) + \sum_{e \in \mathcal{E}_\mathfrak{g}^{\text{ext}}} y_e$.

Something similar happens when we look at the contour integral representation (36) for the canonical function of a cosmological polytope. The integration produces two classes of poles: one class corresponds to facets of the simplices $\mathcal{P}^{(j)}$'s which are also facets of the cos-

mological polytope $\mathcal{P}_{\mathcal{G}}$, while the second one identifies facets of $\mathcal{P}^{(j)}$'s that are *internal* to $\mathcal{P}_{\mathcal{G}}$ and hence are spurious.

Let us consider the contour integral (36) and fix the set $\mathfrak{c}$ of $n_s + n_e$ integration variables via the $\delta$-function so that it acquires the form (37). The solution for each element of $\mathfrak{c}$ is just given by the projection of the space $\tilde{\mathcal{Y}} := \mathcal{Y} - \sum_{c_b \in \bar{\mathfrak{c}}} c_b \mathcal{Z}_{(b)}$ onto the hyperplane $\mathcal{W}_I^{(a_1 \cdots a_{n_s + n_e - 1})} := \epsilon_{I J_1 \cdots J_{n_s + n_e - 1}} \mathcal{Z}_{(a_1)}^{J_1} \cdots \mathcal{Z}_{(a_{n_s + n_e - 1})}^{J_{n_s + n_e - 1}}$ [15] defined by all the vertices associated to $\mathfrak{c}$ but the one associated to the variable which we are fixing:

$$
\begin{aligned}
0 &= \langle \tilde{\mathcal{Y}} \, a_1 \dots a_{n_s + n_e - 1} \rangle - c_{\hat{a}} \langle \hat{a} \, a_1 \dots a_{n_s + n_e - 1} \rangle \\
&= \langle \mathcal{Y} \, a_1 \dots a_{n_s + n_e - 1} \rangle - \sum_{c_b \in \bar{\mathfrak{c}}} c_b \langle b \, a_1 \dots a_{n_s + n_e - 1} \rangle - c_{\hat{a}} \langle \hat{a} \, a_1 \dots a_{n_s + n_e - 1} \rangle \,,
\end{aligned} \tag{49}
$$

$\forall \, \hat{a} \in \mathfrak{c}$, $\{a_j \in \mathfrak{c} \setminus \{\hat{a}\}, \, j = 1, \dots, n + n_e - 1\}$ and with $\mathfrak{c} \cup \bar{\mathfrak{c}} = \mathfrak{C} = \{c_k^{(j)}, \, j = 1, 2, 3, \, k = 1, \dots, n_e\}$. Let us consider now the integration over one of the remaining variables, namely $c_{\hat{b}}$:

$$
\Omega(\mathcal{Y}, \mathcal{P}_{\mathcal{G}}) \sim \int_{\mathbb{R}^{\tilde{n}-1}} \prod_{c_b \in \bar{\mathfrak{c}} \setminus \{c_{\hat{b}}\}} \frac{dc_b}{c_b - i\epsilon_b} \int_{\mathbb{R}} \frac{dc_{\hat{b}}}{c_{\hat{b}} - i\epsilon_{\hat{b}}} \prod_{\substack{\hat{a}, a_j \in a_{\mathfrak{c}} \\ a_j \neq \hat{a}}} \frac{1}{\frac{\langle \mathcal{Y}' a_1 \dots a_{n_s + n_e - 1} \rangle}{\langle \hat{a} a_1 \dots a_{n_s + n_e - 1} \rangle} - c_{\hat{b}} \frac{\langle \hat{b} a_1 \dots a_{n_s + n_e - 1} \rangle}{\langle \hat{a} a_1 \dots a_{n_s + n_e - 1} \rangle} - i\epsilon_{\hat{a}}} \,, \tag{50}
$$

where $\tilde{n} = 2n_e - n_s$ and $\mathcal{Y}' := \mathcal{Y} - \sum_{c_b \in \bar{\mathfrak{c}} \setminus \{c_{\hat{b}}\}} c_b \mathcal{Z}_{(\hat{b})}$. Whether the poles in the $c_{\hat{b}}$-plane lie in the UHP or LHP depends on whether the vertices $\mathcal{Z}_{(\hat{b})}$ and $\mathcal{Z}_{(\hat{a})}$ lie in the same half-space identified by the hyperplane $\mathcal{W}^{(a_1 \cdots a_{n_s + n_e - 1})}$ or not: in the first case, the relative sign is positive and the sign of $c_{\hat{b}}$ stays unchanged, while in the second one, the relative sign is negative and the sign in front of $c_{\hat{b}}$ changes. If $\mathcal{W}^{(a_1 \cdots a_{n_s + n_e - 1})}$ is such that $\mathcal{P}_{\mathcal{G}} \cap \mathcal{W}_{(a_1 \cdots a_{n_s + n_e - 1})} \neq \varnothing$ and coincides with a facet, then both $\mathcal{Z}_{(\hat{b})}$ and $\mathcal{Z}_{(\hat{a})}$ lie on the positive half-space [16] and the associated pole is in the LHP. If $\mathcal{W}^{(a_1 \cdots a_{n_s + n_e - 1})}$ is such that $\mathcal{P}_{\mathcal{G}} \cap \mathcal{W}^{(a_1 \cdots a_{n_s + n_e - 1})} \neq \varnothing$ and intersects $\mathcal{P}_{\mathcal{G}}$ also in its interior, then some of the vertices will lie in its positive half-space and some others in the negative one. In this case $\mathcal{P}_{\mathcal{G}} \cap \mathcal{W}^{(a_1 \cdots a_{n_s + n_e - 1})}$ represents a spurious boundary and it can give rise to a change in the sign of the coefficient of $c_{\hat{b}}$ placing the associated pole in the UHP – see Figure 7.

The integration contour in the UHP then picks poles corresponding to hyperplanes containing spurious boundaries, together with $c_{\hat{b}} = i\epsilon_{\hat{b}}$. Closing the integration path in the UHP, the spurious poles of the canonical form arise from the denominators giving rise to the poles in the UHP computed at the location of any other pole in the UHP. Notice that such poles depend on $\alpha \epsilon_{\hat{b}} - \beta \epsilon_{\hat{a}}$ with $\alpha, \beta$ being positive constants, which seem to signal an ambiguity for the other integration – it could lie in the UHP or LHP depending on whether $\alpha \epsilon_{\hat{b}} - \beta \epsilon_{\hat{a}}$ is positive or negative – or even for the final result if no further integration has to be carried out. However, precisely because these poles are associated to hyperplanes which contain spurious boundaries, the residues of the canonical form with respect to them are zero no matter the ambiguity in the associated $i\epsilon$. The other poles after the $c_{\hat{b}}$ integration have all the following structure

$$
\frac{1}{\frac{\langle \mathcal{Y}' a_1 \dots a_{n_s + n_e - 1} \rangle}{\langle \hat{a} a_1 \dots a_{n_s + n_e - 1} \rangle} - \hat{c}_{\hat{b}} \frac{\langle \hat{b} a_1 \dots a_{n_s + n_e - 1} \rangle}{\langle \hat{a} a_1 \dots a_{n_s + n_e - 1} \rangle} - i(\epsilon_{\hat{a}} + \alpha_{\hat{a}'} \epsilon_{\hat{a}'})} \,, \tag{51}
$$

where $\hat{c}_{\hat{b}}$ is computed at the location of the pole $c_{\hat{a}'}(c_{\hat{b}}) = i\epsilon_{\hat{a}'}$, without the $i\epsilon$-part which is made explicit as $i\alpha_{\hat{a}'} \epsilon_{\hat{a}'}$, $\alpha_{\hat{a}'}$ being a positive constant. In this case the sign of the $i\epsilon$ is unambiguous. If there are further integrations to be performed, one can iterate the very same

---

[15] The notation $\mathcal{Z}_{(a)}$ is a shorthand for $\mathcal{Z}_k^{(j)}$.

[16] While it is never the case that $\mathcal{Z}_{\hat{a}}$ lies on the hyperplane $\mathcal{W}^{(a_1 \cdots a_{n_s + n_e - 1})}$ as such a vertex, together with the ones defining $\mathcal{W}^{(a_1 \cdots a_{n_s + n_e - 1})}$ are chosen to span $\mathbb{R}^{n_s + n_e}$, it can occur that $\mathcal{Z}_{\hat{b}}$ does. However in this case $\langle \hat{b} \, a_1 \dots a_{n_s + n_e - 1} \rangle = 0$ and the denominator where this occurs does not show any pole in the complex $c_{\hat{b}}$-plane.

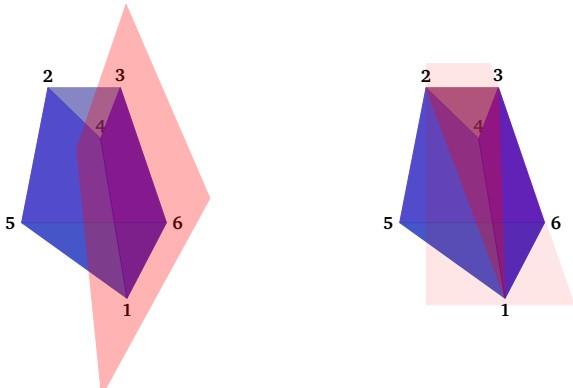

Figure 7: Contour integration, $\mathcal{P}_{\mathcal{G}}$ vertex structure and $i\epsilon$-prescription. The picture represents the cosmological polytope associated with the 2-site 1-loop graph, whose contour integral representation has six integration variables $\mathfrak{C} := \{c_j, j = 1, \ldots, 6\}$ and has support on $\delta^{(4)}(\sum_{j=1}^{6} c_j \mathcal{Z}_{(j)})$. Let us fix $\mathfrak{c} := \{c_j, j = 1, \ldots 4\}$, and analyse the pole structure in the variables $\bar{\mathfrak{c}} = \{c_5, c_6\}$. *On the left.* One pole is determined by a hyperplane containing one of the facets, *e.g.* $\mathcal{W}^{(143)}$. Then all the vertices which are not on it satisfy $\langle j143 \rangle > 0$, *e.g.* $\{\mathcal{Z}_{(j)}, j = 2, 5\}$: the coefficient $\langle 5413 \rangle / \langle 2413 \rangle$ in front of the $c_5$ keeps the same sign as $i\epsilon_2$ and the pole at $\hat{c}_2(c_5) - i\epsilon_2 = 0$ lies in the LHP. *On the right.* Another class of poles is associated to a hyperplane intersecting the polytope in its interior, namely $\mathcal{W}^{(123)}$. Then, $\{\mathcal{Z}_{(j)}, j = 5, 6\}$ lie on the positive half-space and $\mathcal{Z}_{(4)}$ in the negative one: the coefficient $\langle 5123 \rangle / \langle 4123 \rangle$ changes sign with respect to $i\epsilon_4$ and the pole $\hat{c}_4(c_5) - i\epsilon_4 = 0$ lies in the UHP.

analysis until there is no other integration to be carried out. The final result will be a sum of terms with two classes of poles: one spurious, characterised by an ambiguous sign for the $i\epsilon$, and the other of the form (51), but now dependent on the actual point $\mathcal{Y} \in \mathbb{P}^{n_s + n_e - 1}$ rather than $\mathcal{Y}' = \mathcal{Y}'(\mathcal{Y}, \bar{\mathfrak{c}})$

$$\frac{1}{\dfrac{\langle \mathcal{Y} a_1 \ldots a_{n_s + n_e - 1} \rangle}{\langle \hat{a} a_1 \ldots a_{n_s + n_e - 1} \rangle} - i \left( \epsilon_{\hat{b}} + \sum_{\hat{a}'} \alpha_{\hat{a}'} \epsilon_{\hat{a}'} \right)}, \tag{52}$$

where $\alpha_{\hat{a}'}$ are all positive coefficients and $\mathcal{W}^{(a_1 \ldots a_{n_s + n_e - 1})}$ is a hyperplane containing one of the facets of $\mathcal{P}_{\mathcal{G}}$. As the first term is positive, in the local coordinates of the projective space associated to the weights of the graph (52) acquires the explicit form

$$\frac{1}{\dfrac{\langle \mathcal{Y} a_1 \ldots a_{n_s + n_e - 1} \rangle}{\langle \hat{a} a_1 \ldots a_{n_s + n_e - 1} \rangle} - i \left( \epsilon_{\hat{b}} + \sum_{\hat{a}'} \alpha_{\hat{a}'} \epsilon_{\hat{a}} \right)} = \frac{1}{\beta_{\mathfrak{g}} \left( \sum_{s \in \mathcal{V}_{\mathfrak{g}}} x_s + \sum_{e \in \mathcal{E}_{\mathfrak{g}}^{\text{ext}}} y_e \right) - i \left( \epsilon_{\hat{b}} + \sum_{\hat{a}'} \alpha_{\hat{a}'} \epsilon_{\hat{a}} \right)}, \tag{53}$$

$\mathfrak{g} \subseteq \mathcal{G}$ being the subgraph associated to the hyperplane $\mathcal{W}^{(a_1 \ldots a_{n_s + n_e - 1})}$, while $\beta_{\mathfrak{g}}$ is a positive constant.

From a kinematic space viewpoint, it is straightforward to see that the $i\epsilon$-prescription in the canonical function induced by its contour integral representation is equivalent to giving a negative imaginary part to *all* energies

$$x_s \longrightarrow x_s - i\epsilon_{x_s}, \qquad y_e \longrightarrow y_e - i\epsilon_{y_e}, \tag{54}$$

$\forall s \in \mathcal{V}, e \in \mathcal{E}$, with the $\{\epsilon_{x_s}, s \in \mathcal{V}\}$ and $\{\epsilon_{y_e}, e \in \mathcal{E}\}$ a linear combination of those in the prescription of (53).

While the $i\epsilon$-prescription for the external energies at each vertex is associated to the convergence of the time integral definition of the evolution operator, the $i\epsilon$-prescription on the internal energies can be understood as associated to the distributional nature of the bulk-to-bulk propagator

$$
\begin{aligned}
G(y_e, \eta_{s_e}, \eta_{s_{e'}}) &= \int\limits_{-\infty}^{+\infty} \frac{d\omega}{2\pi i} \frac{e^{i\omega(\eta_{s_e}-\eta_{s_{e'}})} - e^{i\omega(\eta_{s_e}+\eta_{s_{e'}})}}{\omega^2 - y_e^2 + i\tilde{\epsilon}} \\
&= \frac{1}{2(y_e - i\epsilon_{y_e})} \Big[ e^{-i(y_e - i\epsilon_{y_e})(\eta_{s_e}-\eta_{s_{e'}})}\vartheta(\eta_{s_e}-\eta_{s_{e'}}) + e^{+i(y_e - i\epsilon_{y_e})(\eta_{s_e}-\eta_{s_{e'}})}\vartheta(\eta_{s_{e'}}-\eta_{s_e}) \\
&\qquad - e^{+i(y_e - i\epsilon_{y_e})(\eta_{s_e}+\eta_{s_{e'}})} \Big] ,
\end{aligned}
\tag{55}
$$

where $\tilde{\epsilon}$ and $\epsilon_{y_e}$ are related to each other via a rescaling.

Hence, the $i\epsilon$-prescription induced by the contour integral representation of the canonical function and determined by the positivity requirement on the geometry of the cosmological polytope, not only contains the prescription that guarantees the well-definiteness of the evolution operator compatibly with unitarity, but also gives rise to a prescription *a la Feynman* for the propagators. Finally, note that the deformations (54) implemented in the time-integral representation make the time integral converge for $x \in \mathbb{R}$ rather than $x \in \mathbb{R}_+$, as it would be the case using the (non-unitary) deformation of the contour integral around $-\infty$.[17]

In the next sections we will see how the effects of this $i\epsilon$-prescription are encoded in the geometry of the cosmological polytope without having to resort to the contour integral representation (36). We will also explore its consequences providing a geometrical formulation of the cosmological optical theorem and the cutting rules for the perturbative wavefunction as well as a complete discussion of the emergence of the flat-space optical theorem and the associated Cutkosky rules.

## 5 The geometry of perturbative unitarity: The optical polytope

Cosmological polytopes provide an invariant definition of the wavefunction coefficients $\psi_{\mathcal{G}}$ associated to any graph $\mathcal{G}$ in which the wavefunction of the universe can be organised perturbatively: as discussed in Section 3, they have their own first principles definition independent of any physical assumption, they are in one-to-one correspondence with graphs, and they are endowed with a unique canonical form which encodes the wavefunction which can be directly determined through their face structure via the compatibility conditions in [46].

The cosmological optical theorem provides a representation for the wavefunction as a sum of terms which show folded singularities, which are spurious. As the wavefunction is given by the canonical function and the different representations of the former correspond to the different polytope subdivisions of the latter, the cosmological optical theorem has to be associated to a specific polytope subdivision $\mathcal{P}_{\mathcal{G}} = \bigcup_{j=1}^{n} \mathcal{P}^{(j)}$, such that the facets of $\{\mathcal{P}^{(j)}, j = 1, \ldots, n\}$ which are not facets of $\mathcal{P}_{\mathcal{G}}$ are associated to folded singularities and one of the $\mathcal{P}^{(j)}$ encodes $\psi_{\mathcal{G}}^{\dagger}(-|\vec{p}_k|, \hat{p}_k \cdot \hat{p}_l)$.

Equivalently, as the statement of the optical theorem for the perturbative wavefunction can be schematically written as

$$
\Delta\psi_{\mathcal{G}} := \psi_{\mathcal{G}}(|\vec{p}_k|, \hat{p}_k \cdot \hat{p}_l) + \psi_{\mathcal{G}}^{\dagger}(-|\vec{p}_k|, \hat{p}_k \cdot \hat{p}_l) = -\sum_{\cancel{e} \in \mathcal{E}} \psi_{\cancel{e}} ,
\tag{56}
$$

---

[17]Said differently, the time integrals can be made convergent as $\eta \longrightarrow -\infty$ by complexifying the external energies $x_s$ with the condition that its imaginary part is negative.

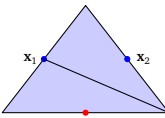 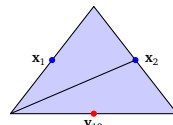 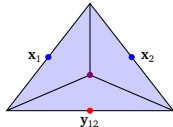 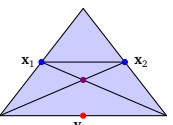 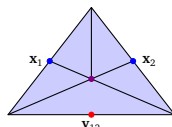

Figure 8: Triangulations of the triangle via special points. The first two triangulations use one of the midpoints of the triangle intersectable edges: they correspond to the recursion relations obtained from the frequency representation. The third triangulation, which makes use of the centroid only, as well as the last two, which instead make use of all the special points, are related to different cuts.

we can look for a geometry directly encoding $\Delta\psi_{\mathcal{G}}$ and whose two sides are just different polytope subdivisions.

Finally, let $\widehat{\mathcal{W}}^{(\mathfrak{g})}$ be the hyperplane associated to the subgraph $\mathfrak{g}$ such that

$$\mathcal{Y} \cdot \widehat{\mathcal{W}}^{(\mathfrak{g})} = \sum_{s \in \mathcal{V}_{\mathfrak{g}}} x_s - \sum_{e \in \mathcal{E}_{\mathfrak{g}}^{\text{ext}}} y_e \,, \tag{57}$$

then $\mathcal{P}_{\mathcal{G}} \cap \widehat{\mathcal{W}}^{(\mathfrak{g})} \neq \varnothing$ and the hyperplane $\widehat{\mathcal{W}}^{(\mathfrak{g})}$ intersects $\mathcal{P}_{\mathcal{G}}$ inside: it intersect the facet $\mathcal{P}_{\mathcal{G}} \cap \mathcal{W}^{(\mathfrak{g})}$ in all the vertices $\{x_s + y_e - x_{s'}, \ s \in \mathcal{V}_{\mathfrak{g}}, \ e \in \mathcal{E}_{\mathfrak{g}}^{\text{ext}}, \ s' \in \mathcal{V}_{\bar{\mathfrak{g}}}\}$, while it intersects the other facets in the $\{x_s, \ s \notin \mathcal{V}_{\mathfrak{g}}\}$ and $\{y_e, \ e \notin \mathcal{E}_{\mathfrak{g}}^{\text{ext}}\}$. This implies that some of the vertices of $\mathcal{P}_{\mathcal{G}}$ lie on the positive half-space identified by $\widehat{\mathcal{W}}^{(\mathfrak{g})}$, while others on the negative half-space. As these hyperplanes intersect a cosmological polytope $\mathcal{P}_{\mathcal{G}}$ in its interior, they allow for a polytope subdivision of $\mathcal{P}_{\mathcal{G}}$ via internal points. Consequently, $\Delta\psi_{\mathcal{G}}$ is described by a *non-convex polytope*.

## 5.1 Unitarity and non-convexity

Let us begin with considering the projective space $\mathbb{P}^2$ with local coordinates $\mathcal{Y} = (x_1, y_{12}, x_2)$ and the cosmological polytope defined as the convex-hull of the vertices

$$\{\mathbf{x}_1 - \mathbf{y}_{12} + \mathbf{x}_2, \ \mathbf{x}_1 + \mathbf{y}_{12} - \mathbf{x}_2, \ -\mathbf{x}_1 + \mathbf{y}_{12} + \mathbf{x}_2\}\,.$$

Let us ask the following question: is there any sense in which we can triangulate such a triangle? Being a simplex, there is just one regular triangulation which corresponds to the triangle itself. Resorting to the notion of canonical-form triangulation, is there any way in which we can canonically select them? At the end of the day we could define the collection of polytopes which canonical-form triangulate our triangle by introducing arbitrary vertices. However, there are special points we can resort to and we can use to identify specific triangulations. Such points are the midpoints $\{\mathbf{x}_1, \mathbf{x}_2\}$ of its intersectable sides, as well as its centroid $\mathbf{x}_1 + \mathbf{y}_{12} + \mathbf{x}_2$.

The two triangulations through one of the midpoints $\mathbf{x}_i$ introduces a spurious boundary given by the segment $\{\mathbf{x}_i, -\mathbf{x}_i + \mathbf{y}_{12} + \mathbf{x}_j\}$, which is identified by the line $\mathcal{Y} \cdot \widehat{\mathcal{W}}^{(\mathfrak{g}_j)} \sim x_j - y_{12} = 0$, $\mathfrak{g}_j$ being the subgraph containing only the vertex labelled by $x_j$, and reproduces the recursion relation derived from the frequency representation [24, 26]. They make manifest the isomorphism among all the facets of the triangle, *i.e.* all its facets are given by the same type of polytope and are mapped into each other via combinatorial automorphisms, the latter being the combinatorial manifestation of the Bunch-Davies condition [26].

Here we will be focusing on those triangulations involving either the centroid $\mathbf{x}_{12} := \mathbf{x}_1 + \mathbf{y}_{12} + \mathbf{x}_2$ only or all the special points. While the former is unique, there are various triangulations through both the centroid as well as the other two special points $\{\mathbf{x}_1, \mathbf{x}_2\}$

Figure 9: Transitioning from a convex to a non-convex geometry. The first picture represents the quadrilateral $\mathcal{Q}_{\mathcal{G}}(a)$, with the red line being the locus of zeroes of the associated canonical form. For $\mathcal{Q}_{\mathcal{G}}(a)$ is defined for $a \in ]0,1[$, which guarantees convexity. For $a = 0$, $\mathcal{Q}_{\mathcal{G}}(0)$ degenerates into a triangle, while for $a = -1$ $\mathcal{Q}_{\mathcal{G}}(-1) = \mathcal{O}_{\mathcal{G}}$ and the locus of the zeroes of the canonical form coincides with the line passing through the midpoints $\mathbf{x}_1$ and $\mathbf{x}_2$.

– see Figure 8. All such triangulations of $\mathcal{P}_{\mathcal{G}}$ have in common one element of the collections of the triangles triangulating $\mathcal{P}_{\mathcal{G}}$: it is given by the triangle identified as convex hull of $\{\mathbf{x}_1 + \mathbf{y}_{12} + \mathbf{x}_2, \mathbf{x}_1 + \mathbf{y}_{12} - \mathbf{x}_2, -\mathbf{x}_1 + \mathbf{y}_{12} + \mathbf{x}_2\}$. Notice that its boundaries are identified by the scattering facet as well as the two medians passing through the special points $\{\mathbf{x}_1, \mathbf{x}_2\}$, *i.e.* $\mathcal{W}^{(\mathcal{G})} = \tilde{\mathbf{x}}_1 + \tilde{\mathbf{x}}_2$, $\widehat{\mathcal{W}}^{(\mathfrak{g}_1)} = \tilde{\mathbf{x}}_1 - \tilde{\mathbf{y}}_{12}$ and $\widehat{\mathcal{W}}^{(\mathfrak{g}_2)} = \tilde{\mathbf{x}}_2 - \tilde{\mathbf{y}}_{12}$, and its canonical function is given by

$$
\begin{aligned}
\Omega(\mathcal{Y}, \mathcal{P}_{\mathcal{G}}') &= \frac{\langle \mathbf{x}_1 \mathbf{x}_2 \mathbf{x}_{12} \rangle^2}{(\mathcal{Y} \cdot \mathcal{W}^{(\mathcal{G})})(\mathcal{Y} \cdot \widehat{\mathcal{W}}^{(\mathfrak{g}_1)})(\mathcal{Y} \cdot \widehat{\mathcal{W}}^{(\mathfrak{g}_2)})} \\
&= \frac{1}{(x_1 + x_2)(x_1 - y_{12})(-y_{12} + x_2)} := -\psi_{\mathcal{G}}^{\dagger}(-x_1, y_{12}, -x_2),
\end{aligned}
\tag{58}
$$

which is precisely the function appearing in the definition of $\Delta\psi_{\mathcal{G}}$. Consequently, the *non-convex* quadrilateral $\mathcal{O}_{\mathcal{G}}$ identified by the vertices:

$$
\{\mathbf{x}_1 - \mathbf{y}_{12} + \mathbf{x}_2, \mathbf{x}_1 + \mathbf{y}_{12} - \mathbf{x}_2, \mathbf{x}_1 + \mathbf{y}_{12} + \mathbf{x}_2, -\mathbf{x}_1 + \mathbf{y}_{12} + \mathbf{x}_2\}
$$

provides an invariant formulation for $\Delta\psi_{\mathcal{G}}$, and the triangulations of such an object would be expected to provide all the possible cutting rules. This is the simplest example of *optical polytope*. Before discussing the triangulations of the optical polytope $\mathcal{O}_{\mathcal{G}}$ and their relation to cuts, it is important to make some general considerations. For any polytope in $\mathbb{P}^{n_s + n_e - 1}$, its canonical function is a rational function with the denominator fixed by its facets, while the numerator is a polynomial of degree $\delta = F - n_s - n_e$, $F$ being the number of its facets, which provides the locus of the intersections of the facets *outside* the polytope itself [56]. In the non-convex case, pairs of facets intersect *inside* one of them without generating a boundary of higher codimension: the locus identified by such points fixes the numerator of the canonical function – we will further discuss this point in the next subsection. For the quadrilateral $\mathcal{O}_{\mathcal{G}}$, such points are precisely the midpoints $\{\mathbf{x}_1, \mathbf{x}_2\}$, which define the line $\mathcal{W}^{(0)} = -2\tilde{\mathbf{y}}_{12}$. Said differently, the canonical function $\Omega(\mathcal{Y}, \mathcal{O}_{\mathcal{G}})$ has to satisfy the following codimension-2 constraints

$$
\text{Res}_{\mathcal{W}^{(\mathfrak{g}_j)}} \text{Res}_{\widehat{\mathcal{W}}^{(\mathfrak{g}_j)}} \Omega(\mathcal{Y}, \mathcal{O}_{\mathcal{G}}) = 0, \qquad \forall j = 1, 2,
\tag{59}
$$

which precisely fix $\mathcal{W}^{(0)}$ to be

$$
\mathcal{W}_I^{(0)} = \epsilon_{IJK} Z_{(A)}^J Z_{(B)}^K, \quad \text{with} \quad
\begin{cases}
Z_{(A)}^I = \epsilon^{IJK} \mathcal{W}_J^{(\mathfrak{g}_1)} \widehat{\mathcal{W}}_K^{(\mathfrak{g}_1)}, \\
Z_{(B)}^I = \epsilon^{IJK} \mathcal{W}_J^{(\mathfrak{g}_2)} \widehat{\mathcal{W}}_K^{(\mathfrak{g}_2)}.
\end{cases}
\tag{60}
$$

Hence, the canonical function $\Omega(\mathcal{Y}, \mathcal{O}_{\mathcal{G}})$ of $\mathcal{O}_{\mathcal{G}}$ can be directly written as

$$
\begin{aligned}
\Omega(\mathcal{Y}, \mathcal{O}_{\mathcal{G}}) &= \frac{\langle \mathcal{Y}AB \rangle}{(\mathcal{Y} \cdot \mathcal{W}^{(\mathfrak{g}_1)})(\mathcal{Y} \cdot \mathcal{W}^{(\mathfrak{g}_2)})(\mathcal{Y} \cdot \widehat{\mathcal{W}}^{(\mathfrak{g}_1)})(\mathcal{Y} \cdot \widehat{\mathcal{W}}^{(\mathfrak{g}_2)})} \\
&= \frac{-2y_{12}}{(x_1 - y_{12})(x_1 + y_{12})(y_{12} + x_2)(-y_{12} + x_2)}.
\end{aligned}
\tag{61}
$$

It is important to remark that while in general a non-convex polytope is defined as a specific union of convex ones, *i.e.* through a specific triangulation, the knowledge of the facets, which are identified by the hyperplanes $\{\mathcal{W}^{(\mathfrak{g}_j)}, \widehat{\mathcal{W}}^{(\mathfrak{g}_j)}, j = 1, 2\}$, together with the compatibility conditions on the facets given by the constraints (59), provide an invariant definition and characterisation for the non-convex quadrilateral $\mathcal{O}_{\mathcal{G}}$.

Alternatively, $\mathcal{O}_{\mathcal{G}}$ can be also defined as the deformation of the convex square $\mathcal{Q}_{\mathcal{G}}(a)$ defined as the convex hull of the vertices

$$
\{\mathbf{x}_1 - \mathbf{y}_{12} + \mathbf{x}_2, \mathbf{x}_1 + \mathbf{y}_{12} - \mathbf{x}_2, -a\mathbf{x}_1 + (2 + a)\mathbf{y}_{12} - a\mathbf{x}_2, -\mathbf{x}_1 + \mathbf{y}_{12} + \mathbf{x}_2\},
$$

where $a$ is an arbitrary *positive* parameter. Then, the non-convex square is recovered in the limit $a \longrightarrow -1$. Importantly, the compatibility conditions (59) are deformed accordingly: for $a > 0$, the vertex which is inside the triangle defined by $\{\mathbf{x}_1 - \mathbf{y}_{12} + \mathbf{x}_2, \mathbf{x}_1 + \mathbf{y}_{12} - \mathbf{x}_2, -\mathbf{x}_1 + \mathbf{y}_{12} + \mathbf{x}_2\}$ is moved outside, accordingly with the two facets defining it, and the adjoint surface, which in the non-convex square intersects the square inside in the midpoints $\{\mathbf{x}_1, \mathbf{x}_2\}$, is also moved outside – see Figure 9.

In the next section we will see how the non-convex polytope associated to an arbitrary graph can be generally defined in an invariant way via both compatibility conditions and as a deformation of a convex polytope. This formulation has the virtue of being independent of triangulations and, hence, it allows to ask general questions on the polytope subdivisions, that, as we will show, are associated to the cosmological optical theorem and make manifest the relation between the latter and the optical theorem in flat-space.

Before closing this subsection, let us observe that another characterisation of the non-convex polytope $\mathcal{O}_{\mathcal{G}}$ can be given in terms of inequalities. Given the four hyperplanes $\{\mathcal{W}^{(\mathfrak{g}_j)}, \widehat{\mathcal{W}}^{(\mathfrak{g}_j)}, j = 1, 2\}$, the non-convex polytope $\mathcal{O}_{\mathcal{G}}$ is identified by the union of the following sets of inequalities

$$
\begin{aligned}
&\left\{\mathcal{Y} \cdot \mathcal{W}^{(\mathfrak{g}_1)} > 0, \quad \mathcal{Y} \cdot \mathcal{W}^{(\mathfrak{g}_2)} > 0, \quad \mathcal{Y} \cdot \widehat{\mathcal{W}}^{(\mathfrak{g}_1)} > 0, \quad \mathcal{Y} \cdot \widehat{\mathcal{W}}^{(\mathfrak{g}_2)} > 0\right\}, \\
&\left\{\mathcal{Y} \cdot \mathcal{W}^{(\mathfrak{g}_1)} > 0, \quad \mathcal{Y} \cdot \mathcal{W}^{(\mathfrak{g}_2)} > 0, \quad \mathcal{Y} \cdot \widehat{\mathcal{W}}^{(\mathfrak{g}_1)} < 0, \quad \mathcal{Y} \cdot \widehat{\mathcal{W}}^{(\mathfrak{g}_2)} > 0\right\}, \\
&\left\{\mathcal{Y} \cdot \mathcal{W}^{(\mathfrak{g}_1)} > 0, \quad \mathcal{Y} \cdot \mathcal{W}^{(\mathfrak{g}_2)} > 0, \quad \mathcal{Y} \cdot \widehat{\mathcal{W}}^{(\mathfrak{g}_1)} > 0, \quad \mathcal{Y} \cdot \widehat{\mathcal{W}}^{(\mathfrak{g}_2)} < 0\right\},
\end{aligned}
\tag{62}
$$

*i.e.* considering both the region determined by the overlap of four positive half-planes and the two regions obtained by alternatively considering the negative half-plane identified by the two lines $\{\widehat{\mathcal{W}}^{(\mathfrak{g}_j)}, j = 1, 2\}$ passing through $\{\mathbf{x}_1 + \mathbf{y}_{12} + \mathbf{x}_2\}$. However, such a definition constitutes a triangulation of $\mathcal{O}_{\mathcal{G}}$ into the quadrilateral $\{\mathbf{x}_1 - \mathbf{y}_{12} - \mathbf{x}_2, \mathbf{x}_1, \mathbf{x}_1 + \mathbf{y}_{12} + \mathbf{x}_2, \mathbf{x}_2\}$, which is identified by the first set of inequalities in (62), and the two triangles $\{\mathbf{x}_1, \mathbf{x}_1 + \mathbf{y}_{12} - \mathbf{x}_2, \mathbf{x}_1 + \mathbf{y}_{12} + \mathbf{x}_2\}$ and $\mathbf{x}_1 + \mathbf{y}_{12} + \mathbf{x}_2, -\mathbf{x}_1 + \mathbf{y}_{12} + \mathbf{x}_2, \mathbf{x}_2$ respectively identified instead by the second and third sets of inequalities in (62).

## 5.2 An invariant definition for the optical polytope

Let us consider a graph $\mathcal{G}$ with weights on both its sites, $\{x_s, s \in \mathcal{V}\}$, and its edges, $\{y_e, e \in \mathcal{E}\}$. Then the *optical polytope* $\mathcal{O}_{\mathcal{G}}$ associated to the graph $\mathcal{G}$ is a polytope living in $\mathbb{P}^{n_s + n_e - 1}$ with a

local patch given by the weights of $\mathcal{G}$, and defined as the non-convex limit of the convex hull $\mathcal{Q}_{\mathcal{G}}(a)$ of the following set of $4n_e$ vertices

$$
\begin{aligned}
\mathcal{O}_{\mathcal{G}} &:= \lim_{a \longrightarrow -1} \mathcal{Q}_{\mathcal{G}}(a) \\
&= \lim_{a \longrightarrow -1} \begin{array}{c} \text{convex} \\ \text{hull} \end{array} \left\{ \mathbf{x}_{s_e} - \mathbf{y}_e + \mathbf{x}_{s'_e}, \, \mathbf{x}_{s_e} + \mathbf{y}_e - \mathbf{x}_{s'_e}, \, -\mathbf{x}_{s_e} + \mathbf{y}_e + \mathbf{x}_{s'_e}, \right. \\
&\qquad\qquad\qquad \left. -a\mathbf{x}_{s_e} + (2+a)\mathbf{y}_e + \sum_{e' \in \mathcal{E} \backslash \{e\}} 2(1+a)\mathbf{y}_{e'} - a\mathbf{x}_{s'_e} \right\}_{e \in \mathcal{E}},
\end{aligned}
\tag{63}
$$

with $\mathcal{Q}(a)$ defined for $a > 0$. The canonical form of $\mathcal{O}_{\mathcal{G}}$ is then obtained as

$$
\omega(\mathcal{Y}, \mathcal{O}_{\mathcal{G}}) = \lim_{a \longrightarrow -1} \omega(\mathcal{Y}, \mathcal{Q}_{\mathcal{G}}(a)).
\tag{64}
$$

Indeed the limit and convex-hull operations do not commute: for each edge, the vertex parametrised by $a$ lies in the interior of the triangle defined by the other three vertices as $a$ becomes negative, and the convex hull of all the vertices is just the cosmological polytope $\mathcal{P}_{\mathcal{G}}$ itself.

We can therefore analyse the combinatorial and geometric structure of $\mathcal{Q}_{\mathcal{G}}(a)$ using all our knowledge on convex polytopes and finally take the limit $a \longrightarrow -1$ in order to extract information about $\mathcal{O}_{\mathcal{G}}$.

**Graphs and optical polytopes.** As for the cosmological polytope $\mathcal{P}_{\mathcal{G}}$, a correspondence between graphs and the optical polytope can be established. As we showed in the previous section, it is possible to associate a square in $\mathbb{P}^2$ with vertices

$$
\{\mathbf{x}_{s_e} - \mathbf{y}_e + \mathbf{x}_{s'_e}, \, \mathbf{x}_{s_e} + \mathbf{y}_e - \mathbf{x}_{s'_e}, \, -\mathbf{x}_{s_e} + \mathbf{y}_e + \mathbf{x}_{s'_e}, \, -a\mathbf{x}_{s_e} + (2+a)\mathbf{y}_e - a\mathbf{x}_{s'_e}\},
$$

($a > 0$) to a two-site line graph. Let us now consider a collection of $n_e$ two-site line graphs and the corresponding squares:

$$
\left\{\mathbf{x}_{s_e} - \mathbf{y}_e + \mathbf{x}_{s'_e}, \, \mathbf{x}_{s_e} + \mathbf{y}_e - \mathbf{x}_{s'_e}, \, -\mathbf{x}_{s_e} + \mathbf{y}_e + \mathbf{x}_{s'_e}, \, -a\mathbf{x}_{s_e} + (2+a)\mathbf{y}_e - a\mathbf{x}_{s'_e}\right\}_{e \in \mathcal{E}}.
$$

They can be embedded in the same space $\mathbb{P}^{n_e(2+n_e)-1}$ with local coordinates $\mathcal{Y} := (\{x_{s_e}, y_e, y_e^{(2)}, \ldots, y_e^{(n_e)} x_{s'_e}\}_{e \in \mathcal{E}})$, given by the weights of the two-line graphs, by mapping them in disconnected tetrahedrons, each of which living in a subspace $\mathbb{P}^3 \subset \mathbb{P}^{2+n_e-1} \subset \mathbb{P}^{n_e(2+n_e)-1}$:

$$
\left\{\mathbf{x}_{s_e} - \mathbf{y}_e + \mathbf{x}_{s'_e}, \, \mathbf{x}_{s_e} + \mathbf{y}_e - \mathbf{x}_{s'_e}, \, -\mathbf{x}_{s_e} + \mathbf{y}_e + \mathbf{x}_{s'_e}, \, -a\mathbf{x}_{s_e} + (2+a)\mathbf{y}_e + 2(1+a)\sum_{j=2}^{n_e} \mathbf{y}_e^{(j)} - a\mathbf{x}_{s'_e}\right\}_{e \in \mathcal{E}}.
$$

For future reference, let us indicate these four vertices for fixed $e \in \mathcal{E}$ as $\{\mathcal{Z}_e^{(j)}\}_{j=1}^4$ following the ordering of appearance in the list above.

Note that, given any tetrahedron from this collection, *i.e.* for fixed $e \in \mathcal{E}$, the triangular facet identified by the vertices $\{\mathcal{Z}_e^{(j)}\}_{j=1}^3$ identifies the cosmological polytope associated to one of the two-site graphs from the collection. As a generic connected graph $\mathcal{G}$ with $n_s$ sites and $n_e$ edges can be obtained from the collection of the $n_e$ two-site line graphs by suitably identifying $r_1 := 2n_e - n_s$ sites, the corresponding polytope $\mathcal{Q}_{\mathcal{G}}(a)$ is obtained by first projecting the tetrahedrons down to a codimension $r_2 := n_e(n_e - 1)$ space via the identification

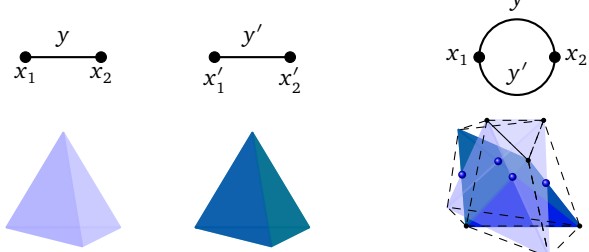

Figure 10: Graphs and $\mathcal{Q}_{\mathcal{G}}(a)$. Each graph $\mathcal{G}$ can be seen as a collection of 2-site line graphs merged in a subset of their sites. As each two-site line graph can be put in correspondence to a square and these squares can be mapped to tetrahedra once they are embedded into the same space, a general graph $\mathcal{G}$ can be associated to a polytope defined as a suitable intersection of such tetrahedra. Here we show a collection of two two-site line graphs and the associated tetrahedra as well as the two-site one-loop graph obtained by merging their sites in pairs and the associated polytope $\mathcal{Q}_{\mathcal{G}}(a)$. The dashed lines show the convex hull obtained by intersecting the two tetrahedra, with the intersection points marked, while the coloured areas represent the defining tetrahedra themselves.

$\{y_e^{(j)} = y_{e_j}, e_j \in \mathcal{E} \setminus \{e\}, j = 2, \ldots, n_e\}$ and then intersecting them in the relevant midpoints $\mathbf{x}_{s_e}, \mathbf{x}_{s_{e'}}, \ldots$ of these triangular facets. The projection maps the vertices to

$$
\left\{ \mathbf{x}_{s_e} - \mathbf{y}_e + \mathbf{x}_{s_{e'}}, \mathbf{x}_{s_e} + \mathbf{y}_e - \mathbf{x}_{s_{e'}}, -\mathbf{x}_{s_e} + \mathbf{y}_e + \mathbf{x}_{s_{e'}}, -a\mathbf{x}_{s_e} + (2+a)\mathbf{y}_e + 2(1+a) \sum_{e' \in \mathcal{E} \setminus \{e\}} \mathbf{y}_{e'} - a\mathbf{x}_{s_{e'}} \right\}_{e \in \mathcal{E}},
$$

imposing the constraints

$$
\mathcal{Z}_e^{(4)} + a\mathcal{Z}_e^{(1)} \sim \mathcal{Z}_e^{(2)} + \mathcal{Z}_e^{(3)} + (1+a) \sum_{e' \in \mathcal{E} \setminus \{e\}} \left( \mathcal{Z}_e^{(2)} + \mathcal{Z}_e^{(3)} \right), \qquad \forall\, e \in \mathcal{E}, \tag{65}
$$

while the intersection imposes pairs of constraints on the vertices of the intersected tetrahedra – for example, let us imagine to identify the midpoints $\mathbf{x}_{s_e} = \mathbf{x}_{s_{e'}}$, then

$$
\begin{aligned}
\mathcal{Z}_e^{(1)} + \mathcal{Z}_e^{(2)} &\sim \mathcal{Z}_{e'}^{(1)} + \mathcal{Z}_{e'}^{(2)}, \\
\mathcal{Z}_e^{(4)} + a\mathcal{Z}_e^{(3)} &\sim \mathcal{Z}_{e'}^{(4)} + a\mathcal{Z}_{e'}^{(3)}.
\end{aligned} \tag{66}
$$

With such identifications, the polytope $\mathcal{Q}_{\mathcal{G}}(a)$ associated to a graph $\mathcal{G}$ lives in $\mathbb{P}^{n_e(2+n_e)-r_1-r_2-1} = \mathbb{P}^{n_s+n_e-1}$ and is given by the convex hull of the vertices as in (63) – see Figure 10.

**Facet structure of the optical polytope.** As we already saw for the cosmological polytope, and generally holds for any convex polytope, a facet is given by the hyperplane identified by a covector $\mathcal{W}$ such that $\mathcal{Q}_{\mathcal{G}}(a) \cap \mathcal{W} \neq \varnothing$ and $\mathcal{Z}_e^{(j)} \cdot \mathcal{W} \geq 0 \; \forall\, e \in \mathcal{E}, \forall\, j = 1, \ldots, 4$, with the equality satisfied if and only if the vertex $\mathcal{Z}_e^{(j)}$ is on the hyperplane. Consider a generic hyperplane expanded in the basis of covectors $\{\tilde{\mathbf{x}}_s, \tilde{\mathbf{y}}_e, s \in \mathcal{V}, e \in \mathcal{E}\}$,

$$
\mathcal{W} = \sum_{s \in \mathcal{V}} \tilde{x}_s \tilde{\mathbf{x}}_s + \sum_{e \in \mathcal{E}} \tilde{y}_e \tilde{\mathbf{y}}_e, \tag{67}
$$

where $\{\tilde{x}_s, \tilde{y}_e, s \in \mathcal{V}, e \in \mathcal{E}\}$ are arbitrary coefficients. Then, the conditions $\mathcal{Z}_e^{(j)} \cdot \mathcal{W} \geq 0 \; \forall\, e \in \mathcal{E}, \, \forall\, j = 1, \dots, 4$ can be rewritten as

$$
\begin{aligned}
\alpha_{(e,e)} &:= \mathcal{Z}_e^{(1)} \cdot \mathcal{W} = \tilde{x}_{s_e} - \tilde{y}_e + \tilde{x}_{s_e'} \geq 0 \,, \\
\alpha_{(e,s_e)} &:= \mathcal{Z}_e^{(2)} \cdot \mathcal{W} = \tilde{x}_{s_e} + \tilde{y}_e - \tilde{x}_{s_e'} \geq 0 \,, \\
\alpha_{(e,s_e')} &:= \mathcal{Z}_e^{(3)} \cdot \mathcal{W} = -\tilde{x}_{s_e} + \tilde{y}_e + \tilde{x}_{s_e'} \geq 0 \,, \\
\widehat{\alpha}_{(e,e)} &:= \mathcal{Z}_e^{(4)} \cdot \mathcal{W} = -a\tilde{x}_{s_e} + (2+a)\tilde{y}_e + 2(1+a)\sum_{e' \in \mathcal{E}\backslash\{e\}} \tilde{y}_{e'} - a\tilde{x}_{s_e'} \geq 0 \,,
\end{aligned}
\tag{68}
$$

with the $\alpha$'s satisfying the very same linear relations (66) as the vertices

$$
\begin{aligned}
\alpha_{(e,e)} + \alpha_{(e,s_e)} &= \alpha_{(e',e')} + \alpha_{(e',s_{e'})} \,, \\
\widehat{\alpha}_{(e,e)} + a \cdot \alpha_{(e,s_e')} &= \widehat{\alpha}_{(e',e')} + a \cdot \alpha_{(e',s_{e'}'')} \,,
\end{aligned}
\tag{69}
$$

Hence, a vertex of $\mathcal{Q}_{\mathcal{G}}(a)$ is on a facet if the associated $\alpha$ is zero, and the hyperplane containing a facet is identified by the non-trivial maximal set of vanishing $\alpha$'s compatible with the conditions (69), where by non-trivial we mean that not all the $\alpha$'s should be set to zero. It is convenient to introduce a marking on the graph for identifying the positive $\alpha$'s

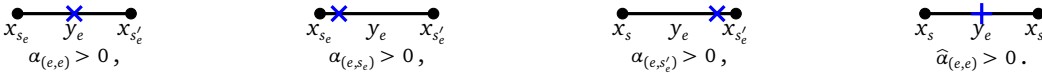

Importantly, as the relations (69) are sums of non-negative quantities in both sides of the equations, if both $\alpha$'s are zero on one side then they ought to be zero also on the other side. This implies that there a facet cannot be identified by a hyperplane such that just one of the sides of any of the equations (69) has all the $\alpha$'s set to zero. Consequently, the following markings are not allowed

$$
\tag{70}
$$

These are obtained by setting to zero all the $\alpha$'s in the second and first equation in (69) respectively, *i.e.* each line in (70) is implied by just one of the linear relation (69). In particular, note that the first three conditions in (68) and the first line in (69) are the very same satisfied by the cosmological polytope $\mathcal{P}_{\mathcal{G}}$ associated to $\mathcal{G}$: the polytope $\mathcal{Q}_{\mathcal{G}}(a)$ contains $\mathcal{P}_{\mathcal{G}}$ and displays (some of) its facets.

A general question now is whether there are further restrictions on setting the $\alpha$'s to zero, and hence further forbidden markings. From the set of constraints in (65), it can be noticed that each $\widehat{\alpha}$ satisfies also the following relation

$$
\widehat{\alpha}_{(e,e)} + a\alpha_{(e,e)} = \alpha_{(e,s_e)} + \alpha_{(e,s_e')} + (1+a) \sum_{\hat{e} \in \mathcal{E}\backslash\{e\}} (\alpha_{(\hat{e},s_{\hat{e}})} + \alpha_{(\hat{e},s_{\hat{e}}')}) \,, \qquad \forall\, e \in \mathcal{E} \,.
\tag{71}
$$

If we set both $\alpha_{(e,e)}$ and $\widehat{\alpha}_{(e,e)}$ to zero for a fixed $e \in \mathcal{E}$, then (71) forces to set all the $\alpha$'s on its right-hand-side to zero, given that it is just a sum of non-negative terms. However, all the relations (71) involve the sum of the same non-negative terms with different positive coefficients. Thus, the validity of (71) for any $e \in \mathcal{E}$ implies that also $\alpha_{(e,e)} = 0$ and $\widehat{\alpha}_{(e,e)} = 0$ for any $e \in \mathcal{E}$. Requiring that two vertices of the $\mathcal{Z}_e^{(1)}$ and $\mathcal{Z}_e^{(4)}$ are on the same hyperplane automatically implies that all the vertices of $\mathcal{Q}_{\mathcal{G}}(a)$ ought to be on the same hyperplane, so that the only solution on the constraints on the $\alpha$'s is the trivial solution: only the whole space can contain such vertices, and its intersection with $\mathcal{Q}_{\mathcal{G}}(a)$ is the whole polytope. This

implies that at least one of the markings ●──✕──● and ●──✚──● should always appear on every edge. Notice also that setting all the $\alpha$'s to zero on the right-hand-side of (71), while keeping $\alpha_{(e,e)} > 0 \, \forall \, e \in \mathcal{E}$ identifies the scattering facet of the cosmological polytope $\mathcal{P}_{\mathcal{G}}$. However, as just emphasised, such a solution is not compatible with the constraints on $Q_{\mathcal{G}}(a)$ and, consequently, the scattering facet of $\mathcal{P}_{\mathcal{G}}$ *is not* a facet of $\mathcal{Q}_{\mathcal{G}}(a)$.[18]

Hence, also the following markings are not allowed

$$
\tag{72}
$$

as they show at least one of the edges unmarked in the middle. Let us now consider the hyperplanes containing the two vertices $\mathcal{Z}_e^{(1)}$ and $\mathcal{Z}_{e'}^{(4)}$, $e$ and $e'$ being two edges with a common site. Hence, $\alpha_{(e,e)} = 0$ and $\widehat{\alpha}_{(e',e')} = 0$. The linear relations (69) then become

$$
\begin{aligned}
\alpha_{(e,s_e)} &= \alpha_{(e',e')} + \alpha_{(e',s_{e'})}, \\
\widehat{\alpha}_{(e,e)} + a \cdot \alpha_{(e,s'_e)} &= a \cdot \alpha_{(e',s'_e)}.
\end{aligned}
\tag{73}
$$

The previous analysis showed that requiring both $\alpha_{(e,e)}$ and $\widehat{\alpha}_{(e,e)}$, associated to the same edge $e$, to vanish, implies the trivial solution only. Consequently, as we are considering $\alpha_{(e,e)} = 0$ and $\widehat{\alpha}_{(e',e')} = 0$, $\widehat{\alpha}_{(e,e)}$ and $\alpha_{(e',e')}$ ought to be positive, and hence, we can just set $\alpha_{(e,s_e)}$ and $\alpha_{(e',s_{e'})}$ to zero in order to have (73) satisfied. The relations (71) then write

$$
\begin{aligned}
0 &= a\alpha_{(e',e')} + \widehat{\alpha}_{(e,e)} + \sum_{\hat{e} \in \mathcal{E} \setminus \{e,e'\}} \left( \alpha_{(\hat{e},s_{\hat{e}})} + \alpha_{(\hat{e},s'_{\hat{e}})} \right), \\
0 &= \alpha_{(e',e')} + a\widehat{\alpha}_{(e,e)} + \sum_{\hat{e} \in \mathcal{E} \setminus \{e,e'\}} \left( \alpha_{(\hat{e},s_{\hat{e}})} + \alpha_{(\hat{e},s'_{\hat{e}})} \right).
\end{aligned}
\tag{74}
$$

As their right-hand-sides are just sums of non-negative terms, they can be satisfied if and only if each term is individually zero: the only solution compatible with the linear constraints (73) is the trivial solution. Thus, there is no facet of $\mathcal{Q}_{\mathcal{G}}(a)$ which can contain the vertices $\mathcal{Z}_e^{(1)}$ and $\mathcal{Z}_{e'}^{(4)}$, and hence the following marking is also not allowed

$$
\tag{75}
$$

Now, given an arbitrary graph $\mathcal{G}$, a facet of $\mathcal{Q}_{\mathcal{G}}(a)$ is identified by a hyperplane $\mathcal{W}$ containing as many vertices of $\mathcal{Q}_{\mathcal{G}}(a)$ as possible (without containing the full polytope). In terms of the markings, this corresponds to mark $\mathcal{G}$ in such a way that no configuration (70), (72) and (75) is present, and that removing any of the markings either forces to remove all of them or yields one of the non-allowed configurations.

This can be translated into the following graphical rule. Given an arbitrary graph $\mathcal{G}$, we can associate *a pair* of facets $(\mathcal{W}^{(\mathfrak{g})}, \widehat{\mathcal{W}}_a^{(\mathfrak{g})})$ to each subgraph $\mathfrak{g} \subset \mathcal{G}$.[19] The vertex structure of $\mathcal{Q}_{\mathcal{G}} \cap \mathcal{W}^{(\mathfrak{g})}$ is obtained by marking the cut edges close to the sites in $\mathfrak{g}$, the edges in $\mathfrak{g}$ in the middle with both ✕ and ✚ , and all the edges outside of $\mathfrak{g}$ with ✚ . The vertex structure of

---

[18]This statement can be checked directly by considering the hyperplane $\mathcal{W}^{(\mathcal{G})} = \sum_{s \in \mathcal{G}} \tilde{\mathbf{x}}_s$, which identifies the scattering facet and is compatible with the relation in the first line of (66). From (68), the $\alpha$'s satisfy the following relations

$$
\alpha_{(e,e)} > 0, \quad \alpha_{(e,s_e)} = 0, \quad \alpha_{(s,s'_e)} = 0, \quad \widehat{\alpha}_{(e,e)} < 0, \quad \forall \, e \in \mathcal{E},
$$

and consequently the hyperplane $\mathcal{W}^{(\mathcal{G})}$ intersects the polytope $\mathcal{Q}_{\mathcal{G}}(a)$ in its interior, with the vertices $\{\mathcal{Z}_e^{(2)}, \mathcal{Z}_e^{(3)}\}_{e \in \mathcal{E}}$ on $\mathcal{Q}_{\mathcal{G}} \cap \mathcal{W}^{(\mathcal{G})}$ but with $\{\mathcal{Z}_e^{(1)}\}_{e \in \mathcal{E}}$ and $\{\mathcal{Z}_e^{(4)}\}_{e \in \mathcal{E}}$ in the positive and negative half space identified by $\mathcal{W}^{(\mathcal{G})}$, respectively.

[19]Note that, as it is not possible to mark only all the edges in the middle because of the relation (71), then no hyperplane is associated to the $\mathfrak{g} = \mathcal{G}$.

$\mathcal{Q}_{\mathcal{G}} \cap \widehat{\mathcal{W}}_a^{(\mathfrak{g})}$ is instead given by marking the cut edges away from the sites in $\mathfrak{g}$, double-marking all the edges in $\mathfrak{g}$, and marking the edges outside of $\mathfrak{g}$ with $\times$ :

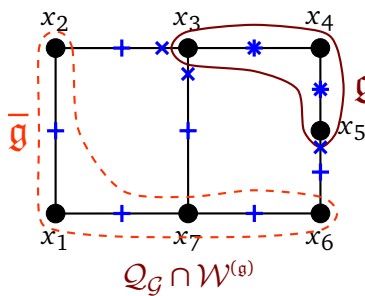
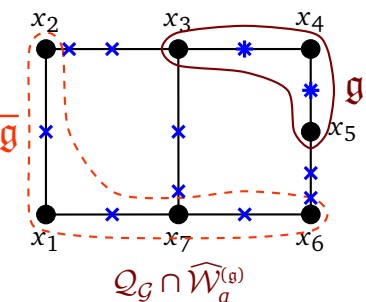

For the sake of concreteness, let us focus on the markings above. It is easy to see that if we remove any of the markings $+$ outside of $\mathfrak{g}$ in the left picture, then we land on a configuration of the type (72), with an edge having both the associated $\alpha$ and $\widehat{\alpha}$ set to zero, which is not allowed. Similarly, for the picture on the right, removing any of the middle markings outside of $\mathfrak{g}$ produces the similar type of non-allowed configurations. If instead any of the markings $\times$ is removed in the left picture, then the linear relation in the first line of (66) is no longer satisfied, while if $+$ inside of $\mathfrak{g}$ is removed, the linear relation in the second line of (66) is violated. A similar reasoning applies also to the right picture. Removing any of the markings in the left picture, which is equivalent to requiring that more vertices of $\mathcal{Q}_{\mathcal{G}}(a)$ belong to the pair of hyperplanes $(\mathcal{W}^{(\mathfrak{g})}, \widehat{\mathcal{W}}_a^{(\mathfrak{g})})$, results into the appearance of inconsistent markings and, consequently, the violations of the relations (66) and (71): the graphical rule above defined provides the facets $\{\mathcal{Q}_{\mathcal{G}}(a) \cup \mathcal{W}^{(\mathfrak{g})} \neq \varnothing, \mathcal{Q}_{\mathcal{G}}(a) \cup \widehat{\mathcal{W}}_a^{(\mathfrak{g})} \neq \varnothing, \forall\, \mathfrak{g} \subset \mathcal{G}\}$. This also implies that if $\tilde{\nu}$ is the number of subgraphs $\mathfrak{g} \subseteq \mathcal{G}$, then the polytope $\mathcal{Q}_{\mathcal{G}}(a)$ has $2(\tilde{\nu}-1)$ facets.

The vertex configurations of the two hyperplanes $\mathcal{W}^{(\mathfrak{g})}$ and $\widehat{\mathcal{W}}_a^{(\mathfrak{g})}$ differ by containing the vertices $\{\mathcal{Z}_e^{(1)}, e \in \mathcal{E}_{\bar{\mathfrak{g}}} \cup \mathcal{L}\}$ and excluding $\{\mathcal{Z}_e^{(4)}, e \in \mathcal{E}_{\bar{\mathfrak{g}}} \cup \mathcal{L}\}$ (the former) and vice versa (the latter). The hyperplanes $\mathcal{W}^{(\mathfrak{g})}$ and $\widehat{\mathcal{W}}_a^{(\mathfrak{g})}$ associated to these facets can easily be seen to be

$$
\begin{aligned}
\mathcal{W}^{(\mathfrak{g})} &= \sum_{s \in \mathcal{V}_{\mathfrak{g}}} \tilde{\mathbf{x}}_s + \sum_{e \in \mathcal{E}_{\mathfrak{g}}^{\text{ext}}} \tilde{\mathbf{y}}_e \,, \\
\widehat{\mathcal{W}}_a^{(\mathfrak{g})} &= \sum_{s \in \mathcal{V}_{\mathfrak{g}}} [1 + (n_{\text{ext}} - 1)(1+a)]\tilde{\mathbf{x}}_s + \sum_{\bar{s} \in \mathcal{V}_{\bar{\mathfrak{g}}}} n_{\text{ext}}(1+a)\tilde{\mathbf{x}}_{\bar{s}} + a \sum_{e \in \mathcal{E}_{\mathfrak{g}}^{\text{ext}}} \tilde{\mathbf{y}}_e \,,
\end{aligned}
\tag{76}
$$

with $\mathcal{V}_{\mathfrak{g}} \cup \mathcal{V}_{\bar{\mathfrak{g}}} = \mathcal{V}$ and $n_{\text{ext}}$ the number of edges departing from $\mathfrak{g}$, i.e. $n_{\text{ext}} := \dim\{\mathcal{E}_{\mathfrak{g}}^{\text{ext}}\}$. It is useful to also introduce a marking for the vertices which _are_ on a facet via

$$
\underset{\substack{\alpha_{(e,e)} = \mathcal{W} \cdot \mathcal{Z}_e^{(1)} = 0}}{\bullet\!\!-\!\!\circ\!\!-\!\!\bullet}_{\substack{x_{s_e} \quad y_e \quad x_{s'_e}}} \,,
\quad
\underset{\substack{\alpha_{(e,s_e)} = \mathcal{W} \cdot \mathcal{Z}_e^{(2)} = 0}}{\bullet\!\!\circ\!\!-\!\!\bullet}_{\substack{x_{s_e} \quad y_e \quad x_{s'_e}}} \,,
\quad
\underset{\substack{\alpha_{(e,s'_e)} = \mathcal{W} \cdot \mathcal{Z}_e^{(3)} = 0}}{\bullet\!\!-\!\!\circ\!\!\bullet}_{\substack{x_s \quad y_e \quad x_{s'_e}}} \,,
\quad
\underset{\substack{\widehat{\alpha}_{(e,e)} = \mathcal{W} \cdot \mathcal{Z}_e^{(4)} = 0}}{\bullet\!\!-\!\!\diamond\!\!-\!\!\bullet}_{\substack{x_s \quad y_e \quad x_{s'_e}}} \,,
$$

and, hence, the vertex configuration of the facets $\mathcal{Q}_{\mathcal{G}} \cap \mathcal{W}^{(\mathfrak{g})}$ and $\mathcal{Q}_{\mathcal{G}} \cap \widehat{\mathcal{W}}_a^{(\mathfrak{g})}$ can be represented alternatively as

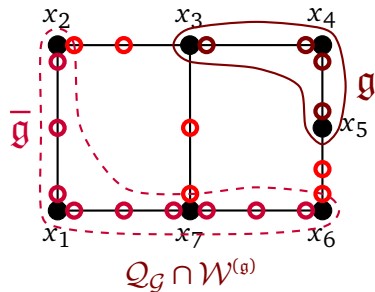
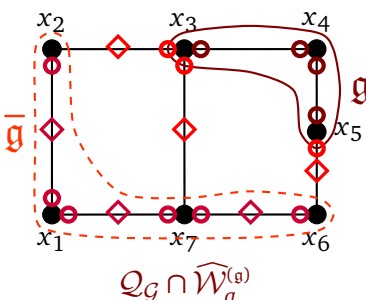

Note that this marking makes manifest the peculiar vertex structure associated to the edges in $\mathfrak{g}$, that characterises a (lower-dimensional) scattering facet $\mathcal{S}_\mathfrak{g}$ [24] – with each vertex on it indicated via $\mathbf{o}$, as well as the factorisation of the canonical function of these facets in the form

$$
\begin{aligned}
\Omega(\mathcal{Y}, \mathcal{Q}_\mathcal{G} \cap \mathcal{W}^{(\mathfrak{g})}) &= \Omega(\mathcal{Y}_\mathfrak{g}, \mathcal{S}_\mathfrak{g}) \times \Omega(\mathcal{Y}_{\overline{\mathfrak{g}}_c}, \mathcal{P}_{\overline{\mathfrak{g}}_c}), \\
\Omega(\mathcal{Y}, \mathcal{Q}_\mathcal{G} \cap \widehat{\mathcal{W}}_a^{(\mathfrak{g})}) &= \Omega(\mathcal{Y}_\mathfrak{g}, \mathcal{S}_\mathfrak{g}) \times \Omega(\mathcal{Y}_{\overline{\mathfrak{g}}_c}, \widehat{\mathcal{P}}_{\overline{\mathfrak{g}}_c}),
\end{aligned}
\tag{77}
$$

where $\overline{\mathfrak{g}}_c := \overline{\mathfrak{g}} \cup \mathscr{L}$ indicates the union[20] between the complementary graph $\overline{\mathfrak{g}}$ and the set of cut edges $\mathscr{L}$, while $\mathcal{P}_{\overline{\mathfrak{g}}_c}$ and $\widehat{\mathcal{P}}_{\overline{\mathfrak{g}}_c}$ are the associated polytopes contained in $\mathcal{Q}_\mathcal{G} \cap \mathcal{W}^{(\mathfrak{g})}$ and $\mathcal{Q}_\mathcal{G} \cap \widehat{\mathcal{W}}_a^{(\mathfrak{g})}$ respectively.

The knowledge of the facet structure of $\mathcal{Q}_\mathcal{G}(a)$ allows to write its canonical form as

$$
\omega\big(\mathcal{Y}, \mathcal{Q}_\mathcal{G}(a)\big) = \frac{\mathfrak{n}_\delta(\mathcal{Y}; a)\langle \mathcal{Y} d^{n_s+n_e-1}\mathcal{Y}\rangle}{\prod\limits_{\mathfrak{g} \subset \mathcal{G}}\big(\mathcal{Y} \cdot \mathcal{W}^{(\mathfrak{g})}\big)\big(\mathcal{Y} \cdot \widehat{\mathcal{W}}_a^{(\mathfrak{g})}\big)},
\tag{78}
$$

with the numerator $\mathfrak{n}_\delta(\mathcal{Y}; a)$ being a polynomial of degree $\delta = 2(\tilde{v}-1) - n_s - n_e$ which is fixed by the compatibility conditions among the facets and, therefore, by the structure of higher codimension faces of $\mathcal{Q}_\mathcal{G}(a)$.

Let us now consider the canonical form (78) in the limit $a \longrightarrow -1$. It provides the canonical form for the optical polytope $\mathcal{O}_\mathcal{G} := \mathcal{Q}_\mathcal{G}(-1)$:

$$
\omega(\mathcal{Y}, \mathcal{O}_\mathcal{G}) := \omega(\mathcal{Y}, \mathcal{Q}_\mathcal{G}(-1)) = \frac{\mathfrak{n}_\delta(\mathcal{Y}; -1)\langle \mathcal{Y} d^{n_s+n_e-1}\mathcal{Y}\rangle}{\prod\limits_{\mathfrak{g} \subset \mathcal{G}}\big(\mathcal{Y} \cdot \mathcal{W}^{(\mathfrak{g})}\big)\big(\mathcal{Y} \cdot \widehat{\mathcal{W}}_{-1}^{(\mathfrak{g})}\big)}.
\tag{79}
$$

In this limit, each vertex $\mathcal{Z}_e^{(4)}$ becomes a linear combination of the other three vertices associated to the same edge. This is reflected into the fact that the relation (71) can be recast in this limit into

$$
\widehat{\alpha}_{(e,e)}\big|_{a=-1} = \alpha_{(e,e)} + \alpha_{(e,s_e)} + \alpha_{(e,s_e')},
\tag{80}
$$

which also implies that $\mathcal{Z}_e^{(4)}$ is the centroid of the triangle identified by the vertices $\{\mathcal{Z}_e^{(1)}, \mathcal{Z}_e^{(2)}, \mathcal{Z}_e^{(3)}\}$ if all the $\alpha$'s on the right-hand-side are positive. The linear dependence (80) makes the two linear relations (69) equivalent for $a = -1$. Because of (80), whenever the three vertices $\{\mathcal{Z}_e^{(1)}, \mathcal{Z}_e^{(2)}, \mathcal{Z}_e^{(3)}\}$ associated to a given edge $e$ are on a facet, also $\mathcal{Z}_e^{(4)}$ is on the same facet, while it is enough that one of them is not on the facet for $\mathcal{Z}_e^{(4)}$ to also not be.

Note also that the expressions for the hyperplanes $(\mathcal{W}^{(\mathfrak{g})}, \widehat{\mathcal{W}}_{-1}^{(\mathfrak{g})})$ in terms of the canonical basis of $\mathbb{R}^{n_s+n_e}$ represented by the covectors $\{\tilde{\mathbf{x}}_s, s \in \mathcal{V}\}$ and $\{\tilde{\mathbf{y}}_e, e \in \mathcal{E}\}$ can be obtained from (76). They are

$$
\mathcal{W}^{(\mathfrak{g})} = \sum_{s \in \mathcal{V}_\mathfrak{g}} \tilde{\mathbf{x}}_s + \sum_{e \in \mathcal{E}_\mathfrak{g}^{\text{ext}}} \tilde{\mathbf{y}}_e, \qquad \widehat{\mathcal{W}}_{-1}^{(\mathfrak{g})} = \sum_{s \in \mathcal{V}_\mathfrak{g}} \tilde{\mathbf{x}}_s - \sum_{e \in \mathcal{E}_\mathfrak{g}^{\text{ext}}} \tilde{\mathbf{y}}_e.
\tag{81}
$$

Let us consider the edges $e \in \mathcal{E}_\mathfrak{g}^{\text{ext}}$ and the sites $s_e \in \mathcal{V}_\mathfrak{g}$ from which they depart. Then, for each of them, $\mathcal{O}_\mathcal{G} \cap \mathcal{W}^{(\mathfrak{g})} \neq \varnothing$ with $\alpha_{(e,e)} = 0$, $\alpha_{(e,s_e')} = 0$, $\alpha_{(e,s_e)} > 0$ and, consequently, $\widehat{\alpha}_{(e,e)} > 0$. Similarly, it is easy to see that when we consider $\mathcal{O}_\mathcal{G} \cap \widehat{\mathcal{W}}_{-1}^{(\mathfrak{g})} \neq \varnothing$, $\alpha_{(e,s_e)} = 0$ and $\widehat{\alpha}_{(e,e)} = 0$ with $\alpha_{(e,e)} > 0$ while $\alpha_{(e,s_e')} < 0$ which signals that the canonical form (79) is associated to a geometry which is not positive. If instead $e \in \mathcal{E}_\mathfrak{g}$, then $\alpha_{(e,s_e)} = 0$, $\alpha_{(e,s_e')} = 0$ with $\alpha_{(e,e)} > 0$ and $\widehat{\alpha}_{(e,e)} > 0$ for both $\mathcal{O}_\mathcal{G} \cap \mathcal{W}^{(\mathfrak{g})} \neq \varnothing$ and $\mathcal{O}_\mathcal{G} \cap \widehat{\mathcal{W}}_{-1}^{(\mathfrak{g})} \neq \varnothing$.

---

[20]This is an abuse of notation since $\overline{\mathfrak{g}}_c := \overline{\mathfrak{g}} \cup \mathscr{L}$ is not a graph in the ordinary sense: $\overline{\mathfrak{g}}_c$ includes $\overline{\mathfrak{g}}$ and all the edges in $\mathscr{L}$, but not those sites outside of $\overline{\mathfrak{g}}$ on which these edges end.



The graphical rules which associate a subgraph $\mathfrak{g} \subset \mathcal{G}$ to a pair of facets of $\mathcal{Q}_{\mathcal{G}}(a)$, can be generalised to $\mathcal{O}_{\mathcal{G}}$. For each $\mathfrak{g} \subset \mathcal{G}$, the vertex structure of $\mathcal{O}_{\mathcal{G}} \cap \mathcal{W}^{(\mathfrak{g})}$ can be obtained by marking the cut edge close to the sites in $\mathfrak{g}$ via ✕, as well as marking the edges inside $\mathfrak{g}$ with both ✕ and ✚ . The vertex structure of $\mathcal{O}_{\mathcal{G}} \cap \mathcal{W}^{(\mathfrak{g})}_{-1}$ is instead obtained by double marking the edges inside $\mathfrak{g}$ in the middle, as for the previous case, as well as by marking the cut edges close to the site in $\overline{\mathfrak{g}}$ as well as in the middle with ✕

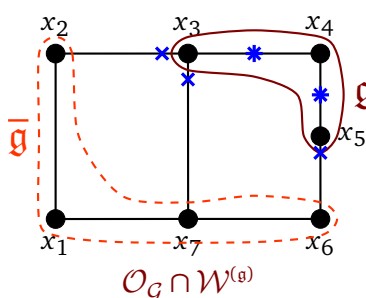

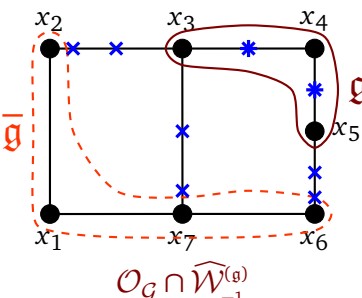

$$\mathcal{O}_{\mathcal{G}} \cap \mathcal{W}^{(\mathfrak{g})} \qquad \mathcal{O}_{\mathcal{G}} \cap \widehat{\mathcal{W}}^{(\mathfrak{g})}_{-1}$$

It is important to notice that $\mathcal{O}_{\mathcal{G}} \cap \widehat{\mathcal{W}}^{(\mathfrak{g})}_{-1}$ allows for markings such as ———●✕✕——— as now $\alpha_{(e,e)} > 0$ and $\alpha_{(e,s_e)} < 0$.

We are now left to discuss the adjoint surface of $\mathcal{Q}_{\mathcal{G}}(a)$ and $\mathcal{O}_{\mathcal{G}}$, which fixes the numerator of the canonical form and is fixed by the compatibility conditions among the facets. As $\mathcal{Q}_{\mathcal{G}}(a)$ is a positive geometry, the adjoint is the locus of the intersections among all the hyperplanes $\{\mathcal{W}^{(\mathfrak{g})}, \widehat{\mathcal{W}}^{(\mathfrak{g})}_{-1}, \forall \mathfrak{g} \subset \mathcal{G}\}$, outside of $\mathcal{Q}_{\mathcal{G}}(a)$. For the non-positive geometry $\mathcal{O}_{\mathcal{G}}$, the adjoint is the locus of such intersections *inside* $\mathcal{O}_{\mathcal{G}}$ and on its boundaries.

The compatibility conditions among the intersections of the facets, are of extreme importance as they provide a different characterisation of a polyope and hence, a further invariant definition: if both the facets and the compatibility conditions are given, then the polytope is determined. As the optical polytope $\mathcal{O}_{\mathcal{G}}$ we are actually interested in has been defined as the non-convex limit $a \longrightarrow -1$ of $\mathcal{Q}_{\mathcal{G}}(a)$, the compatibility conditions for the facets of $\mathcal{Q}_{\mathcal{G}}(a)$ map into compatibility conditions for the facets of the optical polytope $\mathcal{O}_{\mathcal{G}}$. Turning the table around, the set of such compatibility conditions, together with the set of facets of $\mathcal{O}_{\mathcal{G}}$, $\{\mathcal{W}^{(\mathfrak{g})}, \widehat{\mathcal{W}}^{(\mathfrak{g})}_{-1}, \mathfrak{g} \subset \mathcal{G}\}$, provide a novel definition of a non-convex polytope, independent of a previous notion of a convex polytope $\mathcal{Q}_{\mathcal{G}}(a)$, or of polytope subdivision in terms of convex polytope (which is their usual definition), putting them on the same footing.

**Higher codimension faces and compatibility conditions.** A face of codimension $k$ of the polytope $\mathcal{Q}_{\mathcal{G}}(a) \subset \mathbb{P}^{n_s + n_e - 1}$ is a polytope $\mathcal{Q}_{\mathcal{G}}(a) \cap \mathcal{W}^{(\mathfrak{g}_1 \cdots \mathfrak{g}_k)}_a \neq \varnothing$ living in $\mathbb{P}^{n_s + n_e - k - 1}$ identified by the codimension-$k$ hyperplane $\mathcal{W}^{(\mathfrak{g}_1 \cdots \mathfrak{g}_k)}_a := \cap^k_{j=1} \widetilde{\mathcal{W}}^{(\mathfrak{g}_j)}_a$, where each $\widetilde{\mathcal{W}}^{(\mathfrak{g}_j)}_a$ ($j = 1, \ldots, k$) is a hyperplane containing a facet and, hence, it can either be $\mathcal{W}^{(\mathfrak{g}_j)}$ or $\widehat{\mathcal{W}}^{(\mathfrak{g}_j)}_a$. Turning the table around, given a codimension-$k$ hyperplane $\mathcal{W}^{(\mathfrak{g}_1 \cdots \mathfrak{g}_k)}_a$, if $\mathcal{Q}_{\mathcal{G}}(a) \cap \mathcal{W}^{(\mathfrak{g}_1 \cdots \mathfrak{g}_k)}_a = \varnothing$ in codimension-$k$, then $\mathcal{W}^{(\mathfrak{g}_1 \cdots \mathfrak{g}_k)}_a$ lies outside $\mathcal{Q}_{\mathcal{G}}(a)$. The locus of the intersections of hyperplanes containing the facets outside of $\mathcal{Q}_{\mathcal{G}}(a)$ determines the numerator of the canonical form and the conditions which determine such intersections is what we refer to as compatibility conditions—they provide the information about the vanishing multiple residues of the canonical form and, hence, $\Delta \psi_{\mathcal{G}}$.

The general logic for unveiling the face structure in arbitrary codimension is the very same used in [27,46]: the intersection $\mathcal{Q}_{\mathcal{G}}(a) \cap \mathcal{W}^{(\mathfrak{g}_1 \cdots \mathfrak{g}_k)}_a$ factorises into $2^k$ subspaces and, in order to occur in codimension $k$, the sum of the dimensionalities of such subspaces ought to be equal to the dimensionality of $\mathbb{P}^{n_s + n_e - k - 1}$. Because of the correspondence between subgraphs and hyperplanes $\{\mathcal{W}^{(\mathfrak{g})}, \widehat{\mathcal{W}}^{(\mathfrak{g})}_a, \mathfrak{g} \subset \mathcal{G}\}$, when considering the intersections $\mathcal{W}^{(\mathfrak{g}_1 \cdots \mathfrak{g}_k)}$, the graph $\mathcal{G}$ is

decomposed into $2^k$ subgraphs which are identified by the intersection among the different $\{\mathfrak{g}_j \subset \mathcal{G}, j = 1, \ldots, k\}$, their complementaries $\{\bar{\mathfrak{g}}_j \subset \mathcal{G}, j = 1, \ldots, k\}$ and the elements of these two subsets

$$
\mathcal{G}|_{\mathcal{Q}_{\mathcal{G}} \cap \mathcal{W}^{(\mathfrak{g}_1 \cdots \mathfrak{g}_k)}} = \bigcup_{j=1}^{k} \bigcup_{\sigma(j) \in \widetilde{S}_k} \mathfrak{g}_{\sigma(1)} \cap \mathfrak{g}_{\sigma(j)} \cap \bar{\mathfrak{g}}_{\sigma(j+1)} \cap \ldots \cap \bar{\mathfrak{g}}_{\sigma(k)} := \bigcup_{j=1}^{k} \bigcup_{\sigma(j) \in \widetilde{S}_k} \mathfrak{g}_{\sigma(j)}^{(\cap)}, \qquad (82)
$$

where $\widetilde{S}_k = \{1, \ldots, k \,|\, \sigma(r) < \sigma(r+1), r = 1, \ldots, j-1, \& \sigma(s) < \sigma(s+1), s = j, \ldots, k\}$. Importantly, it is not necessary that all the intersections in (82) are non-empty, rather there should be a sufficient number of them which are. It is straightforward to see that, as it happens for the cosmological polytope, for $2^k - 1$ of them the corresponding polytope in principle has the vertex structure of a low-dimensional scattering facet, *i.e.* it corresponds to a polytope given by the vertices $\{\mathbf{x}_{s_e} + \mathbf{y}_e - \mathbf{x}_{s'_e}, -\mathbf{x}_{s_e} + \mathbf{y}_e + \mathbf{x}_{s'_e}, \forall e \in \mathcal{E}_{\mathfrak{g}_{\sigma(j)}^{(\cap)}}\} - \mathcal{E}_{\mathfrak{g}_{\sigma(j)}^{(\cap)}}$ is the set of edges associated to the intersection $\mathfrak{g}_{\sigma(j)}^{(\cap)}$. The intersection of all the complementary graphs is the only one which does not have such a structure. Following the same counting as in [31, 46], the dimension of $\mathcal{Q}_{\mathcal{G}}(a) \cap \mathcal{W}^{(\mathfrak{g}_1 \cdots \mathfrak{g}_k)}$ is given by

$$
\dim\{\mathcal{Q}_{\mathcal{G}}(a) \cap \mathcal{W}^{(\mathfrak{g}_1 \cdots \mathfrak{g}_k)}\} = n_s + n_e - \sum_{\mathcal{S}_{\mathfrak{g}}} 1 - \not{n}_{\not{\mathcal{E}}} - 1, \qquad (83)
$$

with $\mathcal{S}_{\mathfrak{g}}$ indicating the scattering facets, and $\not{n}_{\not{\mathcal{E}}}$ being the number of cut edges whose associated vertices are not in the subspace $\mathcal{Q}_{\mathcal{G}}(a) \cap \mathcal{W}^{(\mathfrak{g}_1 \cdots \mathfrak{g}_k)}$. Hence, in order for $\mathcal{Q}_{\mathcal{G}}(a) \cap \mathcal{W}^{(\mathfrak{g}_1 \cdots \mathfrak{g}_k)}$ to be non-empty in codimension-$k$, the following condition needs to be satisfied

$$
\sum_{\mathcal{S}_{\mathfrak{g}}} 1 + \not{n}_{\not{\mathcal{E}}} = k, \qquad (84)
$$

*i.e.* there should be $k - \not{n}_{\not{\mathcal{E}}}$ non-empty intersections $\mathfrak{g}_{\sigma(j)}^{(\cap)}$ (excluding $\bar{\mathfrak{g}}_1 \cap \ldots \cap \bar{\mathfrak{g}}_k$ which can be equivalently empty or non-empty). If the hyperplane $\mathcal{W}^{(\mathfrak{g}_1 \cdots \mathfrak{g}_k)}$ is such that the compatibility condition is not satisfied, then

$$
\text{Res}_{\mathcal{W}^{(\mathfrak{g}_1 \cdots \mathfrak{g}_k)}}\{\omega(\mathcal{Y}, \mathcal{Q}_{\mathcal{G}})\} := \text{Res}_{\widetilde{\mathcal{W}}^{(\mathfrak{g}_1)}} \ldots \text{Res}_{\widetilde{\mathcal{W}}^{(\mathfrak{g}_k)}} \omega(\mathcal{Y}, \mathcal{Q}_{\mathcal{G}}) = 0. \qquad (85)
$$

Note that

1. if $\mathcal{W}^{(\mathfrak{g}_1 \cdots \mathfrak{g}_k)}$ is defined through $\{\widetilde{\mathcal{W}}^{(\mathfrak{g}_j)} = \mathcal{W}^{(\mathfrak{g}_j)}, j = 1, \ldots, k\}$, then we recover both a subset of the Steinmann-like relations [27] and a subset of the higher-codimension conditions for $k > 2$ [46] for the wavefunction of the universe, as $\mathcal{Q}_{\mathcal{G}} \cap \mathcal{W}^{(\mathfrak{g}_j)} = \mathcal{P}_{\mathcal{G}} \cap \mathcal{W}^{(\mathfrak{g}_j)}$ for each $j = 1, \ldots, k$;

2. if $\mathcal{W}^{(\mathfrak{g}_1 \cdots \mathfrak{g}_k)}$ is instead defined through $\{\widetilde{\mathcal{W}}^{(\mathfrak{g}_j)} = \widehat{\mathcal{W}}^{(\mathfrak{g}_j)}, j = 1, \ldots, k\}$, then (85) gives a set of conditions which turns out to also constrain $\psi^{\dagger}$;

3. if finally $\mathcal{W}^{(\mathfrak{g}_1 \cdots \mathfrak{g}_k)}$ is defined via both types of hyperplanes, then the conditions (85) are truly compatibility conditions between Bunch-Davies singularities and folded ones. In particular, in codimension two

$$
\begin{aligned}
&\text{Res}_{\mathcal{W}^{(\mathfrak{g})}} \text{Res}_{\widehat{\mathcal{W}}^{(\mathfrak{g})}} \Omega(\mathcal{Y}, \mathcal{O}_{\mathcal{G}}) = 0, \\
&\text{Res}_{\mathcal{W}^{(\mathfrak{g})}} \text{Res}_{\widehat{\mathcal{W}}^{(\bar{\mathfrak{g}})}} \Omega(\mathcal{Y}, \mathcal{O}_{\mathcal{G}}) \sim \Omega(\mathcal{Y}_{\mathfrak{g}}, \mathcal{S}_{\mathfrak{g}}) \times \Omega(\mathcal{Y}_{\not{\mathcal{E}}}, \Sigma_{\not{\mathcal{E}}}) \times \Omega(\mathcal{Y}_{\bar{\mathfrak{g}}}, \mathcal{S}_{\bar{\mathfrak{g}}}), \\
&\text{Res}_{\mathcal{W}^{(\bar{\mathfrak{g}})}} \text{Res}_{\widehat{\mathcal{W}}^{(\mathfrak{g})}} \Omega(\mathcal{Y}, \mathcal{O}_{\mathcal{G}}) \sim \Omega(\mathcal{Y}_{\mathfrak{g}}, \mathcal{S}_{\mathfrak{g}}) \times \Omega(\mathcal{Y}_{\not{\mathcal{E}}}, \Sigma'_{\not{\mathcal{E}}}) \times \Omega(\mathcal{Y}_{\bar{\mathfrak{g}}}, \mathcal{S}_{\bar{\mathfrak{g}}}).
\end{aligned} \qquad (86)
$$

The first line holds for all $\mathfrak{g} \subset \mathcal{G}$ except those defined by all sites of $\mathcal{G}$ and all edges but one, for which (84) is satisfied. The other two are proven by considering that $\not{n}_{\not{\mathcal{E}}} = 0$

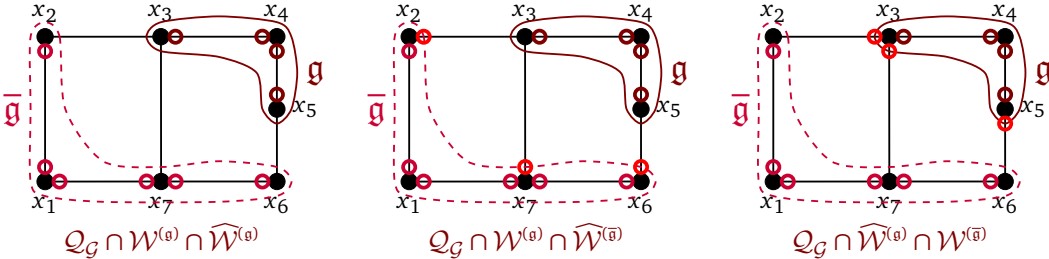

Figure 11: Codimension-2 faces of the optical polytope. The intersection of the two hyperplanes $\mathcal{W}^{(\mathfrak{g})}$ and $\widehat{\mathcal{W}}^{(\mathfrak{g})}$ associated to the same subgraph $\mathfrak{g} \subset \mathcal{G}$ (picture on the left) is not a face of the optical polytope: it should be a $k = 2$ boundary but it has two lower-dimensional scattering facets, identified by the open circles contained in $\mathfrak{g}$ and $\overline{\mathfrak{g}}$ respectively, and it has no vertices on the three cut edges, *i.e.* $\not{n}_{\not{\varepsilon}} = 3$. Hence, the condition (84) is not satisfied ($2 + 3 \neq 2$). The other two intersections have a similar structure, which differ by the presence of a vertex from each cut edge whose convex hull defines a lower-dimensional simplex: now $\not{n}_{\not{\varepsilon}} = 0$ in both cases and the condition (84) is satisfied. The vertices organise themselves into two scattering facets and a simplex which translates into the factorisation of the canonical function (86) and, hence, of $\Delta \psi_{\mathcal{G}}$ in (87).

and $\mathfrak{g} \cap \overline{\mathfrak{g}} = \varnothing$, so that (84) is fulfilled, as well as that the vertex structure associated to each of the two subgraphs is precisely the one characterising a low-dimensional scattering facet, $\mathcal{S}_{\mathfrak{g}}$ and $\mathcal{S}_{\overline{\mathfrak{g}}}$, and a simplex associated to the cut edge, $\Sigma_{\not{\varepsilon}}$ and $\Sigma'_{\not{\varepsilon}}$ in the second and third lines respectively—the two simplices differ from one another by the fact that $\Sigma_{\not{\varepsilon}}$ ($\Sigma'_{\not{\varepsilon}}$) is the convex hull of the vertices marked by an open circle $\circ$ close to $\overline{\mathfrak{g}}$ ($\mathfrak{g}$)—, see Figure 11.

In terms of $\Delta \psi_{\mathcal{G}}$, the conditions (86) translate into

$$\mathrm{Res}_{E_{\mathfrak{g}}} \mathrm{Res}_{\widehat{E}_{\mathfrak{g}}} \Delta \psi_{\mathcal{G}} = 0 \,,$$

$$\mathrm{Res}_{E_{\mathfrak{g}}} \mathrm{Res}_{\widehat{E}_{\overline{\mathfrak{g}}}} \Delta \psi_{\mathcal{G}} = \mathcal{A}_{\mathfrak{g}} \times \left( \prod_{e \in \not{\varepsilon}} \frac{1}{2 y_e} \right) \times \mathcal{A}_{\overline{\mathfrak{g}}} \,,$$

$$\mathrm{Res}_{\widehat{E}_{\mathfrak{g}}} \mathrm{Res}_{E_{\overline{\mathfrak{g}}}} \Delta \psi_{\mathcal{G}} = \mathcal{A}_{\mathfrak{g}} \times \left( \prod_{e \in \not{\varepsilon}} \frac{1}{2 y_e} \right) \times \mathcal{A}_{\overline{\mathfrak{g}}} \,,$$
(87)

for subgraphs $\mathfrak{g} \subset \mathcal{G}$ satisfying the properties mentioned above, with

$$E_{\mathfrak{g}} := \sum_{s \in \mathcal{V}_{\mathfrak{g}}} x_s + \sum_{e \in \not{\varepsilon}} y_e \,, \quad \text{and} \quad \widehat{E}_{\mathfrak{g}} := \sum_{s \in \mathcal{V}_{\mathfrak{g}}} x_s - \sum_{e \in \not{\varepsilon}} y_e \,.$$

The second and third lines in (87) differ from each other by the energy flux in the cut edges $\not{\varepsilon}$ only, which are geometrically identified by the different simplices $\Sigma_{\not{\varepsilon}}$ and $\Sigma'_{\not{\varepsilon}}$ respectively.

## 5.3 Triangulations and cutting rules

The optical polytope $\mathcal{O}_{\mathcal{G}}$ provides an invariant definition for $\Delta \psi_{\mathcal{G}}$, and ultimately is a non-positive part of the cosmological polytope $\mathcal{P}_{\mathcal{G}}$ sharing the very same boundaries except one, the scattering facet.

Let us begin by considering the polytope $\mathcal{Q}_{\mathcal{G}}(a)$. As it has been shown in the previous section, the hyperplane $\mathcal{W}^{(\mathcal{G})} := \sum_{s \in \mathcal{V}} \tilde{\mathbf{x}}_s$, intersects $\mathcal{Q}_{\mathcal{G}}(a)$ in its interior, with the vertices $\{\mathcal{Z}_e^{(2)}, \mathcal{Z}_e^{(3)}, e \in \mathcal{E}\}$ on $\mathcal{Q}_{\mathcal{G}}(a) \cup \mathcal{W}^{(\mathcal{G})}$, while the two sets of vertices $\{\mathcal{Z}_e^{(1)}, e \in \mathcal{E}\}$ and $\{\mathcal{Z}_e^{(4)}, e \in \mathcal{E}\}$ lie on the two different half-spaces identified by $\mathcal{W}^{(\mathcal{G})}$. A polytope subdivision of $\mathcal{Q}_{\mathcal{G}}(a)$ is then given by the union of the two polytopes $\mathcal{P}_{\mathcal{G}}$ and $\mathcal{P}_{\mathcal{G}}^{\dagger}(a)$ with vertices $\{Z_e^{(1)}, Z_e^{(2)}, Z_e^{(3)}\}_{e \in \mathcal{E}}$ and $\{Z_e^{(2)}, Z_e^{(3)}, Z_e^{(4)}\}_{e \in \mathcal{E}}$, respectively. The canonical form of $\mathcal{Q}_{\mathcal{G}}(a)$ can then be written as

$$\omega\left(\mathcal{Y}, \mathcal{Q}_{\mathcal{G}}(a)\right) = \omega\left(\mathcal{Y}, \mathcal{P}_{\mathcal{G}}\right) + \omega\left(\mathcal{Y}, \mathcal{P}_{\mathcal{G}}^{\dagger}(a)\right). \tag{88}$$

The two sides of the cosmological optical theorem can then be seen as different polytope subdivisions of the optical polytope. The left-hand-side is given by the polytope subdivision of $\mathcal{O}_{\mathcal{G}}$ via the hyperplane containing the scattering facet, $\mathcal{W}^{(\mathcal{G})}$, which can be obtained as the $a \longrightarrow -1$ limit of (88). The optical polytope $\mathcal{O}_{\mathcal{G}}$ gets then divided into the cosmolological polytope $\mathcal{P}_{\mathcal{G}}$ and another polytope $\mathcal{P}_{\mathcal{G}}^{\dagger} := \mathcal{P}_{\mathcal{G}}^{\dagger}(-1)$ which is isomorphic to $\mathcal{P}_{\mathcal{G}}$:

$$\omega\left(\mathcal{Y}, \mathcal{O}_{\mathcal{G}}\right) = \omega\left(\mathcal{Y}, \mathcal{P}_{\mathcal{G}}\right) + \omega\left(\mathcal{Y}, \mathcal{P}_{\mathcal{G}}^{\dagger}\right), \tag{89}$$

with $\mathcal{P}_{\mathcal{G}}^{\dagger}$ defined as the convex hull of the vertices

$$\{\mathbf{x}_{s_e} + \mathbf{y}_e - \mathbf{x}_{s_e'}, -\mathbf{x}_{s_e} + \mathbf{y}_e + \mathbf{x}_{s_e'}, \mathbf{x}_{s_e} + \mathbf{y}_e + \mathbf{x}_{s_e'}\}_{e \in \mathcal{E}}. \tag{90}$$

Importantly, the subdivision given by (89) can be obtained directly without making any reference to the convex polytope $\mathcal{Q}_{\mathcal{G}}(a)$.

The canonical form of $\mathcal{P}_{\mathcal{G}}$ provides the wavefunction coefficient $\psi_{\mathcal{G}}(x_s, y_e)$ associated to the graph $\mathcal{G}$. The canonical form of $\mathcal{P}_{\mathcal{G}}^{\dagger}$ contains folded singularities only, with the exception of the total energy singularity: it describes $\psi_{\mathcal{G}}^{\dagger}(-x_s, y_e)$. Besides (89), there are several other polytope subdivisions, or even triangulations. Let $\{\mathcal{P}^{(j)}, j = 1, \ldots, n\}$ be a collection of polytopes in $\mathbb{P}^{n_s + n_e - 1}$ such that their union returns $\mathcal{O}_{\mathcal{G}}$, provided that the elements of such a collection have compatible orientations. Then, its canonical form can be written as

$$\omega\left(\mathcal{Y}, \mathcal{O}_{\mathcal{G}}\right) = \sum_{j=1}^{n} \omega\left(\mathcal{Y}, \mathcal{P}^{(j)}\right), \tag{91}$$

for any collection $\{\mathcal{P}^{(j)}, j = 1, \ldots, n\}$ satisfying the conditions specified above. Note that (89) is a special case of (91), with the chosen collection being $\{\mathcal{P}_{\mathcal{G}}, \mathcal{P}_{\mathcal{G}}^{\dagger}\}$. The equivalence among all these representations for the canonical form of the optical polytope provides in particular the following equality

$$\omega\left(\mathcal{Y}, \mathcal{P}_{\mathcal{G}}\right) + \omega\left(\mathcal{Y}, \mathcal{P}_{\mathcal{G}}^{\dagger}\right) = \sum_{j=1}^{n} \omega\left(\mathcal{Y}, \mathcal{P}^{(j)}\right), \tag{92}$$

for any collection $\{\mathcal{P}^{(j)}, j = 1, \ldots, n\} \neq \{\mathcal{P}_{\mathcal{G}}, \mathcal{P}_{\mathcal{G}}^{\dagger}\}$.[21] The relation (92) is the geometric-combinatorial statement, and extension, of the cosmological cutting rules.

A triangulation is usually obtained by dividing a polytope in simplices via hyperplanes which intersect the polytope passing through a subset of its vertices and being distinct from its facets. These hyperplanes introduce spurious boundaries and translate, at the level of the canonical form, into spurious singularities which cancel upon summation. A systematic study of these triangulations is beyond the scope of the present paper. It would be interesting to apply the algebraic approach used in [57] for the triangulations of cosmological polytopes.

---

[21] The equality would just provide a completely trivial statement.

Another class of triangulations can be obtained by using points in the adjoint surface of $\mathcal{O}_\mathcal{G}$. Note that while for the convex polytope $\mathcal{Q}_\mathcal{G}(a)$ (with $a > 0$) the adjoint surface is identified by the intersections of the hyperplanes containing the facets of $\mathcal{Q}_\mathcal{G}(a)$ outside of $\mathcal{Q}_\mathcal{G}(a)$ itself, the adjoint surface for the non-positive geometry $\mathcal{O}_\mathcal{G}$ intersects $\mathcal{O}_\mathcal{G}$ in its interior, and it is identified by the intersections of the hyperplanes containing the facets of $\mathcal{O}_\mathcal{G}$ inside its facets themselves. Irrespectively of whether we are in presence of a positive or non-positive geometry, the adjoint surface encodes the zeroes of its canonical form and it is determined by the multiple residues (85), each of which identifies a zero of the canonical form which can be used to triangulate our polytope. These triangulations use the very same hyperplanes containing the facets of $\mathcal{Q}_\mathcal{G}(a)/\mathcal{O}_\mathcal{G}$: as no spurious boundaries are introduced, no spurious pole appears when the canonical form is decomposed into the sum of the canonical forms of simplices.

As discussed in Section 5.2, a given subgraph $\mathfrak{g} \subset \mathcal{G}$ is associated to a pair of hyperplanes $(\mathcal{W}^{(\mathfrak{g})}, \widehat{\mathcal{W}}_a^{(\mathfrak{g})})$ such that $\mathcal{Q}_\mathcal{G}(a) \cap \mathcal{W}^{(\mathfrak{g})}$ and $\mathcal{Q}_\mathcal{G}(a) \cap \widehat{\mathcal{W}}_a^{(\mathfrak{g})}$ are facets of $\mathcal{Q}_\mathcal{G}(a)$ – similarly for $\mathcal{O}_\mathcal{G}$ when $a = -1$. Each of these two facets is identfied by a marking. Let $\mathfrak{m}_\mathfrak{g}$ and $\widehat{\mathfrak{m}}_\mathfrak{g}$ be the markings associated to $\mathcal{Q}_\mathcal{G}(a) \cap \mathcal{W}^{(\mathfrak{g})}$ and $\mathcal{Q}_\mathcal{G}(a) \cap \widehat{\mathcal{W}}_a^{(\mathfrak{g})}$ respectively. Let $\mathfrak{M}_\circ$ be the set of markings – which can contain either or both the type of markings $\mathfrak{m}_\mathfrak{g}$ and $\widehat{\mathfrak{m}}_\mathfrak{g}$ – identifying the hyperplane $\mathcal{W}_a^{(\mathfrak{g}_1 \cdots \mathfrak{g}_k)}$ such that $\mathcal{Q}_\mathcal{G}(a) \cap \mathcal{W}_a^{(\mathfrak{g}_1 \cdots \mathfrak{g}_k)} = \varnothing$ – recall that $\mathcal{W}_a^{(\mathfrak{g}_1 \cdots \mathfrak{g}_k)} := \cap_{j=1}^k \widetilde{\mathcal{W}}^{(\mathfrak{g}_j)}$ where $\widetilde{\mathcal{W}}^{(\mathfrak{g}_j)}$ can be either $\mathcal{W}^{(\mathfrak{g}_j)}$ or $\widehat{\mathcal{W}}_a^{(\mathfrak{g}_j)}$. By definition, $\mathcal{W}_a^{(\mathfrak{g}_1 \cdots \mathfrak{g}_k)}$ is a subspace of the adjoint surface of $\mathcal{Q}_\mathcal{G}(a)$. Let us assume that we can perform a triangulation of $\mathcal{Q}_\mathcal{G}(a)$ through it: the simplices involved are given by $k$ inequalities associate to $\mathfrak{M}_\circ$ together with $(n_s + n_e - k)$ more inequalities associated to the hyperplane $\mathcal{W}_a^{(\mathfrak{g}_{\sigma(1)} \cdots \mathfrak{g}_{\sigma(n_s+n_e-k)})} := \bigcap_{j=1}^{n_s+n_e-k} \widetilde{\mathcal{W}}_{(a)}^{(\mathfrak{g}_{\sigma(j)})}$ identified by the markings $\mathfrak{m} \notin \mathfrak{M}_\circ$ such that they identify an $(n_s + n_e - k)$-dimensional face of $\mathcal{Q}_\mathcal{G}(a)$, i.e.

$$\mathrm{Res}_{\widetilde{\mathcal{W}}^{(\mathfrak{g}_{\sigma(1)})}}\mathrm{Res}_{\widetilde{\mathcal{W}}^{(\mathfrak{g}_{\sigma(2)})}} \ldots \mathrm{Res}_{\widetilde{\mathcal{W}}^{(\mathfrak{g}_{\sigma(n_s+n_e-k)})}} \omega(\mathcal{Y}, \mathcal{Q}_\mathcal{G}(a)) \neq 0, \tag{93}$$

where the $\mathfrak{g}_{\sigma(j)}$'s can identify all the subgraphs strictly contained in $\mathcal{G}$ exept $\{\mathfrak{g}_j, j = 1, \ldots, k\}$. Hence, given $\mathfrak{M}_\circ$, each possible set $\mathfrak{M}_c$ of markings with $(n_s + n_e - k)$ elements $\mathfrak{m} \notin \mathfrak{M}_\circ$ such that (86) is satisfied, defines the collection of simplices triangulating $\mathcal{Q}_\mathcal{G}(a)$. The canonical form of $\mathcal{Q}_\mathcal{G}(a)$ can thus be written as

$$\omega(\mathcal{Y}, \mathcal{Q}_\mathcal{G}(a)) = \sum_{\{\mathfrak{M}_c\}} \prod_{\mathfrak{m}' \in \mathfrak{M}_c} \frac{1}{q_{\mathfrak{m}'}(\mathcal{Y})} \frac{\langle \mathcal{Y} d^{n_s+n_e-1} \mathcal{Y} \rangle}{\prod_{\mathfrak{m} \in \mathfrak{M}_\circ} q_\mathfrak{m}(\mathcal{Y})}, \tag{94}$$

where $q_\mathfrak{m}(\mathcal{Y}) := \mathcal{Y} \cdot \widetilde{\mathcal{W}}_a^{(\mathfrak{g})}$, and the sum runs over all the possible sets $\mathfrak{M}_c$. Importantly, (94) represents all the possible ways in which the canonical form $\omega(\mathcal{Y}, \mathcal{Q}_{\mathcal{G}(a)})$ can be triangulated without introducing spurious poles. These representations are identified by the choice of $\mathfrak{M}_\circ$. Subspaces of the adjoint surface are determined by those markings covering completely the graph $\mathcal{G}$. As a further comment, the canonical form triangulation (94) is valid also directly for the optical polytope $\mathcal{O}_\mathcal{G}$: this can be seen by either taking the limit $a \longrightarrow -1$ in (94) or, more invariantly, by using the compatibility conditions for $\mathcal{O}_\mathcal{G}$.

Finally, note that both (20) and the holomorphic cutting rules (23) can be associated to triangulations that do not fall in any of the two classes just described. Rather, they are obtained via the special points $\{\mathbf{x}_s, s \in \mathcal{V}\}$: they introduce spurious boundaries identified by the hyperplanes $\{\mathcal{W}^{(e)} := \tilde{\mathbf{y}}_e, e \in \mathcal{E}\}$ – each of these hyperplanes contains the vertices of $\mathcal{O}_\mathcal{G}$ $\{\mathcal{Z}_{e'}^{(1)}, \mathcal{Z}_{e'}^{(2)}, \mathcal{Z}_{e'}^{(3)}, \mathcal{Z}_{e'}^{(4)}, e' \in \mathcal{E} \setminus \{e\}\}$ as well as all the special points $\{\mathbf{x}_s, s \in \mathcal{V}\}$.

For the sake of clarity and concreteness let us analyse some simple examples.

**The optical polytope and the two-site line graph.** Let us begin with considering the optical polytope associate to the two-site line graph, which is a non-convex quadrilateral with vertices

$$\{\mathbf{x}_1 - \mathbf{y}_{12} + \mathbf{x}_2, \ \mathbf{x}_1 + \mathbf{y}_{12} - \mathbf{x}_2, \ -\mathbf{x}_1 + \mathbf{y}_{12} + \mathbf{x}_2, \ \mathbf{x}_1 + \mathbf{y}_{12} + \mathbf{x}_2 \}.$$

A first class of triangulations can be obtained via a line passing through its non-adjacent vertices:

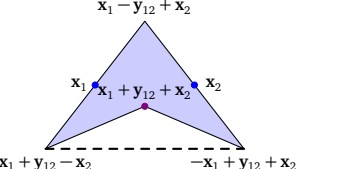
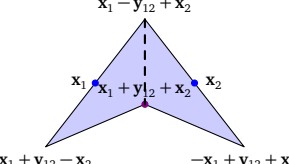

where the spurious boundary is depicted with a dashed line. In the first picture, the spurious boundary intersects the optical polytope in its vertices $\{\mathbf{x}_1 + \mathbf{y}_{12} - \mathbf{x}_2, -\mathbf{x}_1 + \mathbf{y}_{12} + \mathbf{x}_2\}$ only, and triangulates the optical polytope into the two triangles

$$\{\mathbf{x}_1 - \mathbf{y}_{12} + \mathbf{x}_2, \ \mathbf{x}_1 + \mathbf{y}_{12} - \mathbf{x}_2, \ -\mathbf{x}_1 + \mathbf{y}_{12} + \mathbf{x}_2\},$$

and

$$\{\mathbf{x}_1 + \mathbf{y}_{12} - \mathbf{x}_2, \ -\mathbf{x}_1 + \mathbf{y}_{12} + \mathbf{x}_2, \ \mathbf{x}_1 + \mathbf{y}_{12} + \mathbf{x}_2\}.$$

Such triangles correspond respectively to the cosmological polytope and $\mathcal{P}_{\mathcal{G}}^{\dagger}(-1)$, providing the left-hand-side of (92). Note that these two triangles share the segment $\{\mathbf{x}_1 + \mathbf{y}_{12} + \mathbf{x}_2, -\mathbf{x}_1 + \mathbf{y}_{12} + \mathbf{x}_2\}$ but with different orientation. Such a segment is nothing but the scattering facet of $\mathcal{P}_{\mathcal{G}}$ which is a spurious boundary for $\mathcal{O}_{\mathcal{G}}$. The second picture above is the triangulation of $\mathcal{O}_{\mathcal{G}}$ via the line passing through the vertices $\{\mathbf{x}_1 - \mathbf{y}_{12} + \mathbf{x}_2, \mathbf{x}_1 + \mathbf{y}_{12} + \mathbf{x}_2\}$, *i.e.* $\mathcal{W}^{(14)} := \tilde{\mathbf{x}}_1 - \tilde{\mathbf{x}}_2$.

The equivalence between these two triangulations can be interpreted diagrammatically as

$$\overset{y_{12}}{\underset{x_1 \quad x_2}{\bullet\!\!-\!\!\bullet}} \ + \ \overset{y_{12}}{\underset{-x_1 \ -x_2}{\circ\!\!-\!\!\circ}} \ = \ \frac{1}{x_1 - x_2}\left[ \underset{x_1 - y_{12}}{\circ\!-\!-\!-\!\bullet}^{\,y_{12}+x_2} - \underset{x_1 + y_{12}}{\bullet\!-\!-\!-\!\circ}^{\,-y_{12}+x_2} \right], \tag{95}$$

where the one-site graphs in which $\Delta\psi_{\mathcal{G}}$ factorises involve the energy of the sites of $\psi_{\mathcal{G}}$ shifted by $-y_{12}$ (white site) and $+y_{12}$ (black site) on the sides of the cut, with a contribution from both possible shifts. Interestingly, the terms in the right-hand-side of (95) have a directed energy flow along the erased edge.

A second class of triangulations can be obtained via the adjoint surface. First, the adjoint surface is determined by the combination of markings which cover completely the graph [46]

$$\underset{x_1 \ y_{12} \ x_2}{\bullet\!\times\!\!*\!\!\times\!\bullet} = \left\{ \underset{x_1 \ y_{12} \ x_2}{\bullet\!\times\!\!|\!\!-\!\!\bullet} \ , \ \underset{x_1 \ y_{12} \ x_2}{\bullet\!-\!\!\times\!\!\times\!\bullet} \right\} = \left\{ \underset{x_1 \ y_{12} \ x_2}{\bullet\!-\!\!|\!\!\times\!\bullet} \ , \ \underset{x_1 \ y_{12} \ x_2}{\bullet\!\times\!\!\times\!\!-\!\bullet} \right\}. \tag{96}$$

The markings in the curly brackets identify the points $\mathcal{Z}_{\mathrm{A}}^{I} := \epsilon^{IJK}\mathcal{W}_J^{(\mathfrak{g}_1)}\widehat{\mathcal{W}}_K^{(\mathfrak{g}_1)} \sim \mathbf{x}_2$ and $\mathcal{Z}_{\mathrm{B}}^{I} := \epsilon^{IJK}\mathcal{W}_J^{(\mathfrak{g}_2)}\widehat{\mathcal{W}}_K^{(\mathfrak{g}_2)} \sim \mathbf{x}_1$. The adjoint surface is then a line characterised by the co-vector $\mathcal{C}_I = \epsilon_{IJK}\mathcal{Z}_{\mathrm{A}}^{J}\mathcal{Z}_{\mathrm{B}}^{K}$. It is possible to triangulate our quadrilateral either via $\mathcal{Z}_A$ or $\mathcal{Z}_B$. From (94), we respectively obtain

$$\omega(\mathcal{Y}, \mathcal{O}_{\mathcal{G}}) = \left[\frac{1}{q_{\mathfrak{m}_{\mathfrak{g}_2}}} + \frac{1}{q_{\widehat{\mathfrak{m}}_{\mathfrak{g}_2}}}\right]\frac{\langle \mathcal{Y}d^2\mathcal{Y}\rangle}{q_{\mathfrak{m}_{\mathfrak{g}_1}}q_{\widehat{\mathfrak{m}}_{\mathfrak{g}_1}}} = \frac{1}{x_1^2 - y_{12}^2}\left[\frac{1}{x_2 + y_{12}} - \frac{1}{x_2 - y_{12}}\right]\frac{dx_1 \wedge dy_{12} \wedge dx_2}{\mathrm{Vol}\{GL(1)\}}, \tag{97}$$



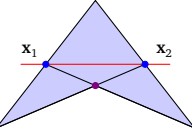 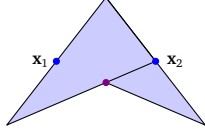 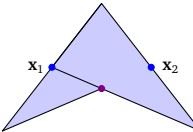

Figure 12: Triangulations of the $\mathcal{O}_{\mathcal{G}}$ associated to the two-site line graph via its adjoint surface. On the left: The adjoint surface for $\mathcal{O}_{\mathcal{G}}$ is represented via the red line. The two intersections between the hyperplanes containing non-adjacent facets are $\mathcal{Z}_{\mathcal{A}} \sim \mathbf{x}_2$ and $\mathcal{Z}_{\mathcal{B}} \sim \mathbf{x}_1$. The center and right pictures are the triangualtions using the two special points on the adjoint surface and respectively represent (97) and (98).

and

$$\omega(\mathcal{Y}, \mathcal{O}_{\mathcal{G}}) = \left[ \frac{1}{q_{\mathfrak{m}_{\mathfrak{g}_1}}} + \frac{1}{q_{\widehat{\mathfrak{m}}_{\mathfrak{g}_1}}} \right] \frac{\langle \mathcal{Y} d^2 \mathcal{Y} \rangle}{q_{\mathfrak{m}_{\mathfrak{g}_2}} q_{\widehat{\mathfrak{m}}_{\mathfrak{g}_2}}} = \frac{1}{x_2^2 - y_{12}^2} \left[ \frac{1}{x_1 + y_{12}} - \frac{1}{x_1 - y_{12}} \right] \frac{dx_1 \wedge dy_{12} \wedge dx_2}{\mathrm{Vol}\{GL(1)\}}. \tag{98}$$

What about the (holomorphic) cutting rules? For this specific graph, the standard and holomorphic cutting rules turn out to coincide. Notice that the adjoint surface is identified by the linear polynomial $\mathfrak{n}_1(\mathcal{Y}) = \mathcal{Y} \cdot \mathcal{C} = -2y_{12}$ which constitutes the numerator of the canonical form. It is possible to triangulate $\mathcal{O}_{\mathcal{G}}$ requiring that the adjoint is an actual boundary for all the simplices, *i.e.*

$$(\mathcal{Z}_1 \mathcal{Z}_2 \mathcal{Z}_4 \mathcal{Z}_3) = (\mathcal{Z}_1 \mathbf{x}_1 \mathbf{x}_2) + (\mathbf{x}_1 \mathcal{Z}_2 \mathbf{x}_2) + (\mathbf{x}_1 \mathcal{Z}_3 \mathbf{x}_2) + (\mathbf{x}_1 \mathbf{x}_2 \mathcal{Z}_4),$$

where each round bracket contains a sequence of vertices which represent the polygon (a quadrilateral on the left-hand-side and triangles on the right-hand one) and their order indicates the orientation. The canonical form of $\mathcal{O}_{\mathcal{G}}$ can then be written as

$$\omega(\mathcal{Y}, \mathcal{O}_{\mathcal{G}}) = \left[ \frac{\langle 1 \mathbf{x}_1 \mathbf{x}_2 \rangle^2}{\langle \mathcal{Y} 1 \mathbf{x}_1 \rangle \langle \mathcal{Y} \mathbf{x}_1 \mathbf{x}_2 \rangle \langle \mathcal{Y} \mathbf{x}_2 1 \rangle} + \frac{\langle \mathbf{x}_1 2 \mathbf{x}_2 \rangle^2}{\langle \mathcal{Y} \mathbf{x}_1 2 \rangle \langle \mathcal{Y} 2 \mathbf{x}_2 \rangle \langle \mathcal{Y} \mathbf{x}_2 \mathbf{x}_1 \rangle} \right. \\ \left. + \frac{\langle \mathbf{x}_1 3 \mathbf{x}_2 \rangle^2}{\langle \mathcal{Y} \mathbf{x}_1 3 \rangle \langle \mathcal{Y} 3 \mathbf{x}_2 \rangle \langle \mathcal{Y} \mathbf{x}_2 \mathbf{x}_1 \rangle} + \frac{\langle \mathbf{x}_1 \mathbf{x}_2 4 \rangle^2}{\langle \mathcal{Y} \mathbf{x}_1 \mathbf{x}_2 \rangle \langle \mathcal{Y} \mathbf{x}_2 4 \rangle \langle \mathcal{Y} 4 \mathbf{x}_1 \rangle} \right] \langle \mathcal{Y} d^2 \mathcal{Y} \rangle, \tag{99}$$

which is the r.h.s. of the cutting rules of [32, 35, 37].

As a final comment, it is worth noticing that our quadrilateral $\mathcal{O}_{\mathcal{G}}$ has further triangulations using the two special points which identify its adjoint surface:

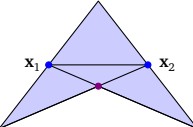 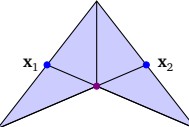

They correspond to two new four-term decompositions of the canonical form of $\mathcal{O}_{\mathcal{G}}$, namely

$$\omega(\mathcal{Y}, \mathcal{O}_{\mathcal{G}}) = \left[ \frac{\langle 1 \mathbf{x}_1 \mathbf{x}_2 \rangle^2}{\langle \mathcal{Y} 1 \mathbf{x}_1 \rangle \langle \mathcal{Y} \mathbf{x}_1 \mathbf{x}_2 \rangle \langle \mathcal{Y} \mathbf{x}_2 1 \rangle} + \frac{\langle \mathbf{x}_1 4 \mathbf{x}_2 \rangle^2}{\langle \mathcal{Y} \mathbf{x}_1 4 \rangle \langle \mathcal{Y} 4 \mathbf{x}_2 \rangle \langle \mathcal{Y} \mathbf{x}_2 \mathbf{x}_1 \rangle} \right. \\ \left. + \frac{\langle \mathbf{x}_1 2 4 \rangle^2}{\langle \mathcal{Y} \mathbf{x}_1 2 \rangle \langle \mathcal{Y} 2 4 \rangle \langle \mathcal{Y} 4 \mathbf{x}_1 \rangle} + \frac{\langle 4 3 \mathbf{x}_2 \rangle^2}{\langle \mathcal{Y} 4 3 \rangle \langle \mathcal{Y} 3 \mathbf{x}_2 \rangle \langle \mathcal{Y} \mathbf{x}_2 4 \rangle} \right] \langle \mathcal{Y} d^2 \mathcal{Y} \rangle, \tag{100}$$

and

$$\omega(\mathcal{Y}, \mathcal{O}_{\mathcal{G}}) = \left[ \frac{\langle 1 \mathbf{x}_1 4 \rangle^2}{\langle \mathcal{Y} 1 \mathbf{x}_1 \rangle \langle \mathcal{Y} \mathbf{x}_1 4 \rangle \langle \mathcal{Y} 4 1 \rangle} + \frac{\langle 4 \mathbf{x}_2 1 \rangle^2}{\langle \mathcal{Y} 4 \mathbf{x}_2 \rangle \langle \mathcal{Y} \mathbf{x}_2 1 \rangle \langle \mathcal{Y} 1 4 \rangle} \right. \\ \left. + \frac{\langle \mathbf{x}_1 2 4 \rangle^2}{\langle \mathcal{Y} \mathbf{x}_1 2 \rangle \langle \mathcal{Y} 2 4 \rangle \langle \mathcal{Y} 4 \mathbf{x}_1 \rangle} + \frac{\langle 4 3 \mathbf{x}_2 \rangle^2}{\langle \mathcal{Y} 4 3 \rangle \langle \mathcal{Y} 3 \mathbf{x}_2 \rangle \langle \mathcal{Y} \mathbf{x}_2 4 \rangle} \right] \langle \mathcal{Y} d^2 \mathcal{Y} \rangle, \tag{101}$$

which differ from each other in how the convex quadrilateral $(\mathcal{Z}_1 \mathbf{x}_1 \mathcal{Z}_4 \mathbf{x}_2)$ is triangulated.

**The optical polytope and the three-site line graph.** Let us now consider the optical polytope $\mathcal{O}_{\mathcal{G}} \in \mathbb{P}^4$ associated to the three-site line graph, which is characterised by 8 vertices and 10 facets. This is the simplest case in which the standard and holomorphic cutting rules differ from each other, as depicted in Figures 2 and 3, and both show spurious singularities, $\{y_{12} = 0, y_{23} = 0, -y_{12} + x_2 + y_{23} = 0\}$ and $\{y_{12} = 0, y_{23} = 0, -y_{12} + x_2 + y_{23} = 0, y_{12} + x_2 - y_{23} = 0\}$ respectively – note that the last set contains the same spurious singularities as the first set, plus an additional one. In the optical polytope picture, the spurious poles correspond to hyperplanes which are not actually boundaries of the geometry.

Let us consider the hyperplanes $\mathcal{W}^{(y_{12})} := \tilde{\mathbf{y}}_{12}$, $\mathcal{W}^{(y_{23})} := \tilde{\mathbf{y}}_{23}$, $\mathcal{W}^{(2\mp)} := -\tilde{\mathbf{y}}_{12} + \tilde{\mathbf{x}}_2 + \tilde{\mathbf{y}}_{23}$ and $\mathcal{W}^{(2\pm)} := +\tilde{\mathbf{y}}_{12} + \tilde{\mathbf{x}}_2 - \tilde{\mathbf{y}}_{23}$, where the indices $_{ij}$ label the edge between the $i$-th and $j$-th site. They turn out to intersect $\mathcal{O}_{\mathcal{G}}$ in its interior and each of them contains a subset of the vertices of $\mathcal{O}_{\mathcal{G}}$ and intersects the boundaries of $\mathcal{O}_{\mathcal{G}}$ in at least one of the special points $\mathbf{x}_1$ and $\mathbf{x}_3$. Concretely

$$
\begin{aligned}
\mathcal{W}^{(y_{12})} &: \{\mathbf{x}_1, \mathcal{Z}_{(23)}^{(1)}, \mathcal{Z}_{(23)}^{(2)}, \mathcal{Z}_{(23)}^{(3)}, \mathcal{Z}_{(23)}^{(4)}, \mathbf{x}_3\}, &
\mathcal{W}^{(y_{23})} &: \{\mathbf{x}_1, \mathcal{Z}_{(12)}^{(1)}, \mathcal{Z}_{(12)}^{(2)}, \mathcal{Z}_{(12)}^{(3)}, \mathcal{Z}_{(12)}^{(4)}, \mathbf{x}_3\}, \\
\mathcal{W}^{(2\mp)} &: \{\mathbf{x}_1, \mathcal{Z}_{(12)}^{(3)}, \mathcal{Z}_{(12)}^{(4)}, \mathcal{Z}_{(23)}^{(1)}, \mathcal{Z}_{(23)}^{(3)}, \mathbf{x}_3\}, &
\mathcal{W}^{(2\pm)} &: \{\mathbf{x}_1, \mathcal{Z}_{(12)}^{(1)}, \mathcal{Z}_{(12)}^{(2)}, \mathcal{Z}_{(23)}^{(2)}, \mathcal{Z}_{(23)}^{(4)}, \mathbf{x}_3\}.
\end{aligned}
\tag{102}
$$

We can triangulate $\mathcal{O}_{\mathcal{G}}$ using the special points $\{\mathbf{x}_1, \mathbf{x}_3\}$ as vertices, and having the hyperplanes in (102) as spurious boundaries. One of the possible triangulations is given by the collection of simplices in $\mathbb{P}^4$ $\{\mathcal{P}_{(j)}, j = 1, \ldots, 8\}$ defined as convex hull of the vertices

$$
\begin{aligned}
\mathcal{P}^{(1)} &: \{\mathbf{x}_1, \mathcal{Z}_{(12)}^{(3)}, \mathcal{Z}_{(23)}^{(1)}, \mathcal{Z}_{(23)}^{(2)}, \mathcal{Z}_{(23)}^{(3)}\}, &
\mathcal{P}^{(2)} &: \{\mathbf{x}_1, \mathcal{Z}_{(12)}^{(4)}, \mathcal{Z}_{(23)}^{(1)}, \mathcal{Z}_{(23)}^{(2)}, \mathcal{Z}_{(23)}^{(3)}\}, \\
\mathcal{P}^{(3)} &: \{\mathbf{x}_1, \mathcal{Z}_{(12)}^{(1)}, \mathcal{Z}_{(23)}^{(1)}, \mathcal{Z}_{(23)}^{(2)}, \mathcal{Z}_{(23)}^{(3)}\}, &
\mathcal{P}^{(4)} &: \{\mathbf{x}_1, \mathcal{Z}_{(12)}^{(2)}, \mathcal{Z}_{(23)}^{(1)}, \mathcal{Z}_{(23)}^{(2)}, \mathcal{Z}_{(23)}^{(3)}\}, \\
\mathcal{P}^{(5)} &: \{\mathcal{Z}_{(12)}^{(2)}, \mathcal{Z}_{(12)}^{(3)}, \mathcal{Z}_{(12)}^{(4)}, \mathcal{Z}_{(23)}^{(3)}, \mathbf{x}_3\}, &
\mathcal{P}^{(6)} &: \{\mathbf{x}_1, \mathcal{Z}_{(23)}^{(4)}, \mathcal{Z}_{(23)}^{(1)}, \mathcal{Z}_{(23)}^{(2)}, \mathcal{Z}_{(23)}^{(3)}\}, \\
\mathcal{P}^{(7)} &: \{\mathbf{x}_1, \mathcal{Z}_{(12)}^{(1)}, \mathcal{Z}_{(23)}^{(1)}, \mathcal{Z}_{(23)}^{(2)}, \mathcal{Z}_{(23)}^{(3)}\}, &
\mathcal{P}^{(8)} &: \{\mathbf{x}_1, \mathcal{Z}_{(12)}^{(2)}, \mathcal{Z}_{(23)}^{(1)}, \mathcal{Z}_{(23)}^{(2)}, \mathcal{Z}_{(23)}^{(3)}\},
\end{aligned}
\tag{103}
$$

which makes use of the hyperplanes $\{\mathcal{W}^{(y_{12})}, \mathcal{W}^{(y_{23})}, \mathcal{W}^{(2\mp)}\}$ and decomposes the canonical form of $\omega(\mathcal{Y}, \mathcal{O}_{\mathcal{G}})$ into the cutting rules in Figure 2. Another triangulation through the special points $\{\mathbf{x}_1, \mathbf{x}_3\}$ makes use of all the four hyperplanes (102) and decomposes $\mathcal{O}_{\mathcal{G}}$ in 16 simplices: this triangulation decomposes the canonical form of $\mathcal{O}_{\mathcal{G}}$ into the holomorphic cutting rules depicted in Figure 3.

Let us now turn to the class of triangulations which do not introduce spurious boundaries. Let us list here some of the subspaces of the adjoint surface of $\mathcal{O}_{\mathcal{G}}$, namely the ones identified by the markings $\{\mathfrak{m}_{\mathfrak{g}_1}, \widehat{\mathfrak{m}}_{\mathfrak{g}_1}, \mathfrak{m}_{\mathfrak{g}_3}, \widehat{\mathfrak{m}}_{\mathfrak{g}_3}, \}$ and $\{\mathfrak{m}_{\mathfrak{g}_2}, \widehat{\mathfrak{m}}_{\mathfrak{g}_2}\}$:[22]

$$
\begin{aligned}
&\text{(figure)} \\
&\text{(figure)}
\end{aligned}
\tag{104}
$$

_________________

[22]Note that the equalities in (104) just mean that the vertex configuration on the left-hand side, *i.e.* with no vertices, can be obtained as on the two right-hand sides. However, they identify different subspaces of the adjoint surface: the first line identifies a subspace of codimension-4, while the last line a subspace of codimension-2.



The canonical form triangulation of $\mathcal{O}_{\mathcal{G}}$ via the subspace identified by the first two lines above is given by

$$\omega(\mathcal{Y}, \mathcal{O}_{\mathcal{G}}) = \left[ \frac{1}{q_{\mathrm{m}_{\mathfrak{g}_2}}} + \frac{1}{q_{\widehat{\mathrm{m}}_{\mathfrak{g}_2}}} + \frac{1}{q_{\mathrm{m}_{\mathfrak{g}_{12}}}} + \frac{1}{q_{\widehat{\mathrm{m}}_{\mathfrak{g}_{12}}}} + \frac{1}{q_{\mathrm{m}_{\mathfrak{g}_{23}}}} + \frac{1}{q_{\widehat{\mathrm{m}}_{\mathfrak{g}_{23}}}} \right] \frac{\langle \mathcal{Y} d^4 \mathcal{Y} \rangle}{q_{\mathrm{m}_{\mathfrak{g}_1}} q_{\widehat{\mathrm{m}}_{\mathfrak{g}_1}} q_{\mathrm{m}_{\mathfrak{g}_3}} q_{\widehat{\mathrm{m}}_{\mathfrak{g}_3}}} \,, \qquad (105)$$

which is equivalent to what we would obtain were we to apply the tree-level recursion relation in [24]. The subspace identified by the second set of markings in (104) instead provides the following canonical form triangulation

$$\omega(\mathcal{Y}, \mathcal{O}_{\mathcal{G}}) = \left\{ \frac{1}{q_{\mathrm{m}_{\mathfrak{g}_{12}}} q_{\mathrm{m}_{\mathfrak{g}_{23}}}} \left[ \frac{1}{q_{\widehat{\mathrm{m}}_{\mathfrak{g}_1}}} + \frac{1}{q_{\widehat{\mathrm{m}}_{\mathfrak{g}_3}}} \right] + \frac{1}{q_{\widehat{\mathrm{m}}_{\mathfrak{g}_{12}}} q_{\widehat{\mathrm{m}}_{\mathfrak{g}_{23}}}} \left[ \frac{1}{q_{\mathrm{m}_{\mathfrak{g}_1}}} + \frac{1}{q_{\mathrm{m}_{\mathfrak{g}_3}}} \right] + \frac{1}{q_{\mathrm{m}_{\mathfrak{g}_{12}}} q_{\widehat{\mathrm{m}}_{\mathfrak{g}_{23}}}} \left[ \frac{1}{q_{\mathrm{m}_{\mathfrak{g}_1}}} + \frac{1}{q_{\widehat{\mathrm{m}}_{\mathfrak{g}_3}}} \right] \right.$$

$$+ \frac{1}{q_{\widehat{\mathrm{m}}_{\mathfrak{g}_{12}}} q_{\mathrm{m}_{\mathfrak{g}_{23}}}} \left[ \frac{1}{q_{\widehat{\mathrm{m}}_{\mathfrak{g}_1}}} + \frac{1}{q_{\mathrm{m}_{\mathfrak{g}_3}}} \right] + \left[ \frac{1}{q_{\mathrm{m}_{\mathfrak{g}_{12}}}} + \frac{1}{q_{\widehat{\mathrm{m}}_{\mathfrak{g}_{12}}}} + \frac{1}{q_{\mathrm{m}_{\mathfrak{g}_{23}}}} + \frac{1}{q_{\widehat{\mathrm{m}}_{\mathfrak{g}_{23}}}} \right]$$

$$\left. \times \left[ \frac{1}{q_{\mathrm{m}_{\mathfrak{g}_1}} q_{\mathrm{m}_{\mathfrak{g}_3}}} + \frac{1}{q_{\mathrm{m}_{\mathfrak{g}_1}} q_{\widehat{\mathrm{m}}_{\mathfrak{g}_3}}} + \frac{1}{q_{\widehat{\mathrm{m}}_{\mathfrak{g}_1}} q_{\mathrm{m}_{\mathfrak{g}_3}}} + \frac{1}{q_{\widehat{\mathrm{m}}_{\mathfrak{g}_1}} q_{\widehat{\mathrm{m}}_{\mathfrak{g}_3}}} \right] \right\} \cdot \frac{\langle \mathcal{Y} d^4 \mathcal{Y} \rangle}{q_{\mathrm{m}_{\mathfrak{g}_2}} q_{\widehat{\mathrm{m}}_{\mathfrak{g}_2}}} \,. \qquad (106)$$

**From the cutting rules to unitarity and the wavefunction.** Let us now briefly comment on the following question. Imagine that we are given $\Delta\psi_{\mathcal{G}}$ via any of the cutting rules, which information about the wavefunction and the unitarity of the processes it describes can we infer?

As pointed out in [37], in a non-unitary theory it is still possible to arrange a quantity $\psi'_{\mathcal{G}}$ such that $\psi_{\mathcal{G}} + \psi'_{\mathcal{G}}$ satisfies the (holomorphic) cutting rules as well as any other coming from $\mathcal{O}_{\mathcal{G}}$. However, in order for the theory to be unitary, it is necessary that $\psi'_{\mathcal{G}}$ can be identified with $\psi^\dagger_{\mathcal{G}}(-x_s, y_e)$. Said differently, the peculiarity of a unitary theory is that the quantity which, combined with $\psi_{\mathcal{G}}$, gives rise to the cuts, is $\psi^\dagger_{\mathcal{G}}(-x_s, y_e)$. The "cutting rules" can be derived algebraically via partial fraction identities in the integrand and, thus, they do not require unitarity. Unitary relates $\psi'_{\mathfrak{g}}$ to $\psi^\dagger_{\mathfrak{g}}$ for all $\mathfrak{g} \subseteq \mathcal{G}$.

Finally, note that $\Delta\psi_{\mathcal{G}}$ has singularities both of the Bunch-Davies and folded types. As such, irrespectively of which cutting rules are used to compute $\Delta\psi_{\mathcal{G}}$, it is necessary to impose the absence of folded singularities to extract the Bunch-Davies wavefunction – in the optical polytope picture, this is equivalent to choosing the polytope subdivision which separates the vertices $\{\mathcal{Z}^{(1)}_e, e \in \mathcal{E}\}$ from the vertices $\{\mathcal{Z}^{(4)}_e, e \in \mathcal{E}\}$.

## 5.4 From the universal integrand to the integrated cutting rules

The triangulations of the optical polytope provide different decomposition of the universal wavefunction integrand for $\Delta\psi_{\mathcal{G}}$. In some cases they precisely represent the decomposition of the actual $\Delta\tilde{\psi}_{\mathcal{G}}$, e.g. when the interactions are conformal as for $\phi^3$ interactions in $d = 5$ and $\phi^4$ interactions in $d = 3$.

Also, as they provide a decomposition of the universal wavefunction integrand, they induce a decomposition of the integrals, which can be written schematically as

$$\prod_{s \in \mathcal{V}} \left[ \int_{X_s}^{+\infty} dx_s \, \tilde{\mu}(x_s - X_s) \right] \Omega(\mathcal{Y}, \mathcal{O}_{\mathcal{G}}) = \sum_{\{\Sigma_{\mathcal{G}}\}} \left[ \int_{X_s}^{+\infty} dx_s \, \tilde{\mu}(x_s - X_s) \right] \Omega(\mathcal{Y}, \Sigma_{\mathcal{G}}), \qquad (107)$$

where the integration is over all the weights associated to the sites of the graph $\mathcal{G}$ with measure $\tilde{\mu}(x_s - X_s)$, encoding the effects of the expanding background and of the specific states

involved.[23] The dependence of the site weights $\{x_s, s \in \mathcal{V}\}$ is encoded in $\mathcal{Y}$, forming, together with the edge weights $\{y_e, e \in \mathcal{E}\}$ the local homogeneous coordinates for the projective space $\mathbb{P}^{n_s+n_e-1}$ where $\mathcal{O}_{\mathcal{G}}$ lives. The sum runs over the elements of the set $\{\Sigma_{\mathcal{G}}\}$ which identifies a triangulation into the simplices $\Sigma_{\mathcal{G}}$. However, what happens at the level of the integrated functions?

Firstly, it is important to keep in mind that in principle the integrals over the site weights could individually be divergent as $x_s \longrightarrow +\infty$, which is a manifestation of certain infra-red divergences [58]. One example is given by the wavefunction coefficients associated to a single site for a cubic interaction in $(1+3)$-dimensional de Sitter space $(dS_{1+3})$ with measure $\mu = \ell_1$

$$\tilde{\psi}_{\mathcal{G}}(X) = \ell_1 \int_X^{+\infty} dx\, \psi_{\mathcal{G}}(x) = \ell_1 \int_X^{+\infty} \frac{dx}{x}, \tag{108}$$

which has a logarithmic divergence at infinity – $\ell_1$ is the chacteristic length of $dS_{1+3}$. As we will see, such logarithmic singularities cancel in (107) when present. In any case, despite possible cancellations, these divergences might appear in a decomposition we consider and they need to be taken care of by either utilizing the usual hard cut-off or via analytic regularisation – for the latter, see [58]. Said differently, a given decomposition (107) can introduce not only spurious singularities at a finite location in kinematic space, but also spurious/more severe infra-red singularities.

For measures of the type $\tilde{\mu}(x_s - X_s) \sim (x_s - X_s)^{\alpha_s - 1}$ ($\alpha_s \in \mathbb{Z}_+$), the geometry fixes the symbols for each integral – see [24,59]. A detailed account of the integrations is outside the scope of the present work, as it involves both taking care of the appearance of infra-red divergences and generalising the treatment for non-integer $\alpha_s$'s. Here, we will inspect in a simple case the map between the integrand and integrated structures as different triangulations of the relevant optical polytope are taken, to illustrate: 1. that spurious singularities in the integrand are mapped into spurious singularities of the integrated function and 2. the appearance of spurious infra-red singularities.

The simplest example is given by the optical polytope associated to the two-site line graph. It lives in $\mathbb{P}^2$ and its canonical function is given by (61), which we rewrite here for convenience:

$$\Omega(\mathcal{Y}, \mathcal{O}_{\mathcal{G}}) = \frac{\langle \mathcal{Y}AB \rangle}{(\mathcal{Y} \cdot \mathcal{W}^{(\mathfrak{g}_1)})(\mathcal{Y} \cdot \mathcal{W}^{(\mathfrak{g}_2)})(\mathcal{Y} \cdot \widehat{\mathcal{W}}^{(\mathfrak{g}_1)})(\mathcal{Y} \cdot \widehat{\mathcal{W}}^{(\mathfrak{g}_2)})} = \frac{-2y_{12}}{(x_1^2 - y_{12}^2)(x_2^2 - y_{12}^2)}. \tag{109}$$

Let us integrate over the site weights with the measure $\tilde{\mu} = \ell_1^2$. The two integrals turn out to factorise completely and they are well-behaved as $x_j \longrightarrow +\infty$ ($j = 1, 2$). The canonical function (109) integrates to a simple product of logarithms

$$\widetilde{\Omega}(\tilde{\mathcal{Y}}, \mathcal{O}_{\mathcal{G}}\}) = \ell_1^2 \prod_{j=1}^2 \left[ \int_{X_j}^{+\infty} dx_j \right] \Omega(\mathcal{Y}, \mathcal{O}_{\mathcal{G}}) = -\frac{\ell_1^2}{2y_{12}} \log \frac{X_1 + y_{12}}{X_1 - y_{12}} \times \log \frac{X_2 + y_{12}}{X_2 - y_{12}}, \tag{110}$$

where $\widetilde{\Omega}(\tilde{\mathcal{Y}}, \mathcal{O}_{\mathcal{G}}\}) = \Delta\tilde{\psi}_{\mathcal{G}}$. Note that despite (110) shows a possible singularity in $y_{12} = 0$, such singularity not only is spurious but the line $y_{12} = 0$ is still a zero:

$$\widetilde{\Omega}(\tilde{\mathcal{Y}}, \mathcal{O}_{\mathcal{G}}\}) \overset{y_{12} \to 0}{\sim} \ell_1^2 \frac{-2y_{12}}{X_1 X_2} [1 + \dots]. \tag{111}$$

The singularities and zeroes of $\Omega(\mathcal{Y}, \mathcal{O}_{\mathcal{G}})$ turn out to be respectively mapped into singularities and zeroes of $\widetilde{\Omega}(\tilde{\mathcal{Y}}, \mathcal{O}_{\mathcal{G}})$.

---

[23]Recall that for conformally coupled scalars $\tilde{\mu}(z)$ coincides with the function $\tilde{\lambda}(z)$ in (28), while when other states are involved there are extra contributions due to the factors of $(-\eta_s)^{1/2-\nu}$ associated to each bulk-to-boundary and bulk-to-bulk propagator at the site $s$. See [31, 53].

Let us now consider the triangulation of the optical polytope which returns the left-hand-side of the cosmological optical theorem. In this case

$$\widetilde{\Omega}(\tilde{\mathcal{Y}}, \mathcal{O}_{\mathcal{G}}) = \prod_{j=1}^{2} \left[ \int_{X_j}^{+\infty} dx_j \right] \left[ \Omega(\mathcal{Y}, \mathcal{P}_{\mathcal{G}}) + \Omega(\mathcal{Y}, \mathcal{P}_{\mathcal{G}}^{\dagger}(-1)) \right]. \tag{112}$$

The symbols for the first integral were extracted in [24,59] and the constant ambiguity fixed in [59]:[24]

$$\prod_{j=1}^{2} \left[ \int_{X_j}^{+\infty} dx_j \right] \Omega(\mathcal{Y}, \mathcal{P}_{\mathcal{G}}) = \frac{\ell_1^2}{2y_{12}} \left[ \mathrm{Li}_2\left( \frac{X_1 - y_{12}}{X_1 + X_2} \right) + \mathrm{Li}_2\left( \frac{X_2 - y_{12}}{X_1 + X_2} \right) \right.$$
$$\left. + \log\left( \frac{X_1 + y_{12}}{X_1 + X_2} \right) \log\left( \frac{X_2 + y_{12}}{X_1 + X_2} \right) - \frac{\pi^2}{6} \right], \tag{113}$$

which shows logarithmic branch points as $X_1 + y_{12} \longrightarrow 0$, $X_2 + y_{12} \longrightarrow 0$ and $X_1 + X_2 \longrightarrow 0$ as expected from the analytic structure of $\Omega(\mathcal{Y}, \mathcal{P}_{\mathcal{G}})$. The second integral in (112) can be obtained from (113) via $X_j \longrightarrow -X_j$ ($j = 1, 2$):

$$\prod_{j=1}^{2} \left[ \int_{X_j}^{+\infty} dx_j \right] \Omega(\mathcal{Y}, \mathcal{P}_{\mathcal{G}}^{\dagger}(-1)) = \frac{\ell_1^2}{2y_{12}} \left[ \mathrm{Li}_2\left( \frac{X_1 + y_{12}}{X_1 + X_2} \right) + \mathrm{Li}_2\left( \frac{X_2 + y_{12}}{X_1 + X_2} \right) \right.$$
$$\left. + \log\left( \frac{X_1 - y_{12}}{X_1 + X_2} \right) \log\left( \frac{X_2 - y_{12}}{X_1 + X_2} \right) - \frac{\pi^2}{6} \right], \tag{114}$$

which has the expected logarithmic branch points as $X_1 - y_{12} \longrightarrow 0$, $X_2 - y_{12} \longrightarrow 0$ and $X_1 + X_2 \longrightarrow 0$ as expected from the analytic structure of $\Omega(\mathcal{Y}, \mathcal{P}_{\mathcal{G}}^{\dagger}(-1))$. Let $z_1$ and $z_2$ be respectively the arguments of the dilogarithms in (113). Then, the arguments in the dilogarithms in (114) can be respectively written as $1 - z_2$ and $1 - z_1$. Upon summation of (113) and (114), the right-hand-side of (112) can be rewritten as[25]

$$\widetilde{\Omega}(\tilde{\mathcal{Y}}, \mathcal{O}_{\mathcal{G}}) = \frac{\ell_1^2}{2y_{12}} \left[ -\log\left( \frac{X_1 - y_{12}}{X_1 + X_2} \right) \times \log\left( \frac{X_2 + y_{12}}{X_1 + X_2} \right) - \log\left( \frac{X_2 - y_{12}}{X_1 + X_2} \right) \times \log\left( \frac{X_1 + y_{12}}{X_1 + X_2} \right) \right.$$
$$\left. + \log\left( \frac{X_1 - y_{12}}{X_1 + X_2} \right) \times \log\left( \frac{X_2 - y_{12}}{X_1 + X_2} \right) + \log\left( \frac{X_1 + y_{12}}{X_1 + X_2} \right) \times \log\left( \frac{X_2 + y_{12}}{X_1 + X_2} \right) \right], \tag{115}$$

which can be straightforwardly recast into (110).

A similar structure appears when we consider the other trangulation via the line passing through the non-adjacent vertices $Z^{(1)}$ and $Z^{(4)}$, which is diagrammatically expressed via (95):

$$\widetilde{\Omega}(\tilde{\mathcal{Y}}, \mathcal{O}_{\mathcal{G}}) = \prod_{j=1}^{2} \left[ \int_{X_j}^{+\infty} dx_j \right] \left[ \Omega(\mathcal{Y}, \Sigma_{124}) + \Omega(\mathcal{Y}, \Sigma_{431}) \right], \tag{116}$$

---

[24]Note that this result appears also as contribution to the in-in correlator from part of the integration path in [23].
[25]Here, just the following identity between dilogarithms is used:

$$\mathrm{Li}_2(z) + \mathrm{Li}_2(1-z) = \frac{\pi^2}{6} - \log z \times \log(1-z).$$

where $\Sigma_{ijk}$ represents the simplex identified by the vertices $\mathcal{Z}^{(i)}$, $\mathcal{Z}^{(j)}$ and $\mathcal{Z}^{(k)}$. Concretely:

$$\prod_{j=1}^{2}\left[\int_{X_j}^{+\infty} dx_j\right]\Omega(\mathcal{Y},\Sigma_{124}) = \frac{\ell_1^2}{2y_{12}}\left[\text{Li}_2\left(\frac{X_1+y_{12}}{X_1-X_2}\right)+\text{Li}_2\left(-\frac{X_2-y_{12}}{X_1-X_2}\right)\right.$$
$$\left.+\log\left(\frac{X_1-y_{12}}{X_1-X_2}\right)\log\left(-\frac{X_2+y_{12}}{X_1-X_2}\right)-\frac{\pi^2}{6}\right], \quad (117)$$

and

$$\prod_{j=1}^{2}\left[\int_{X_j}^{+\infty} dx_j\right]\Omega(\mathcal{Y},\Sigma_{431}) = \frac{\ell_1^2}{2y_{12}}\left[\text{Li}_2\left(\frac{X_1-y_{12}}{X_1-X_2}\right)+\text{Li}_2\left(-\frac{X_2+y_{12}}{X_1-X_2}\right)\right.$$
$$\left.+\log\left(\frac{X_1+y_{12}}{X_1-X_2}\right)\log\left(-\frac{X_2-y_{12}}{X_1-X_2}\right)-\frac{\pi^2}{6}\right]. \quad (118)$$

Let us analyse the triangulations of the optical polytope via points in its adjoint surface. These two triangulations are given in (97) and (98). Both of them are two-term triangulations for which the two integrations factorise. Let us consider for concreteness (97):

$$\widetilde{\Omega}(\widetilde{\mathcal{Y}},\mathcal{O}_{\mathcal{G}}) = \prod_{j=1}^{2}\left[\int_{X_j}^{+\infty} dx_j\right]\left[\Omega(\mathcal{Y},\Sigma_{12\mathbf{x}_2})+\Omega(\mathcal{Y},\Sigma_{43\mathbf{x}_2})\right]$$
$$= \ell_1^2\int_{X_1}^{+\infty}\frac{dx_1}{x_1^2-y_{12}^2}\int_{X_2}^{+\infty}\frac{dx_2}{x_2+y_{12}}-\ell_1^2\int_{X_1}^{+\infty}\frac{dx_1}{x_1^2-y_{12}^2}\int_{X_2}^{+\infty}\frac{dx_2}{x_2-y_{12}}. \quad (119)$$

Note that in both terms, the integration in $x_2$ shows a logarithmic branch point as the region around infinity is approached, while the integration in $x_1$ turns to be well-defined in this region. It is possible to regulate such an integration via a hard cut-off or via analytic regularisation. Let us consider the latter [58]. Then

$$\widetilde{\Omega}^{(\epsilon)}(\widetilde{\mathcal{Y}},\mathcal{O}_{\mathcal{G}}) = \ell_1^{2-\epsilon}\int_{X_1}^{+\infty}\frac{dx_1}{x_1^2-y_{12}^2}\left[\int_{X_2}^{+\infty} dx_2\frac{(x_2-X_2)^{-\epsilon}}{x_2+y_{12}}-\int_{X_2}^{+\infty} dx_2\frac{(x_2-X_2)^{-\epsilon}}{x_2-y_{12}}\right]$$
$$= \frac{\ell_1^{2-\epsilon}}{2y_{12}}\Gamma(1-\epsilon)\Gamma(-\epsilon)\log\left(\frac{X_1+y_{12}}{X_1-y_{12}}\right)\left[(X_2+y_{12})^{-\epsilon}-(X_2-y_{12})^{-\epsilon}\right], \quad (120)$$

$\epsilon$ being the regulator. In analytic regularisation, the logarithmic singularity in the infra-red is mapped into a pole, which manifests in the $\Gamma$-functions. While in the previous cases the structure of the triangulations reflects into the introduction of a spurious singularity at finite location, *i.e.* $X_1 \pm X_2 = 0$, in this case there is also a spurious singularity which is introduced, but as $x_2 \longrightarrow +\infty$ – indeed, such a singularity has the same coefficient but with different sign between the two terms. Expanding in the small regulator

$$\widetilde{\Omega}^{(\epsilon)}(\widetilde{\mathcal{Y}},\mathcal{O}_{\mathcal{G}}) = \frac{\ell_1^2}{2y_{12}}\log\left(\frac{X_1+y_{12}}{X_1-y_{12}}\right)\left[-\frac{1}{\epsilon}+\log(\ell(X_2+y_{12}))-2\gamma_{\text{EM}}\right.$$
$$\left.+\frac{1}{\epsilon}-\log(\ell(X_2-y_{12}))+2\gamma_{\text{EM}}+\mathcal{O}(\epsilon)\right], \quad (121)$$

with $\gamma_{\text{EM}}$ the Euler-Mascheroni constant. Note that triangulations of this class completely factorise the two interaction sites, at the price of introducing a spurious singularity in the infra-red.

Let us finally consider the triangulation corresponding to the original "cutting" rules. Such a triangulation shows the part of the adjoint surface contained inside of the optical polytope as the only spurious boundary. As we already discussed, the canonical function of the optical polytope gets divided into four terms

$$\widetilde{\Omega}(\widetilde{\mathcal{Y}}, \mathcal{O}_{\mathcal{G}}) = -\frac{\ell_1^2}{2y_{12}} \sum_{\sigma_1, \sigma_2 = \{\pm\}} \sigma_1 \sigma_2 \int_{X_1}^{+\infty} \frac{dx_1}{x_1 + \sigma_1 y_{12}} \int_{X_2}^{+\infty} \frac{dx_2}{x_2 + \sigma_2 y_{12}}, \quad (122)$$

each of which is a product of two decoupled integrals with logarithmic singularities at infinity. Proceeding as in the previous case

$$
\begin{aligned}
\widetilde{\Omega}^{(\epsilon)}(\widetilde{\mathcal{Y}}, \mathcal{O}_{\mathcal{G}}) &= -\frac{\ell_1^{2-\epsilon}}{2y_{12}} \sum_{\sigma_1, \sigma_2 = \{\pm\}} \sigma_1 \sigma_2 \int_{X_1}^{+\infty} dx_1 \frac{(x_1 - X_1)^{-\epsilon}}{x_1 + \sigma_1 y_{12}} \int_{X_2}^{+\infty} dx_2 \frac{(x_2 - X_2)^{-\epsilon}}{x_2 + \sigma_2 y_{12}} \\
&= -\ell_1^{2-\epsilon} [\Gamma(-\epsilon)\Gamma(1-\epsilon)]^2 \sum_{\sigma_1, \sigma_2 = \{\pm\}} \sigma_1 \sigma_2 (X_1 + \sigma_1 y_{12})^{-\epsilon} (X_2 + \sigma_1 y_{12})^{-\epsilon} \\
&= -\frac{\ell_1^2}{2y_{12}} \sum_{\sigma_1, \sigma_2 = \{\pm\}} \sigma_1 \sigma_2 \left[ -\frac{1}{\epsilon} + \log\left(\ell_1(X_1 + \sigma_1 y_{12})\right) - 2\gamma_{\text{EM}} + \mathcal{O}(\epsilon) \right] \\
&\qquad\qquad\qquad\qquad \times \left[ -\frac{1}{\epsilon} + \log\left(\ell_1(X_2 + \sigma_1 y_{12})\right) - 2\gamma_{\text{EM}} + \mathcal{O}(\epsilon) \right].
\end{aligned}
\quad (123)
$$

A comment is now in order. A superficial analysis would suggest that just when resorting to canonical form triangulations of the optical polytope associated to boundaries intersecting or containing its adjoint surface, the factorised structure at integrand level translates into a factorised structure for the integrated canonical function. However, when we talk about a factorised structure, we should think about sums of terms each of which can be thought of as a product between the physical singularities $(X_j + \sigma_j y_{12})$ $(j = 1, 2, \sigma_j = \pm)$ associated to the two different interaction sites, up to spurious singularities. Note that any expression for an integrated simplex in any of the triangulations has the form

$$
-\frac{2y_{12}}{\ell_1^2} \widetilde{\Omega}(\widetilde{\mathcal{Y}}, \mathcal{O}_{\mathcal{G}}) = \sum_{\{\Sigma_{\mathcal{G}}\}} \left[ f_1^{(2)} \otimes \mathbb{I}_2 + \mathbb{I}_1 \otimes f_2^{(2)} + f_1^{(1)} \otimes f_2^{(1)} \right] = f_1^{(1)}\left(\frac{X_1 + y_{12}}{X_1 - y_{12}}\right) \otimes f_2^{(1)}\left(\frac{X_2 + y_{12}}{X_2 - y_{12}}\right),
$$
(124)

here $f_j^{(k)}$ is a trascendental function of trascendental degree $(k)$ with singularities $X_j + \sigma_j y_{12}$ and, eventually, with an extra (spurious) singularity, while $\mathbb{I}_j$ is just the identity associated to the variable $X_j + \sigma_j y_{12}$. In the example just inspected, $f_j^{(2)}(z_j) := \text{Li}_2(1 - z_j)$ and $f_j^{(1)}(z_j) := \log(z_j)$, and the spurious singularities appeared at $X_1 \pm X_2$ and at infinity, the latter signaled by the presence of $\ell_1$ in the trascendental functions to form dimensionless arguments. The choice of triangulation of the optical polytope $\mathcal{O}_{\mathcal{G}}$ determines the detailed form of the sum with a "choice" of spurious singularity. Notice that the actual integrated function $\widetilde{\Omega}(\tilde{\mathcal{Y}}, \mathcal{O}_{\mathcal{G}})$ has the simple structure $f_1^{(1)} \otimes f_2^{(1)}$: spurious boundaries in the triangulation of $\mathcal{O}_{\mathcal{G}}$ map into spurious singularities that depend on both $\{X_j, j = 1, 2\}$ and the appearence of the structure $f_1^{(2)} \otimes \mathbb{I}_2 + \mathbb{I}_1 \otimes f_2^{(2)}$; the absence of spurious boundaries in the triangulation of $\mathcal{O}_{\mathcal{G}}$ or the spurious boundary associated to the full adjoint surface of $\mathcal{O}_{\mathcal{G}}$, instead, map into a dependence of $f_j^{(k)}(z_j)$ on $z_j = \ell_1(X_j + \sigma_j y_{12})$.

Despite these considerations seem to extend also to more complicated graphs $\mathcal{G}$, at least in the cases which have been checked explicitly, it would be interesting to apply the methods in [24, 54, 59] to the optical polytope $\mathcal{O}_{\mathcal{G}}$ and its triangulations. We leave this to future work.

On a similar line, the class of wavefunctions we studied allow to extract the information about wavefunction coefficients with the exchange of other scalar or spinning states via differential operators [53, 60, 61]. It would be interesting to systematically see how the structure

of the triangulations of the optical polytopes is mapped into the functional structure of the cutting rules for the wavefunction of such propagating states. We also leave this investigation for future work.

## 5.5 From cosmological to flat-space cutting rules

In the previous subsection we have just seen how the cosmological optical theorem straightforwardly emerges from the geometrical structure of the optical polytope as the equivalence among different polytope subdivisions/triangulations.

An important question that needs to be addressed is whether flat-space unitarity can emerge from cosmological unitarity and, if so, how. The usual formulation of the cosmological optical theorem in terms of "cutting rules" (56) does not make it obvious – on the one hand, approaching the total energy singularity, $\psi_{\mathcal{G}}(x_s, y_e)$ reduces to the (high energy limit of the) flat-space scattering amplitude, while $\psi_{\mathcal{G}}^{\dagger}(-x_s, y_e)$ should reduce to its complex conjugate, up to a sign. On the other hand, the right-hand-side of (56) does not show the total energy singularity at all. Therefore, the flat-space limit of the usual formulation of the cosmological optical theorem seems to return the identity $0 = 0$ rather than the Cutkosky cutting rules.

A careful implementation of the $i\epsilon$-prescription allows to identify the left-hand-side of (56) as the total energy singularity is approached with the imaginary part of the flat-space scattering amplitude, and its right-hand-side with the Cutkosky cutting rules. However, the combinatorial-geometrical picture in terms of the optical polytope provides a more transparent way not only of obtaining the flat-space optical theorem but also of relating it to the cosmological one, so in this section we will attack the problem from that point of view.

Before going into the details, one comment is in order. Flat-space unitarity is already encoded into the cosmological polytope $\mathcal{P}_{\mathcal{G}}$, concretely into the vertex structure of its scattering facet $\mathcal{S}_{\mathcal{G}}$ [48]: at any of its facets $\mathcal{S}_{\mathcal{G}} \cap \mathcal{W}^{(\mathfrak{g})}$, with $\mathcal{G}$ and $\mathfrak{g}$ satisfying the codimension-2 compatibility conditions [27, 31, 46], the vertices span three factorised subspaces in $\mathbb{P}^{n_s + n_e - 3}$ – two lower dimensional scattering facets (corresponding to $\mathcal{G} \cap \mathfrak{g} = \mathfrak{g}$ and $\mathcal{G} \cap \overline{\mathfrak{g}} = \overline{\mathfrak{g}}$), and a simplex $\Sigma_{\not{\mathcal{E}}}$ formed by the vertices of the edges $\not{\mathcal{E}}$ connecting $\mathfrak{g}$ and $\overline{\mathfrak{g}}$ identified by a marking close to $\overline{\mathfrak{g}}$. The canonical function on each facet $\mathcal{S}_{\mathcal{G}} \cap \mathcal{W}^{(\mathfrak{g})}$ is therefore given by the product of the canonical functions of these three lower-dimensional polytopes, with the ones of the two scattering facets returning the flat-space amplitudes $\mathcal{A}_{\mathfrak{g}}$ and $\mathcal{A}_{\overline{\mathfrak{g}}}$ associated to $\mathfrak{g}$ and $\overline{\mathfrak{g}}$, and the canonical function of $\Sigma_{\not{\mathcal{E}}}$ giving the measure of the Lorentz invariant phase-space (the location of the vertices, all close to $\overline{\mathfrak{g}}$, specifies the direction of the energy flow, which is incoming for $\mathfrak{g}$ and outgoing for $\overline{\mathfrak{g}}$). From the cosmological polytope perspective, flat-space unitarity arises on the boundaries of its scattering facet, where $\delta(E_{\mathfrak{g}})$ is enforced on the total energy conservation sheet and contributes to the imaginary part of the flat-space amplitude, since $\pi^{-1}\text{Im}\{1/(E_{\mathfrak{g}} - i\epsilon_{\mathfrak{g}})\} = \delta(E_{\mathfrak{g}})$.

From the perspective of the optical polytope $\mathcal{O}_{\mathcal{G}}$, the story has one similarity and one fundamental difference. The former is that the flat-space cutting rules are expected to emerge in codimension-2 as a constraint on the energy conservation sheet: as $\Omega(\mathcal{Y}, \mathcal{O}_{\mathcal{G}}) = \Delta\psi_{\mathcal{G}} = \psi_{\mathcal{G}}(x_s, y_e) + \psi_{\mathcal{G}}^{\dagger}(-x_s, y_e)$, the expectation is that on the total energy conservation sheet $\psi_{\mathcal{G}}(x_s, y_e)$ and $\psi_{\mathcal{G}}^{\dagger}(-x_s, y_e)$ reduce to $\mathcal{A}_{\mathcal{G}}$ and $-\overline{\mathcal{A}}_{\mathcal{G}}$ respectively, returning directly the discontinuities across the singularities of $\mathcal{A}_{\mathcal{G}}$ provided that $\mathcal{A}_{\mathcal{G}}$ and $-\overline{\mathcal{A}}_{\mathcal{G}}$ are equipped with the correct $i\epsilon$-prescription. The fundamentally different aspect is that, contrarily to what happens for the cosmological polytope, the scattering facet is not a boundary of $\mathcal{O}_{\mathcal{G}}$. If the scattering facet is not a boundary of the optical polytope, how can we expect to take the intersection $\mathcal{O}_{\mathcal{G}} \cap \mathcal{W}^{(\mathcal{G})}$ and see a codimension-2 boundary on it? As we will see, the solution to this puzzle will come from the non-convexity of $\mathcal{O}_{\mathcal{G}}$.

**Flat-space unitarity from the geometry of $\mathcal{O}_{\mathcal{G}}$.** Let us begin with considering the convex polytope $\mathcal{Q}_{\mathcal{G}}(a)\,(a>0)$ and its intersection $\mathcal{Q}_{\mathcal{G}}(a)\cap\mathcal{W}^{(\mathcal{G})}$ with the hyperplane $\mathcal{W}^{(\mathcal{G})}=\sum_{s\in\mathcal{V}}\tilde{\mathbf{x}}_{\mathbf{s}}$. As we showed in Section 5.2, such hyperplane intersects the polytope $\mathcal{Q}_{\mathcal{G}}$ *inside*, with the vertices $\{\mathcal{Z}_e^{(2)},\,\mathcal{Z}_e^{(3)},\,\forall\,e\in\mathcal{E}\}$ on $\mathcal{Q}_{\mathcal{G}}(a)\cap\mathcal{W}^{\mathcal{G}}$, and the vertices $\{\mathcal{Z}_e^{(1)},\,e\in\mathcal{E}\}$ and $\{\mathcal{Z}_e^{(4)},\,e\in\mathcal{E}\}$ on the positive and negative half spaces identified by $\mathcal{W}^{(\mathcal{G})}$ respectively. As a consequence: *i)* $\mathcal{Q}_{\mathcal{G}}(a)\cap\mathcal{W}^{(\mathcal{G})}$ is not a facet of $\mathcal{Q}_{\mathcal{G}}(a)$; *ii)* this intersection is *inside* $\mathcal{Q}_{\mathcal{G}}(a)$ and, thus, it is given by a polytope of codimension-1 defined by the convex hull of the vertices $\{\mathcal{Z}_e^{(2)},\,\mathcal{Z}_e^{(3)},\,e\in\mathcal{E}\}$.

Let us now consider $\mathcal{O}_{\mathcal{G}}$. The intersection $\mathcal{O}_{\mathcal{G}}\cap\mathcal{W}^{(\mathcal{G})}$ is still characterised by the vertices $\{\mathcal{Z}_e^{(2)},\,\mathcal{Z}_e^{(3)},\,e\in\mathcal{E}\}$, but now the other vertices $\{\mathcal{Z}_e^{(1)},\,\mathcal{Z}_e^{(4)},\,e\in\mathcal{E}\}$ are all on the positive half-space identified by $\mathcal{W}^{(\mathcal{G})}$. Consequently, not only $\mathcal{O}_{\mathcal{G}}\cap\mathcal{W}^{(\mathcal{G})}$ is not a facet of $\mathcal{O}_{\mathcal{G}}$, as we showed in Section 5.2, but also this intersection does not lie inside $\mathcal{O}_{\mathcal{G}}$. This implies that such an intersection is of codimension higher than 1. In order to understand the codimension of $\mathcal{O}_{\mathcal{G}}\cap\mathcal{W}^{(\mathcal{G})}$, we need to understand how the vertices $\{\mathcal{Z}_e^{(2)},\,\mathcal{Z}_e^{(3)}\}$ organize in the hyperplane $\mathcal{W}^{(\mathcal{G})}$. This translates into the identification of the higher codimension hyperplanes $\mathcal{W}^{(\mathcal{G})}\cap\widetilde{\mathcal{W}}^{(\mathfrak{g}_1\cdots\mathfrak{g}_k)}$ which have non-vanishing intersection with $\mathcal{O}_{\mathcal{G}}$ – as usual $\widetilde{\mathcal{W}}^{(\mathfrak{g}_1\cdots\mathfrak{g}_k)}:=\widetilde{\mathcal{W}}^{(\mathfrak{g}_1)}\cap\cdots\cap\widetilde{\mathcal{W}}^{(\mathfrak{g}_k)}$, where $\widetilde{\mathcal{W}}^{(\mathfrak{g}_j)}$ can be either $\mathcal{W}^{(\mathfrak{g}_j)}$ or $\widehat{\mathcal{W}}^{(\mathfrak{g}_j)}$.

We can begin the analysis with codimension-2 hyperplanes $\mathcal{W}^{(\mathcal{G})}\cap\widetilde{\mathcal{W}}^{(\mathfrak{g})}$ and check which of them are such that $\mathcal{O}_{\mathcal{G}}\cap\left(\mathcal{W}^{(\mathcal{G})}\cap\widetilde{\mathcal{W}}^{(\mathfrak{g})}\right)\neq\varnothing$.[26] First, notice that the codimension-2 hyperplane $\mathcal{W}^{(\mathcal{G})}\cap\widetilde{\mathcal{W}}^{(\mathfrak{g})}$ is equivalently identified by $\widetilde{\mathcal{W}}'^{(\mathcal{G})}\cap\widetilde{\mathcal{W}}^{(\mathfrak{g})}$:

$$\mathcal{W}^{(\mathcal{G})}\cap\widetilde{\mathcal{W}}^{(\mathfrak{g})}\,\sim\,\widetilde{\mathcal{W}}'^{(\mathcal{G})}\cap\widetilde{\mathcal{W}}^{(\mathfrak{g})}\,,\tag{125}$$

where

$$\text{for}\;\;\widetilde{\mathcal{W}}^{(\mathfrak{g})}=\mathcal{W}^{(\mathfrak{g})}\,,\quad\text{then}\quad\widetilde{\mathcal{W}}'^{(\mathcal{G})}=\begin{cases}\widehat{\mathcal{W}}^{(\bar{\mathfrak{g}})}\,,&\text{if}\;\mathfrak{g}\in\mathfrak{G}_{\text{ind}}\,,\\\widehat{\mathcal{W}}^{(\mathfrak{g})}\,,&\text{if}\;\mathfrak{g}\in\mathfrak{G}_{\not{k}}\,,\;\not{k}\in\mathbb{Z}^+\,,\end{cases}\tag{126}$$

$$\text{for}\;\;\widetilde{\mathcal{W}}^{(\mathfrak{g})}=\widehat{\mathcal{W}}^{(\mathfrak{g})}\,,\quad\text{then}\quad\widetilde{\mathcal{W}}'^{(\mathcal{G})}=\begin{cases}\mathcal{W}^{(\bar{\mathfrak{g}})}\,,&\text{if}\;\mathfrak{g}\in\mathfrak{G}_{\text{ind}}\,,\\\mathcal{W}^{(\mathfrak{g})}\,,&\text{if}\;\mathfrak{g}\in\mathfrak{G}_{\not{k}}\,,\;\not{k}\in\mathbb{Z}^+\,,\end{cases}$$

with $\mathfrak{G}_{\text{ind}}$ the set of induced subgraphs of $\mathcal{G}$ – *i.e.* subgraphs of $\mathcal{G}$ whose sites $\mathcal{V}_{\mathfrak{g}}\subset\mathcal{G}$ are connected with each other as in $\mathcal{G}$ – and $\mathfrak{G}_{\not{k}}$ the subsets of subgraphs with the same sites as $\mathcal{G}$ but $\not{k}\in\mathbb{Z}^+$ edges removed keeping the subgraph connected. For reasons that will become clear, the subgraphs $\mathfrak{g}$ which do not belong to either of these classes do not need to be considered.

The projective equivalence (125) can be easily understood by considering the explicit form of the hyperplanes $\mathcal{W}^{(\mathfrak{g})}$ and $\widehat{\mathcal{W}}^{(\mathfrak{g})}$ as given in (81), as well as the form of $\mathcal{W}^{(\mathcal{G})}$:

$$\mathcal{W}^{(\mathcal{G})}=\sum_{s\in\mathcal{V}}\tilde{\mathbf{x}}_s\,,\qquad\mathcal{W}^{(\mathfrak{g})}=\sum_{s\in\mathcal{V}_{\mathfrak{g}}}\tilde{\mathbf{x}}_s+\sum_{e\in\mathcal{E}_{\mathfrak{g}}^{\text{ext}}}\tilde{\mathbf{y}}_e\,,\qquad\widehat{\mathcal{W}}^{(\mathfrak{g})}=\sum_{s\in\mathcal{V}_{\mathfrak{g}}}\tilde{\mathbf{x}}_s-\sum_{e\in\mathcal{E}_{\mathfrak{g}}^{\text{ext}}}\tilde{\mathbf{y}}_e\,.\tag{127}$$

The intersection $\mathcal{W}^{(\mathcal{G})}\cap\mathcal{W}^{(\mathfrak{g})}$ identifies the codimension-2 hyperplane whose defining conditions can be written in local cordinates as the pair of homogeneous equations

$$\mathcal{Y}\cdot\mathcal{W}^{(\mathcal{G})}=\sum_{s\in\mathcal{V}}x_s=0\,,\qquad\mathcal{Y}\cdot\mathcal{W}^{(\mathfrak{g})}=\sum_{s\in\mathcal{V}_{\mathfrak{g}}}x_s+\sum_{e\in\mathcal{E}_{\mathfrak{g}}^{\text{ext}}}y_e=0\,.\tag{128}$$

However, the same codimension-2 hyperplane is identified by any pair of homoegenous equations obtained as a linear combination of (128), in particular by

$$\mathcal{Y}\cdot\widetilde{\mathcal{W}}'^{(\mathcal{G})}=\sum_{\bar{s}\in\mathcal{V}_{\bar{\mathfrak{g}}}}x_{\bar{s}}-\sum_{e\in\mathcal{E}_{\mathfrak{g}}^{\text{ext}}}y_e=0\,,\qquad\mathcal{Y}\cdot\mathcal{W}^{(\mathfrak{g})}=\sum_{s\in\mathcal{V}_{\mathfrak{g}}}x_s+\sum_{e\in\mathcal{E}_{\mathfrak{g}}^{\text{ext}}}y_e=0\,,\tag{129}$$

---

[26]Here we want to emphasize that it is important to first identify a codimension-2 hyperplane and then intersect it with the polytope $\mathcal{O}_{\mathcal{G}}$. This avoids the ambiguity associated to the order with which the hyperplanes are intersected with the polytope, which, from the perspective of the canonical function, is given by the non commutativity in taking two residues.

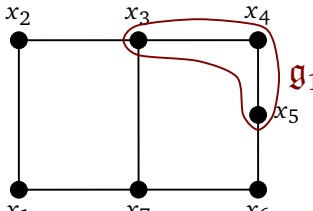 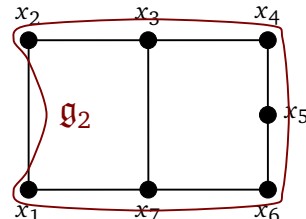 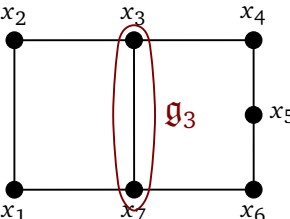

Figure 13: Subgraphs and codimension-2 boundaries. We depict subgraphs whose associated hyperplane, together with $\mathcal{W}^{(\mathcal{G})}$, identifies a codimension-2 face of $\mathcal{O}_{\mathcal{G}}$. The subgraph $\mathfrak{g}_1$ on the left is induced and has a connected $\bar{\mathfrak{g}}_1$, so $\mathfrak{g}_1 \in \mathfrak{G}^{\mathrm{c}}_{\mathrm{ind}}$. The subgraph $\mathfrak{g}_2$ in the center comprises all vertices of $\mathcal{G}$ and all edges but one, so $\mathfrak{g}_2 \in \mathfrak{G}_{\ell}$. The subgraph $\mathfrak{g}_3$ on the right does not belong to either of the previous classes, $\mathfrak{g}_3 \notin \{\mathfrak{G}^{\mathrm{c}}_{\mathrm{ind}}, \mathfrak{G}_{\ell}\}$.

with the latter associated to the hyperplane $\mathcal{W}^{(\mathfrak{g})}$, and the former to $\widetilde{\mathcal{W}}'^{(\mathcal{G})}$, this is, $\widehat{\mathcal{W}^{(\mathfrak{g})}}$ for $\mathfrak{g} \in \mathfrak{G}_{\mathrm{ind}}$ or $\widehat{\mathcal{W}^{(\mathfrak{g})}}$ for $\mathfrak{g} \in \mathfrak{G}_{\ell}$. An analogous reasoning applies when we consider $\mathcal{W}^{(\mathcal{G})} \cap \widehat{\mathcal{W}^{(\mathfrak{g})}}$.

The projective equivalence (125) allows us to formulate our problem in terms of codimension-2 hyperplanes formed as intersections of the codimension-1 hyperplanes that contain the facets of $\mathcal{O}_{\mathcal{G}}$. The intersection between such hyperplanes and $\mathcal{O}_{\mathcal{G}}$ is non empty in codimension-2 if and only if the compatibility condition (84) is satified:

$$\sum_{\mathcal{S}_{\mathfrak{g}}} 1 + \not{n}_{\not{\ell}} = 2\,. \tag{130}$$

This condition selects those hyperplanes among (126) such that if $\mathfrak{g} \in \mathfrak{G}_{\mathrm{ind}}$ then $\bar{\mathfrak{g}}$ is connected, and if $\mathfrak{g} \in \mathfrak{G}_{\not{\ell}}$ then $\not{\ell} = 1$ (see Figure 13). Let us refer to these two set of subgraphs as $\mathfrak{G}^{\mathrm{c}}_{\mathrm{ind}}$ and $\mathfrak{G}_{\ell}$ respectively:

$$\begin{aligned} \mathfrak{G}^{\mathrm{c}}_{\mathrm{ind}} &:= \left\{ \mathfrak{g} \subset \mathcal{G} \,|\, \mathfrak{g} = \mathcal{G}[\mathcal{V}_{\mathfrak{g}}], \bar{\mathfrak{g}} \text{ connected} \right\}\,, \\ \mathfrak{G}_{\ell} &:= \left\{ \mathfrak{g} \subset \mathcal{G} \,|\, \mathfrak{g} = \mathcal{G} \setminus \{e\}, \mathfrak{g} \text{ connected}, \forall\, e \in \mathcal{E} \right\}\,, \end{aligned} \tag{131}$$

with $\mathcal{G}[\mathcal{V}_{\mathfrak{g}}]$ being the subgraph induced in $\mathcal{G}$ by the set of sites $\mathcal{V}_{\mathfrak{g}}$. Any subgraph that does not belong to either of these classes does not satisfy (130). Summing up, we have a non-empty intersection $\mathcal{O}_{\mathcal{G}} \cap \mathcal{W}^{(\mathcal{G})} \cap \widehat{\mathcal{W}^{(\mathfrak{g})}} \neq \varnothing$ if $\mathfrak{g} \in \mathfrak{G}^{\mathrm{c}}_{\mathrm{ind}}$ or $\mathfrak{g} \in \mathfrak{G}_{\ell}$. Importantly, as shown in Section 5.2, these are nothing but *all* the codimension-2 boundaries of the scattering facet! Since, as we argued, $\mathcal{O}_{\mathcal{G}} \cap \mathcal{W}^{(\mathcal{G})} = \varnothing$ in codimension-1, this implies that $\mathcal{O}_{\mathcal{G}} \cap \mathcal{W}^{(\mathcal{G})}$ is a codimension-2 object constituted by the union of all the facets of the scattering facet. Hence, its canonical function can be written distributionally as

$$\begin{aligned} \Omega(\mathcal{O}_{\mathcal{G}} \cap \mathcal{W}^{(\mathcal{G})}) = \delta\left(\mathcal{Y} \cdot \mathcal{W}^{(\mathcal{G})}\right) &\Bigg[ \sum_{\mathfrak{g} \in \mathfrak{G}^{\mathrm{c}}_{\mathrm{ind}}} \left[ \delta\left(\mathcal{Y} \cdot \mathcal{W}^{(\mathfrak{g})}\right) \Omega(\mathcal{S}_{\mathcal{G}} \cap \mathcal{W}^{(\mathfrak{g})}) + \delta\left(\mathcal{Y} \cdot \widehat{\mathcal{W}^{(\mathfrak{g})}}\right) \Omega(\mathcal{S}_{\mathcal{G}} \cap \widehat{\mathcal{W}^{(\mathfrak{g})}}) \right] \\ &+ \sum_{\mathfrak{g} \in \mathfrak{G}_{\ell}} \left[ \delta\left(\mathcal{Y} \cdot \mathcal{W}^{(\mathfrak{g})}\right) \Omega(\mathcal{S}_{\mathcal{G}} \cap \mathcal{W}^{(\mathfrak{g})}) + \delta\left(\mathcal{Y} \cdot \widehat{\mathcal{W}^{(\mathfrak{g})}}\right) \Omega(\mathcal{S}_{\mathcal{G}} \cap \widehat{\mathcal{W}^{(\mathfrak{g})}}) \right] \Bigg]\,, \end{aligned} \tag{132}$$

where $\mathcal{S}_{\mathcal{G}}$ is the scattering facet, and we use this notation in the argument of the canonical functions to emphasize that they are canonical functions of the boundaries $\mathcal{S}_{\mathcal{G}} \cap \mathcal{W}^{(\mathfrak{g})}$. The canonical functions in the right-hand-side of (132) are associated to lower point scattering

amplitudes [48] – see also Section 5.2. In particular

$$\Omega(\mathcal{S}_G \cap \mathcal{W}^{(\mathfrak{g})}) = \left( \prod_{e \in \mathcal{L}} \frac{1}{2y_e} \right) \mathcal{A}[\mathfrak{g}] \times \mathcal{A}[\overline{\mathfrak{g}}], \qquad \text{for } \mathfrak{g} \in \mathfrak{G}^{c}_{\text{ind}},$$

$$\Omega(\mathcal{S}_G \cap \mathcal{W}^{(\mathfrak{g})}) = \mathcal{A}[\mathfrak{g}], \qquad\qquad \text{for } \mathfrak{g} \in \mathfrak{G}_{\cancel{\ell}},$$

(133)

where $\cancel{\ell}$ is the set of edges between $\mathfrak{g}$ and $\overline{\mathfrak{g}}$, which get cut, and the energy flows along $\cancel{\ell}$ from $\overline{\mathfrak{g}}$ to $\mathfrak{g}$. Recall that $\widehat{\mathcal{W}}^{(\mathfrak{g})} \sim \mathcal{W}^{(\overline{\mathfrak{g}})}$ if $\mathfrak{g} \in \mathfrak{G}^{c}_{\text{ind}}$, and then the canonical function $\Omega(\mathcal{S}_G \cap \widehat{\mathcal{W}}^{(\mathfrak{g})})$ is the same as the first line in (133) but now with energies flowing from $\mathfrak{g}$ to $\overline{\mathfrak{g}}$, which means that it can acquire a minus sign.

The first line in (132) precisely corresponds to the flat-space cutting rules! However, there is a second line so the expression (132) seems to contain more information than just the usual Cutkosky rules. It is important to note that the terms in the second line are present if and only if we are considering a loop diagram – recall that $\mathfrak{G}_{\cancel{\ell}}$ is the set of connected subgraphs obtained from $\mathcal{G}$ by just deleting one edge, which cannot exist at tree level as the deletion of one edge would map the graph into a disconnected subgraph. For such terms the delta functions in the second line of (132) force $\sum_{s \in \mathcal{V}} x_s \pm 2y_e$ to vanish. On the intersection with the total energy hyperplane $\mathcal{W}^{(\mathcal{G})}$, this is just $\delta(y_e)$. Indeed, this is not a cut of an amplitude. Let us now show that these terms do not play any important role, as they vanish when the loop integral is performed. To see this it is important to take into account the measure in loop momentum space; in the edge-weight variables it acquires the schematic form [58]

$$d^d l = \prod_{e \in \mathcal{E}'} dy_e \, y_e \, \mu(y), \qquad \mathcal{E}' = \begin{cases} \mathcal{E}, & \text{if } d \geq n_e, \\ \mathcal{E}_d \subset \mathcal{E}, & \text{if } d < n_e, \end{cases}$$

(134)

$\mathcal{E}_d$ and $\mu(y)$ being respectively a $d$-dimensional subset of $\mathcal{E}$ chosen to parametrise the loop momentum when the number of spatial dimension is less than the number of edges of the graph, and a function which can be a ration containing either polynomial or square-roots of polynomial, depending on the dimensions – see [58]. We can see then that when we consider the measure (134) together with the canonical function (132), the second line of (132) shows factors of the type $y_e \, \delta(y_e)$, which vanish! The intersection of the optical polytope with the total energy conserving hyperplane gives rise to the flat-space cutting rules with additional terms that vanish when the loop integration measure is taken into account.

## 6 Conclusion and outlook

Unitarity is one of the fundamental principles that govern the time evolution of physical processes. In the context of scattering amplitudes its implications are understood both in perturbation theory and for the non-perturbative S-matrix, as the optical theorem is generally valid.

In this paper, we have re-examined perturbative unitarity for the Bunch-Davies wavefunction of the universe, under the loupe of the combinatorics of the cosmological polytopes and with a more extensive, but not yet comprehensive, analysis of the $i\epsilon$-prescription. Interestingly, the $i\epsilon$-prescription is encoded into the geometry of the boundaries of the cosmological polytope and can be made explicit via a contour integral representation of its canonical form, where it is fixed by the requirement of the positivity of the geometry and the preservation of the overall orientation. These requirements also imply an analytic continuation of the internal energies, which reminds of the Feynman $i\epsilon$. The information about the cosmological optical

theorem is encoded into a non-convex part of the cosmological polytopes and its triangulations result in the different cutting rules. This formulation also provides a transparent way to see the flat-space optical theorem emerge from the cosmological one. Our work represents a step forward in understanding how fundamental principles are encoded into the Bunch-Davies wavefunctional in perturbation theory, but we are still at the beginning of the road towards a satisfactory understanding. Let us summarise here the main future directions, organising them in a similar way as our main results in the introduction.

**The $i\epsilon$-prescription.** In this paper we emphasised how a correct way to simultaneously obtain convergence of the time integrals and compatibility with unitarity is by analytically continuing the energies to be complex with a small negative imaginary part. This also makes the time-integrals convergent for any real value of the real part of the energies. The geometrical analysis also enforces such an analytic continuation for both external and internal energies. The first one is related to the convergence of the time integral, and assigning a small imaginary part to the external energies only recalls the kinematic $i\epsilon$ discussed in [52] in the context of the flat-space S-matrix. The second one instead is tied to the distributional nature of the bulk-to-bulk propagator and provides a $i\epsilon$-prescription also for the loop propagators, in a similar fashion as the Feynman $i\epsilon$. In flat-space scattering amplitudes, the Feynman $i\epsilon$ implements the notion of causality. It would be interesting to analyse the relation between the class of $i\epsilon$-prescriptions suggested by our analysis and causality. This also implies the necessity of acquiring a deeper understanding of the analytic structure of the wavefunction, a goal towards which the first systematic steps were made in [30]. It is useful to make a parallel with scattering amplitudes. In that context the Feynman $i\epsilon$ is an unphysical parameter which selects the right contour for the propagator consistent with causality, and can be introduced as long as the kinematics is taken to be in the physical region. Said differently, the Feynman $i\epsilon$ deforms the analytic structure of the scattering amplitudes and causality dictates how to approach the branch cuts when the $\epsilon$ is taken to zero and, consequently, how to access the physical region from the correct side (see [52] and references therein). In the wavefunction case, all the poles and branch cuts lie outside of the physical region, so from this perspective it is not clear how the requirement of causality can select a correct way to approach a branch cut. Also, the cosmological optical theorem does not relate, at least naively, the discontinuities along the singular points of the wavefunction to unitarity. So it does not seem obvious how to relate the analytic structure to any of these two fundamental principles. However, as emphasised earlier, the $i\epsilon$-prescription suggested by the cosmological polytope description makes the time integrals well defined for energies running along all the real axis; therefore, the Bunch-Davies wavefunctional can be safely extended outside the physical region (as long as the energies stay real) without modifying its analytic structure. With such an extension, the singularities become accessible in any of these new regions if some of the energies stay positive and others become negative. In this case, the problem becomes similar to the one for scattering amplitudes, and how to approach the branch cut could be dictated by the requirement of causality in these regions.

**Perturbative unitarity and the analytic structure of the wavefunction.** The cosmological optical theorem relates the wavefunction coefficients to their hermitian conjugates with the external energies taken to be negative. It was pointed out in Section 5.3 that if we begin with the "cutting rules" and no additional information, they alone do not imply unitarity and they do not even provide us with information about the analytic structure (they do not compute discontinuities across singularities). The first point is also true for flat-space scattering amplitudes – see [20]. However, in that case the cutting rules come equipped with positivity conditions once we consider all the graphs contributing to the same cut and we sum over all the states

propagating along the cut edges. An issue that, to our knowledge, is not yet understood is whether the positivity condition associated to flat-space unitarity translates into any condition on the wavefunction coefficients. As mentioned in the previous paragraph, the $i\epsilon$-prescription we propose allows to safely extend the wavefunction coefficients to regions were some of the energies become negative. In these regions, the singularities can be accessed and, for states with a flat-space counter-part, their coefficient is related to a flat-space process. Then, the positivity conditions for the flat-space unitary evolution need to reflect into these coefficients. It is important to recall that, for these states, the coefficients of all the singularities are related to flat-space scattering and, consequently, these positivities are not associated just to the total energy singularities.

**The integrated wavefunctional.** The combinatorial description in terms of cosmological polytopes (and optical polytopes as far as the cosmological optical theorem is concerned) provides a transparent picture of the analytic properties of the universal wavefunction integrand (29). The wavefunction coefficients are then obtained by integrating this integrand over the external energies with an appropriate measure that encodes the cosmology and, for loop graphs, over the loop momenta too. For conformally-flat cosmologies with warp factor $a(\eta) = [\ell/(-\eta)]^\gamma$, the measure is a polynomial of degree one in the external energies with a power which depends on the space-time dimensions, the points of the interaction, and the parameter $\gamma$ appearing in the warp factor. The integration over the external energies for such cosmologies produces polylogarithms and combinations of polylogarithms and polynomial [24,59], as long as the power in the measure is an integer. In these cases, the singularity structure of the integrand maps to the singularity structure of the integrated wavefunction. We see this at work in Section 5.4, when we discussed the integration of the cutting rules coming from the triangulation of the optical polytope associated to the two-site line graph. However, a systematic analysis of such integrations has been done just for the cases in which the measure is polynomial of degree zero (*i.e.* a constant) – which is precisely the case we inspected. As for the integration of the cutting rules, we have been cautious in the claim about how the integrand structure extends to integrated functions. It would be interesting to perform a more systematic analysis using the methods discussed in [24,59]: it would allow us to predict, on one side, the result of the integration of the canonical form of the optical polytope, and on the other the integrated cutting rules from the integrand ones. The loop integration, meanwhile, is a territory which has not been much explored. In this case, it is not even clear what the space of functions is that one should expect once both the external energy and the loop integrations have been performed.

**Causality and the analytic structure.** As mentioned in the first paragraph of this section, flat-space causality prescribes how to approach the branch-cuts and, ultimately, the physical region of the S-matrix. Another avatar of causality is provided by the Steinmann relations, which constrain the double discontinuities in the physical region. Similar relations are also valid for the wavefunction universal integrand, and extend to discontinuities of the integrated wavefuction when the external energies are integrated [27]. Understanding the loop integration or at least being able to predict the function that the loop integration produces would allow to extend these Steinmann-like relations to the full, integrated wavefunction. Furthermore, it would be interesting to understand whether they can be related to causality, as it happens in flat-space.

# Acknowledgments

It is a pleasure to thank Daniel Baumann, Hofie Hannesdottir, Austin Joyce, Mehrdad Mirbabayi, Sebastian Mizera, Enrico Pajer and David Stefanyszyn for valuable discussions. P.B. would like to thank Elias Kiritsis, Francesco Nitti and the AstroParticle and Cosmology Laboratory (APC) as well as the organisers of *Amplitudes 2022* for the possibility of presenting the results reported in this paper. C.D.P. would like to thank the Max-Planck-Institut für Physik for their hospitality during the first stages of this project. We would also like to thank the developers of SageMath [62], Maxima [63], Polymake [64–67], TOPCOM [68], and Tikz [69].

**Funding information**   P.B. is supported by the European Research Council (ERC) under the European Union's Horizon 2020 research and innovation programme (grant agreement No 725110), *Novel structures in scattering amplitudes*. S.A. is supported by a VIDI grant of the Netherlands Organisation for Scientific Research (NWO) that is funded by the Dutch Ministry of Education, Culture and Science (OCW). C.D.P. is also supported by a VIDI grant with Project No. 680-47-535.

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
