# Peer review of "Perturbative Unitarity and the Wavefunction of the Universe"

_SciPost Physics, doi:SciPost Phys. 16, 157 (2024)_

## Round 1 · Referee Report · Anonymous (Referee 1) · 2024-1-30

Strengths

1- Settles the connection between cutting rules in flat space for Scattering amplitudes and the cutting rules for the cosmological wavefunction, showing that they agree on the ``total energy pole".

2- Introduces a new cosmological polytope, the ``optical polytope," that has interesting properties on its own right. The authors also achieve to geometrize the constraints of unitarity of the wavefunction through the polytope.

Weaknesses

None.

Report

This is a beautiful paper that addresses an apparent clash between the constraints of unitarity in flat space and cosmology. The cosmological wavefunction is supposed to match a scattering amplitude as the sum of energies goes to zero, and yet, before the authors' work, this limit didn't commute with application of the ``cosmological optical theorem." The resolution is a careful treatment of the i\epsilon prescription for the wavefunction, while maintaining unitarity. Along the way, the authors uncover a new ``optical polytope" that makes many properties of the cutting rules manifest. It's an important paper and I'm happy to recommend it for publication at SciPost.

Requested changes

I mostly have some questions for the authors, rather than requests:

1- In the Baumgart&Sundrum proposal of deforming the Hamiltonian, it seems that depending on the sign of \epsilon, the resulting time evolution operator is ill-defined, as it blows up at past infinity. Yet the wavefunction seems to have no singularity as one flips the sign of \epsilon. What is the explanation for this?

2- The simple prescription seems to ultimately be to shift the external and internal energies by -i \epsilon. What happens to dot products of spatial momenta? Are they untouched? If so, is it always clear how to identify energies versus dot products of spatial momenta?

3- Related to the question above: if the cosmological correlator is specified by a differential equation rather than an explicit rational function, is the i\epsilon prescription implemented through the boundary conditions of the differential equation? The flat space wavefunction satisfies such differential equations, so perhaps the authors thought about it within the class of models considered in the paper.

I'd be happy to see some of these questions addressed by the authors, or, if they have been addressed, a pointer from the authors. Either way, these are the requests of a curious reader rather than a demanding referee.

  • validity: top
  • significance: top
  • originality: top
  • clarity: top
  • formatting: perfect
  • grammar: excellent

Author:  Carlos Duaso Pueyo  on 2024-04-15  [id 4419]

(in reply to Report 1 on 2024-01-30)
Category:
answer to question

Dear referee,

Thank you very much for the valuable report. Following are the answers to the questions formulated in it:

  1. This is a matter of the order of the operations. If one fixes the iε-prescription in such a way that the evolution operator converges, one obtains an unambiguous answer in perturbation theory. But if starting from the perturbation theory answer one changes the iε-prescription there, then the wavefunction is just taken on a different Riemann sheet (i.e. on a different side of the branch cuts), and it does not come from the usual evolution operator (hence, it is not the object we are interested in). As pointed out in the report, if one does this change of sign of ε at the level of the evolution operator, it simply does not converge.

  2. For scalar theories, the wavefunction coefficients can be written in terms of the momentum magnitudes (x,y), so the different iε prescriptions are most naturally formulated in these variables. In any case, the dot products of external momenta are parametrised by the y's which are associated to the tree structure of a graph. Hence, at tree-level, when one uses the iε prescription on the y's, one is shifting the dot products. At loop level and if there is no tree structure, then the y's parametrise the loop momentum and the iε on the y's is really like the Feynman iε for flat-space loops (which is applied on the loop momenta) and the dot products of external momenta stay untouched. If instead at loop level one also has a tree structure, one is shifting both the dot products and the loop momenta.

  3. This is a very interesting question but, regrettably, we do not have a deep answer for it at the moment. Just as a comment we will note that the boundary conditions in the differential equations fix the singularity structure of the final answer, so they ought to be compatible with the iε prescription (as otherwise one could obtain a different type of function).

---

## Round 1 · Referee Report · Anonymous (Referee 2) · 2024-2-10

Strengths

  1. The paper is written in a way so as to make the optical polytope and its boundary structure accessible to non-experts.
  2. The different cuts in the optical theorem of the wavefunction are understood as triangulations of the (non-convex) optical polytope.
  3. The flat space optical theorem is recovered from the non-convex polytope.
  4. The ie prescription for the internal and external energies can be determined from requiring consistency with certain positivity and orientation conditions.

Weaknesses

One missing discussion is: Does the cosmological polytope provide the simplest way to address the unitarity of the wavefunction/correlator? Or does it provide an advantage compared to other methods?

Report

I recommend this paper for publication on Scipost. It is clearly written and provides a novel framework for addressing unitarity and the optical theorem for the wavefunction of the universe.

Requested changes

Depending on how involved the discussion is, the answers to the following questions may either be added to the current manuscript, the report response or future work:

  1. When discussing the ie-prescription in the cosmological polytope, for example in Section 4, what allows one to ignore the possibility of non-zero residues at infinity?
  2. Is it clear how much of the singularity structure of the cosmological polytope explored in this paper is inherited by the integrated result i.e. the wavefunction of the universe itself?
  3. The optical polytope is a non-convex one, and as the authors note there is not an extensive literature on them. Presumably the set of possible optical polytopes is a subset of the set all non-convex ones. Is there a principle that allows one to distinguish which ones will be the former?
  4. My understanding is that the residues on the spurious singularities introduced in the optical polytope are supposed to vanish upon summation. Does this refer to summation across the different diagrams?

  • validity: high
  • significance: good
  • originality: high
  • clarity: high
  • formatting: excellent
  • grammar: excellent

Author:  Carlos Duaso Pueyo  on 2024-04-15  [id 4420]

(in reply to Report 2 on 2024-02-10)
Category:
answer to question

Dear referee,

Thank you very much for the valuable comments, questions and criticism.

We would first like to make a comment on the "Weaknesses" paragraph of the report. In our opinion, we have stressed the three main advantages of the polytope point of view on unitarity: 1. that it provides a natural iε prescription, 2. that it makes explicit the existence of a large number of cutting rules, and 3. that it makes explicit how the S-matrix cutting rules emerge from the wavefunction ones. There is a fourth advantage which is admittedly less emphasised in the manuscript: it makes explicit how this notion of perturbative unitarity is less constraining with respect to flat-space perturbative unitarity. The latter constrains scattering amplitudes at its discontinuities. In the wavefunction case, the cosmological polytope already contains all the singularity structure that one ascribes to the integrand of the wavefunction, with the singularity structure of the integrated function coming from how the contour of integration intersect the surfaces identifies by both the integration measure and the singularities of the integrand. There is no novel relation between the discontinuities of the wavefunction and the optical polytope/optical theorem.

Following are the answers to the points raised in the "Requested changes" paragraph:

  1. The discussion of the iε prescription is based on the integral representation Eq. (4.1). It shows a delta function with dimension equal or lower than the number of integration variables. In the first case, the delta localises completely the integral and hence no issue about eventual poles at infinity arises. In the second case, it localises a subset of integration variables via linear relations. This implies that the resulting integrand is a rational function with the denominator being a polynomial of degree higher than the numerator—more concretely, the denominator has the degree equal to the original number of integration variables, while the numerator is of degree zero (i.e. it does not depend on the integration variables). Hence, the integrand vanishes as any of the integration variables is taken to infinity along any given complex direction.

  2. This point is discussed in Section 5.4 of the manuscript. There we integrated the cutting rules for a simple tree-level graph, finding that physical singularities, zeroes, and spurious singularities map to physical singularities, zeroes, and spurious singularities of the integrated functions, respectively. A more systematic analysis going beyond the case inspected here is left for future work (see outlook in Section 6).

  3. The optical polytope has a definition and that is what distinguishes it from a random non-convex polytope. To draw a parallel with the cosmological polytopes, one can ask the same question: given a set of convex polytopes, how can we know which one is a cosmological polytope? The answer is "if it obeys its definition", which can be put in two forms: a) If it can be constructed out of intersection of a collection of triangles along at most two out of the 3 sides of each of them (i.e. if their vertices satisfy specific linear relations) or b) if it satisfies specific compatibility conditions—and the reason for this, is that in these cases they uniquely (up to the usual overall constant), fix the adjoint surface. One can follow the very same line of reasoning for the optical polytope and test if a candidate non-convex polytope is such that: a) Its vertices satisfy the linear relations of Eqs. (5.10)-(5.11) (in the limit a->-1) or b) its facets satisfy the compatibility conditions derived in Section 5.2—which again uniquely (up to an overall normalisation constant) fix its adjoint surface.

  4. Spurious singularities can be introduced whenever a polytope subdivision is taken. A polytope corresponding to a diagram can be subdivided in different ways into sets of polytopes (if the subdivisions are simplices this is called triangulation). Some (but not all) subdivisions of the optical polytope introduce spurious boundaries, that manifest as spurious singularities in the corresponding canonical forms. When summing over the canonical forms of the elements of a given subdivision these singularities cancel. Hence, this phenomenon happens at the level still of a single graph.

---

## Editorial Decision

published